# Classification of Sea Ice Types in Sentinel-1 SAR images

Jeong-Won Park[1,2], Anton A. Korosov[1], Mohamed Babiker[1], Joong-Sun Won[3], Morten W. Hansen[1,4], Hyun-cheol Kim[2]

[1]Ocean and Sea Ice Remote Sensing Group, Nansen Environmental and Remote Sensing Center, Bergen, 5006, Norway
[2]Unit of Arctic Sea Ice Prediction, Korea Polar Research Institute, Incheon, 21990, South Korea
[3]Department of Earth System Sciences, Yonsei University, Seoul, 03722, South Korea
[4]Department of Remote Sensing and Data Management, Norwegian Meteorolotical Institute, 0371 Oslo, Norway

*Correspondence to*: Jeong-Won Park (jeong-won.park@kopri.re.kr)

**Abstract.** A new Sentinel-1 image-based sea ice classification algorithm using a machine learning-based model trained in a
semi-automated manner is proposed to support daily ice charting. Previous studies mostly rely on manual work in selecting training and validation data. We show that the readily available ice charts from the operational ice services can reduce the amount of manual work in preparation of large amounts of training/testing data. Furthermore, they can feed highly reliable data to the trainer by indirectly exploiting the best ability of the sea ice experts working at the operational ice services. The proposed scheme has two phases: training and operational. Both phases start from the removal of thermal, scalloping, and
textural noise from Sentinel-1 data and calculation of gray level co-occurrence matrix and Haralick texture features in a sliding window. In the training phase, the weekly ice charts are reprojected into the SAR image geometry. A random forest classifier is trained with the texture features on input and labels from the rasterized ice charts on output. Then, the trained classifier is directly applied to the texture features from Sentinel-1 images operationally. Test results from the two datasets spanning winter (Jan-Mar) and summer (Jun-Aug) seasons acquired over the Fram Strait and the Barents Sea showed that the classifier is
capable of retrieving three generalized cover types (open water, mixed first-year ice, old ice) with overall accuracies of 87% and 67% in winter and summer seasons, respectively. For summer season, the classifier failed in distinguishing mixed first-year ice from old ice with accuracy of only 12%; however, it performed rather like an ice-water discriminator with high accuracy of 98% as the misclassification between the mixed first-year ice and old was among themselves. The accuracy for five cover types (open water, new ice, young ice, first-year ice, old ice) in winter season was 60%. The errors are attributed
both to incorrect manual classification on the ice charts and to the semi-automated algorithm. Finally, we demonstrate the potential for near-real time service of the ice map using daily mosaiced Sentinel-1 images.

## 1 Introduction

Wide swath SAR observation from several spaceborne SAR missions (RADARSAT-1, 1995-2013; Envisat ASAR, 2002-2012; ALOS-1 PALSAR, 2006-2011; RADARSAT-2, 2007-present; Sentinel-1, 2014-present) played an important role in
studying global ocean and ice-covered Polar Regions. The Sentinel-1 constellation (1A and 1B) is producing dual-polarization

observation data with the largest Arctic coverage and the highest temporal resolution ever. The cross-polarization is known to be more sensitive to the difference in scattering from sea ice and open water than the co-polarization (Scheuchl et al., 2004), and the combination of HH- and HV-polarizations has been widely used for ice edge detection and ice type classification (a nice overview is given in the paper by Zakhvatkina et al., 2019). However, most of the recent ice classification algorithms were developed using RADARSAT-2 ScanSAR (Leigh et al., 2014; Liu et al., 2015; Zakhvatkina et al., 2017) which has different sensor characteristics from Sentinel-1 TOPSAR, and the use of Sentinel-1 for the same purpose is very limited in literature. The main drawback of applying existing algorithms to Sentinel-1 TOPSAR data is the relatively high level of thermal noise contamination and its propagation to image textures.

For proper use of dense time-series of Earth observations using SAR sensors, radiometric properties must be well-calibrated. Thermal noise is often neglected in many cases, but can seriously impact the utility of dual-polarization SAR data. Sentinel-1 TOPSAR image intensity is particularly disturbed by the thermal noise in the cross-polarization channel. Although the European Space Agency (ESA) provides calibrated noise vectors for noise power subtraction, residual noise contribution is still significant considering the relatively narrow backscattering distribution of the cross-polarization channel. In our previous study (Park et al. 2018), a new denoising method with azimuth de-scalloping, noise scaling, and inter-swath power balancing was developed and showed improved performance in various SAR intensity-based applications. Furthermore, when it came to texture-based image classification, we suggested a correction method for textural noise (Park et al., 2019) which distorts local statistics, and thus degrades texture information in the Sentinel-1 TOPSAR images.

In many of the previous studies on ice-water and/or sea ice classification (Soh and Tsatsoulis, 1999; Zakhvatkina et al., 2013; Leigh et al., 2014; Liu et al., 2015; Ressel et al., 2015; Zakhvatkina et al., 2017; Aldenhoff et al., 2018), the training and validation were done using manually produced ice maps. Although the authors claimed that the manual ice maps were drawn by ice experts, the selection of SAR scenes and interpretation could be inconsistent, and the number of samples might not be enough to generalize the results because of the laborious manual work. Furthermore, the results are hardly reproducible by others because the reference sources are not open to public. Therefore, increasing objectivity is crucial, and automating the classification process is encouraged. The idea of training using SAR images and accompanying image analysis charts, which is a direct manual interpretation of SAR images by trained ice analysts working at operational ice services, were tested for sea ice concentration estimation by Wang et al. (2017); however, such image analysis charts are not accessible to the public.

The use of a public ice chart as training and validation reference data may help in solving the validation problem. The preparation of a public ice chart is also through manual inspection of various sources of satellite imagery and other sources of data (Partington et al., 2003; Johannessen et al., 2006); however, training using a large volume of these charts would reduce operator-to-operator bias, such as inconsistent decisions against similar ice conditions. The overall bias may exist since the public ice charts are produced in the interest of marine safety. Nevertheless, as the human interpretation available in the ice chart is currently considered as the best available information of sea ice (Karvonen et al., 2015), the best practice to make a sea ice type classifier is to train with the public ice chart so that the best knowledge of ice analysts is mimicked.

In this work, we present a semi-automated Sentinel-1 image-based sea ice classification algorithm which takes advantage of our denoising method. The noise corrected dual-polarization images are processed into image textures that capture sea ice features in various spatial scales, and they are used for supervised classification with a random forest classifier by relating with ice charts published by operational ice services. The use of ice charts has dual purposes: semi-automatization of classifier training, and minimization of human error.

## 2 Data and methods

### 2.1 Study area and used data

The region of study for developing and testing the proposed algorithm is the Fram Strait and the Barents Sea including a part of the Arctic Ocean (10ºW-70ºE, 75ºN-85ºN) as shown in Figure 1. Various sea ice types are found in this area due to the intensive export of multi-year ice through the Fram Strait (Smedsrud et al., 2017), and development of young and first-year ice between Svalbard and Franz Josef Land.

Sentinel-1 TOPSAR data in Extended Wide-swath (EW) mode acquired in summer (June-August in 2016-2018) and winter (January-March in 2017-2019) seasons were collected from the Copernicus Open Access Hub (https://scihub.copernicus.eu). The number of daily image acquisitions covering the study area ranges from 6 to 10 depending on the orbits. The images from the first two years (hereafter called DS1) are used to train the classifier and those from the third year (hereafter called DS2) are used for validation.

The ice charts covering the same periods were collected. There are two ice services that publish weekly ice charts with Pan-Arctic coverage: U. S. National Ice Center (NIC) of the United States of America, and Arctic and Antarctic Research Institute (AARI) of Russia. Although the accuracies are known to be comparable (Pastusiak, 2016) to each other, there is no partial ice concentration information in the AARI ice chart. In this study, we use the ice charts downloaded from the NIC website (https://www.natice.noaa.gov/Main_Products.htm). The NIC ice products are produced primarily using radar, microwave radiometer, scatterometer, visible, and infrared imagery from a variety of sources. In addition to imagery, drifting buoy data, ice model predictions, limited ship reports, meteorological and oceanographic observations, and ice information provided by other international centers are also used to make a comprehensive analysis of ice conditions (WMO, 2017).

### 2.2 Methods

Figure 2 shows the flow of the semi-automated ice classification scheme that we propose. It is divided into two phases: training and operational. Both phases start from the removal of thermal noise from Sentinel-1 data (Section 2.2.2), incidence angle calibration (Section 2.2.3), and calculating texture features (Section 2.2.4). The training phase (shown in gray in Figure 2) continues with preprocessing and collocation of the ice charts with the Sentinel-1 data (Section 2.2.1) and machine learning step (Section 2.2.5 and 2.2.6). The operational phase uses the classifier of which developed during the training phase for

processing texture features that were computed from the input SAR data and for generating ice charts. Detailed explanations for each step are given in the following subsections.

### 2.2.1 Ice chart preprocessing

To take advantage of the objective identification of the ice type from expert sources yet open to public and to develop a semi-automated processing scheme, the proposed algorithm uses electronic ice charts published by international ice chart services. The electronic ice chart follows SIGRID-3 format (JCOMM, 2014a), which is based on a vector format called shapefile (ESRI, 1998). The first step is to reproject the ice chart into the geometry of each SAR image. Although an accurate reprojection needs several pieces of information such as orbit, look angle, topographic height, etc., our interest is in the sea ice where the topographic difference does not exceed more than few meters, hence the reprojection of coordinates of ice chart polygons is done with Geospatial Data Abstraction Library (GDAL; GDAL/OGR contributors, 2019) using a simple 3rd order polynomial fitted using the ground control points information from the Sentinel-1 product-included auxiliary data.

After the reprojection, the following three layers are extracted: total ice concentration (CT), partial ice concentration of each ice type (CP), and stage of development (SoD). CT is important because areas with low CT can be misinterpreted as open ocean in a SAR image. Heinrichs et al. (2006) reported that the ice edge determined from AMSR-E passive microwave radiometer data using the isoline of 15% concentration matches best the ice edge determined from RADARSAT-1 SAR data using visual inspection. After the visual comparison of many SAR images and the corresponding reprojected ice charts, we set a threshold of 20% for CT to discard water-like pixels. Note that the ice concentration label in the SIGRID-3 format is assigned in an increment of 10%. CP is also important in finding the dominant ice type in the given polygons. SoD is a so-called ice type. It is challenging to differentiate ice types using SAR data only, thus we merged the SoDs into simple five classes: open water, new ice, young ice, first-year ice, and old ice. For summer season (Jun-Aug), there is almost no new ice and young ice annotated in the NIC ice chart, the SoDs are further merged into three classes: open water, mixed first-year ice, and old ice.Figure 3 demonstrates an example of the ice chart preprocessing explained above with the colors following the WMO nomenclature (JCOMM, 2014b). Comparing the original SoD in the top left panel with the processed SoD in the bottom left panel, it is clear that the ice edge of the processed SoD match better with the SAR backscattering images.

### 2.2.2 Denoising of Sentinel-1 imagery

Sentinel-1 cross-polarization images suffer from strong noise, some which originate from combined effects of the relatively low signal-to-noise ratio of the sensor system and insufficient noise vector information in the Extra Wide-swath mode Level-1 product (Park et al., 2018). For surfaces with low backscattering such as calm open water and level sea ice without the presence of frost flowers on top, the effects from thermal noise contamination are visible not only in the backscattering image but also in some of the texture images (Park et al., 2019). The authors have developed an efficient method for textural denoising which is essential for the preprocessing of Sentinel-1 TOPSAR dual polarization products. Denoising ensures beam-normalized texture properties for all subswaths, which helps seamless mosaic of multi-pass images regardless of the satellite

orbit and image acquisition geometry. By following the methods developed in Park et al. (2018, 2019), each of the Sentinel-1 images was denoised before further processes are applied. As the noise power subtraction yields negative intensity values where the backscattering power is close to the noise floor, more often in HV polarization which has lower backscatter than in HH polarization, we added mean of the noise power back to the denoised result so that those pixels do not turn into NaN (not a number) by the sigma naught conversion of linear scale to log scale (decibel).

### 2.2.3 Incidence angle correction

It is well known that there is a strong incidence angle dependence in the SAR backscattering intensity for open water and sea ice surface (Mäkynen et al., 2002; Mäkynen and Karvonen, 2017). For wide-swath SAR system like Sentinel-1 TOPSAR, varying backscatter intensity confuses image interpretation. The quasi-linear slopes in the plane of incidence angle versus sigma nought in decibel scale are reported as -0.24 and -0.16 dB/degree for typical first-year ice (Mäkynen and Karvonen, 2017), -0.27 and -0.26 dB/degree for level first-year ice, and -0.23 and -0.23 dB/degree for multi-year ice (Lohse et al., 2020) in HH- and HV-polarization, respectively. To normalize the backscattering intensity for all swath range, these slopes were compensated for or used as input layer in several ice classification algorithms in the literature (Liu et al., 2015; Zakhvatkina et al., 2013, 2017; Karvonen, 2014, 2017; Aldenhoff et al., 2018). Although the angular dependency is not a system-dependent variable but is governed by physical characteristics of the backscattered surface, the numbers need to be reassessed because the estimations of Mäkynen and Karvonen (2017) might have been affected by the residual thermal noise which used to be very strong before the ESA has updated the noise removal scheme in 2018 (Miranda, 2018).

Figure 4 shows incidence angle dependence in the SAR backscattering intensity for mixed sea ice types. From the Sentinel-1 dataset described in Section 2.1, sea ice pixels were extracted by using daily global sea ice edge products available from the EUMETSAT Ocean and Sea Ice Satellite Application Facilities (OSI SAF). For mid-winter season (Jan-Mar displayed by blue background), the estimated mean slope in HH polarization was -0.21 dB/degree, which is slightly different from the estimation of the first-year ice (-0.24 dB/degree) in Mäkynen and Karvonen (2017) and in between the estimations for first-year ice (-0.22 dB/degree) and multi-year ice (-0.16 dB/degree) in Mahmud et al. (2018). For HV polarization, the estimated slope was only -0.06 dB/degree, which is much lower than -0.16 dB/degree for deformed first-year ice in Mäkynen and Karvonen (2017), however, it is in line with the estimations in Liu et al., 2015. Work by Leigh et al. (2014) stated that the HV polarization backscatter signatures are largely unaffected by incidence angle variation in their RADARSAT-2 dataset. For summer season (Jun-Aug displayed by red background), the mean slopes increased to -0.28 and -0.08 dB/degree in HH and HV polarization, respectively. Scharien et al. (2014) reported significant slopes for ice adjacent to melt ponds in June, and Gill et al. (2015) also found slopes of -0.33 and -0.25 for smooth first-year ice in May in HH and HV polarization, respectively. The smaller slopes in our estimation are likely due to the mixed ice types and structures; the SAR backscattering of deformed ice has lower incidence angle dependency as shown in Mäkynen and Karvonen (2017).

We compensate for the incidence angle dependence using the estimated slopes with respect to the nominal scene center angle of 34.5 degrees as reference. Although the incidence angle dependence changes with ice type and radar frequency (Mahmud

et al., 2018), the compensation is done for all pixels in the image using a single value of mean slope because the ice types are not identified in this stage. Open water areas of the image are also affected; however, the correction is also beneficial since the incidence angle dependence for open water is stronger (-0.65 dB/degree for wind velocity of 5 m/s, computed from CMOD5 C-band geophysical model function in Hersbach et al., 2007), thus the corrected image has less incidence angle dependence.

**2.2.4 Texture feature computation**

Like many of the previously developed sea ice type classification methods (Shokr, 1991; Barber and LeDrew, 1991; Soh and Tsatsoulis, 1999; Deng and Clausi, 2005; Zakhvatkina et al., 2013; Leigh et al., 2014; Liu et al., 2015), the proposed approach starts from gray level co-occurrence matrices (GLCM) calculation. The GLCM is a four-dimensional matrix $P(i, j, d, a)$ calculated from two grey tones of reference pixel $i$, and its neighbor $j$, with co-occurrence distance $d$, and orientation $a$.

Haralick et al. (1973) have introduced a set of GLCM-based texture features called Haralick features, and its practicality has been reported in numerous research. Since 13 Haralick features can be calculated for each of the two-dimensional slices $P(i, j)$ for multiple $d$ and $a$, the maximum number of texture features is to be as $2 \times 13 \times d \times a = 26da$, where 2 is for accounting dual polarization. It is common to take the directional average for 0°, 45°, 90°, and 135° to reduce GLCM dimensionality. Further averaging for multiple distances (1 to $w/2$ where $w$ is the size of subwindow for GLCM computation) is taken after

computing normalized GLCM. The spatial resolution of the texture features is the pixel spacing of Sentinel-1 EW-mode GRDM image (40 m) multiplied by $w$. In this study, we set $w$ as 25 so that the grid spacing of the result of texture analysis is 1 km.

An important factor that influences the computed texture features is the number of gray levels, $L$. Considering the radiometric stability of Sentinel-1 EW mode (0.32 dB; Miranda, 2018) and the range of sigma nought for various ice types (-31 to 0 dB

for HH, -32 to -7 dB for HV; estimated from DS1 and DS2 after incidence angle correction), the number of gray level should be sufficiently large enough to capture their actual differences in sigma nought values. The optimal quantization level can be calculated using the ratio of sigma nought range to radiometric resolution as follows:

For HH, $\qquad \frac{(0\ dB) - (-31\ dB)}{0.32\ dB} = 96.875$ $\qquad\qquad\qquad\qquad\qquad\qquad$ (1)

For HV, $\qquad \frac{(-7\ dB) - (-32\ dB)}{0.32\ dB} = 78.125$ $\qquad\qquad\qquad\qquad\qquad\qquad$ (2)

Since $L$ should be sufficiently large to take the full advantage of system capability and yet, the computation cost should not be too expensive, in this study, we set $L$ as 64, which is the closest power of 2 to the resulting numbers from the equations above. In addition to 13 Haralick features, the coefficient of variation (CV) which is reported as a useful feature for ice-water

discrimination (Keller et al., 2017) is included. The CV is defined as follows:

$CV = \sigma/\mu$ $\qquad\qquad\qquad\qquad\qquad\qquad\qquad\qquad\qquad\qquad\qquad\qquad\qquad\qquad\qquad$ (3)

where $\sigma$ and $\mu$ are the standard deviation and mean of the samples in a given subwindow. Since CV can be computed for each polarization image, the number of texture features for Sentinel-1 dual-polarization data is extended to 28. Incidence angle and day-of-the-year can also be added. The former is adopted to account for possible residuals from the angular dependency

correction while the latter is to account for seasonal variability. Although these two are not any type of textures, they can be used as input features for image classification. Note that it is important to have each ice type spatially and temporally even distributions if these two additional features are included; otherwise, the trained classifier will result a biased prediction. The effects of including these extra features will be tested and discussed in later sections.

### 2.2.5 Machine learning classifier

Since there are hundreds of algorithms in the field of machine learning (ML) and each of the different algorithms has its own pros and cons, it is not easy to compare their performances and decide what to use. Fernández-Delgado et al. (2014) evaluated that the Random Forest (RF; Ho, 1998) was the best classifier for various types of datasets with slight difference from Support Vector Machine (SVM; Cortes and Vapnik, 1995). In the previous studies about sea ice classification (e.g., Leigh et al., 2014; Liu et al., 2015; Zakhvatkina et al., 2017), the SVM was used often because by nature it works relatively well even when the

number of datasets are small. When training dataset is prepared by manual work (i.e., manual classification by human expert), the number of images is not large, usually less than 20 (e.g., 12 scenes in Zakhvatkina et al., 2013; 20 scenes in Leigh et al., 2013; 1 scene in Liu et al., 2015; 4 scenes in Ressel et al., 2015). However, the number can increase with less effort when the readily available ice charts are used as training references. Besides, there is no need to rely on additional manual work prone to contamination by biased decisions. The RF has two practical advantages when processing a large number of datasets. First,

the RF is scale-invariant and does not require preprocessing of the datasets whereas the SVM requires scaling and normalization. Second, the computational complexity of the RF is lower than that of the SVM. For the SVM, the number of operations is $O(n^2 p + n^3)$ and $O(n_{sv} p)$ for training and prediction while for RF, $O(n^2 p n_{tr})$ and $O(n_{tr} p)$, respectively, where $n$ is the number of samples, $p$ is the number of features, $n_{sv}$ for the number of support vectors, and $n_{tr}$ for the number of trees. Considering the practical requirements of fast processing for near-real time ice charting services, the RF can be a

reasonable solution. We use the RF with the Python Scikit-Learn implementation (Pedregosa et al., 2011).

We split the RF classifier into several binary classifiers using a one-vs-all scheme (Anand et al., 1995). Although the standard RF algorithm can inherently deal with a multiclass problem, the one-vs-all binarization to the RF results in better accuracy with smaller forest sizes than the standard RF (Adnan and Islam, 2015).

Three hyperparameters of the RF classifier were tuned: number of trees ($N_T$), maximum tree depth ($D$), and maximum number

of features ($N_F$). Usually, with the higher $N_T$ and $D$, the model better fits to the data. However, increasing forest size can slow down the training process considerably, and more importantly, it can cause overfitting. Therefore, it is important to tune these hyperparameters adequately so that the processing time and performance are in balance. To determine the best values of the hyperparameters, a grid search with five-fold cross-validation (Kohavi, 1995) is used. The grid (all possible combinations of

$N_T$, $D$, and $N_F$ values) is set in a logarithmic scale (Table 1) because the performance change with hyperparameter is typically in a logarithmic scale. Classification scores with values ranging from 0 (worst performance) to 1 (best performance) are evaluated for each node of the grid and are interpolated between the nodes by curve fitting. The Richards' Curve (Richards, 1959) was used as the fit model because it allows easy estimation of the model's maximum value. The optimal values for $N_T$,

$D$, and $N_F$ are selected based on the saturation of score increment, difference between training and testing score, and computational load considerations.

### 2.2.6 Training and validation

To train an ice type classifier, a set of collocated SAR images and ice charts is required. After the preprocessing of the ice chart including reprojection into the SAR image geometry, only the samples with spatially and temporally good matches should

be fed to the training phase. The goodness of matching should be examined as the weekly ice chart is produced by merging information from many image sources acquired in different time instances, hence the ice locations and conditions are unlikely match to those in every SAR image. As no explicit scene identifier or time information of the images used in ice charting is provided with the ice chart itself, the basic strategy in image selection is to find a pair of SAR image and ice chart which match well visually. Such an image selection is trivial, but not easy to automate. Since the weekly ice chart is made partly based on

the SAR images acquired in the past three days from the date of publication, the ice edges in some images match well with those in the ice chart.

In order to automate image selection, the ice edges in SAR images need to be identified first. Since even an ice/water classifier has not been well developed yet for Sentinel-1, the image selection procedure has to be done manually in the beginning. However, once a classifier is generated with high accuracy, it can be used to automate the procedure, then the whole process

in the proposed scheme will be fully automated. This is why the proposed algorithm is named "semi-" automated for now. Nevertheless, the manual selection to guarantee a "good match" is done by visual inspection of ice-water boundaries overlaid on SAR images. The ice-water boundary can be extracted easily from the reprojected ice chart. Then the SAR backscattering image contrasts across the ice-water boundaries are examined both in HH- and HV-polarization because the image contrast between ice-water is larger in HV while smooth level ice is more easily identified in HH.

After the image selection, the samples in the selected images are split randomly into training and test datasets with a ratio of 7:3. For the training dataset, further data selection is made by excluding the samples residing close to the polygon boundaries. This is to account for possible mismatch due to various reasons (e.g., ice drift, vector mapping error, image geocoding error, etc.). In this study, only the data from pixels more than 3 km away from the polygon boundaries was fed into the training process. Once the hyperparameter optimization is done, the RF classifier is trained for the training dataset. The trained classifier

is then applied to the test dataset. For performance evaluation, we use confusion matrix and Cohen's kappa coefficient $\kappa$ (Cohen, 1960), which measures the agreement between two rasters (in this study, they are the output from the trained classifier and the reference ice chart) with taking account of the possibility of the agreement occurring by chance. The validation is done in the same way but using a completely independent dataset. The DS1 was used to run the training phase. Among 4485 images

in total, we selected 840 images (419 for winter season and 421 for summer season) of which ice edges match well with the collocated ice chart. From the selected images, 120 million samples covering open water and sea ice were divided into training and test dataset. The DS2 was used to evaluate the performance of the trained classifier using temporally independent dataset of 513 images (281 for winter season and 232 for summer season). The distribution of the image acquisition dates prior to the publication of the reference ice chart is shown in Table 2.

It might be not enough to assess the quality of the classifier output when it is trained with, and evaluated against, only NIC ice charts. The accuracy could be indirectly investigated by comparing the output from our classifier against another data source, such as OSI SAF sea ice type product (OSI-403-c). The ice classes of OSI-403-c are assigned from atmospherically corrected brightness temperatures of passive microwave radiometers (SSMIS and AMSR2) and backscatter values of radar scatterometer (ASCAT), using a Bayesian approach (Aaboe et al., 2018).

## 3 Results and discussion

We trained three RF classifiers with different feature configurations: i) FC1: Haralick texture features and CV, ii) FC2: Haralick texture features, CV, and incidence angle, iii) FC3: Haralick texture features, CV, incidence angle, and day-of-the-year.

As expected, the classification score increases with the number of trees (crosses on Figure 5, upper panel) and Richards' curve (dashed line) fits well to the observations (RMSE=$2.3 \times 10^{-4}$). The optimal $N_T$ value is selected where the score increment per tree (i.e., local slope) becomes less than 0.001 (i.e., accuracy increase of 0.1%) and constitutes 11 trees thus keeping the forest size small. The scores also increase with the maximum tree depth (crosses on Figure 5, middle panel) but Richards' curve (dashed line) doesn't fit so well (RMSE=$3.6 \times 10^{-3}$) and cannot be used for finding the optimal $D$ value. This can be explained by overfitting of the classifier and illustrated by the difference between training and testing scores (Figure 5, lower panel): small difference between the scores (for $D \leq 8$) indicate similar performance on training and testing datasets, while large difference (for $D > 8$) indicate that testing dataset is processed with worse results. The optimal $D$ value is therefore selected where the score difference become higher than 0.03 and constitutes 8 levels. The optimal value of the number of features ($N_F$) was selected using the same criterion as for $N_T$ and the value constitutes 10 features. As a result, the optimal hyperparameters of the number of trees, the maximum tree depth, and the number of features were 11, 8, and 10, respectively. The trained five-class classifier consists of five binary sub-classifiers, each of them is used for discriminating one specific class from the others. For each sub-classifier, each texture feature has a different weight in decision making. The fraction of the samples that each texture feature contributes can be used to compute the relative importance of the features, and the averaged estimates of them over several randomized trees serve as an indicator of feature importance (Louppe, 2014). The feature importance of the sub-classifiers is presented in Figure 6. The overall pattern shows that the features of HV polarization play a more important role than those of HH polarization. For HH polarization, the sum average, which is equal to the mean backscattering intensity in each subwindow, was the prominent feature. For HV polarization, however, contrast, variance- and

entropy-related features were more important. The classifiers for open water and old ice have more strong dependencies on HV polarization than others. This is understandable because the main radar scattering mechanisms for those two types are strongly characterized by the portion of volume scattering: low for calm water and high for dry ice with low salinity (old ice). The classifier for new ice has a distinctive pattern that the sum averages in both polarizations are much more important than other features. This might be because the new ice has different types of recently formed ice including nilas, which is smooth but rafting can make rough features, and frost flowers, which introduces high surface roughness and volume scattering (Isleifson et al., 2014), thus the new ice can appear either featureless dark or complex bright in SAR image (Dierking, 2010). The large range in backscatter values makes it hard to define characteristic texture in the new ice patch.

The confusion matrix for testing the trained classifier for winter season with the test dataset (DS1) is shown is in Table 3. Three cases with different feature configurations (FC1-FC3) were tested. The accuracies for open water and old ice were higher than 90%; however, those for young ice and first-year ice were around 60%. The mean difference between the results of FC1 and FC2 was only 1.2%, indicating that residual angular dependency was negligible after the incidence angle correction. However, the accuracy significantly improved from FC2 to FC3, especially with new ice (21.2%). The Cohen's kappa coefficient $\kappa$ for FC1, FC2, and FC3 were 0.70, 0.71, and 0.77, respectively. It should be noted that the evaluation of the DS1 was carried out with the input dataset that was used for training. Thus, the test and training data share the same ice conditions as well as spatio-temporal coverage. As a result, the $\kappa$ might contain correlation which is not preferable for proper evaluation. Table 4 shows the confusion matrix for validation results from the DS2 of which the accuracy of open water and old ice was at a similar level, compared to the DS1. Meanwhile, for the accuracy of young ice, and first-year ice decreased considerably. The differences between the results of FC1, FC2, and FC3 were insignificant. This result is opposite to the DS1 inferring that the training with FC3 was overfitted and the day-of-the-year may not correspond to the temperature, air-sea fluxes, or weather regimes. The $\kappa$ for FC1, FC2, and FC3 with the DS2 were 0.67, 0.67, and 0.67, respectively.

To see how the denoising step in Section 2.2.2 led to improvements in the classification accuracies, the same training and evaluation were conducted for the same dataset without applying the textural noise correction (Table 5). In all configurations (FC1-FC3), the accuracies improved for young ice (+2.4% to +6.2%) and old ice (+3.9% to +4.5%) which were most pronounced compared to those for open water (+1.1% to +1.9%) and first-year ice (-0.2% to +0.9%). On the contrary, a small accuracy decrease was observed for new ice (=2.4% to -0.3%). Nevertheless, the improvement in $\kappa$ (+0.08) demonstrates a clear improvement in the overall classification result.

The confusion matrix for testing the trained classifier for summer season with the test dataset (DS1) is shown is in Table 6. As described in Section 2.2.1, the further simplified three-class classification is applied. The accuracies for open water and old ice were higher than 92%; however, the accuracies for mixed first-year ice were only around 15% both in FC1 and FC2, and 42% in FC3. The large difference between the results of FC1-FC2 and FC3 indicates that the mixed first-year ice likely changes at short time scales. The misclassifications for mixed first-year ice were mostly into old ice. This might be because of the surface melting and the corresponding image textures which makes the discrimination between the mixed first-year ice and old ice difficult. The same patterns were observed from the confusion matrix (Table 7) for validation results from the DS2,

except that the accuracy decrease from Table 6 to Table 7 was particularly large for FC3, meaning overfitting for DS1. However, it is unclear if the low accuracy for mixed first-year ice is due to the classifier itself or the data. To unravel this, the data were divided into three of one month each (June, July, August) and then separate classifiers were trained and tested. Table 8 and Table 9 show the results with FC1 configuration for DS1 and DS2, respectively. There was a rapid accuracy decrease

for mixed first-year ice from June to August in both results (from 60.6% to 26.5% for DS1, and from 55.6% to 11.0% for DS2). As there is no particularly large difference between the numbers in Table 8 and Table 9, meaning the classifiers were not overfitting, the very low accuracies for mixed first-year ice in Table 6 (14.9%) and Table 7 (12.0%) seem to be due to a too large temporal extent for a single classifier. In other words, the classifier for summer season needs to be split into multiple with shorter time span, and/or a feature that can effectively account for surface melting needs to be introduced into the

algorithm for further development. Based on these results so far, the trained classifiers in summer season for three months in bulk failed in distinguishing mixed first-year ice from old ice, thus they are close to ice-water discriminators rather than ice type classifiers.

Figure 7 shows a daily mosaic of Sentinel-1 SAR images over the study area and the classified ice map in the winter season. For comparison, the NIC weekly ice chart is also displayed. Despite the SAR images had been acquired three days before the

15 ice chart was published, the ice edges of the ice chart match well with the SAR mosaic in most parts because the same SAR data were used. In overall, the discriminations between ice and non-ice, old ice and other ice types, and detection of new ice patches look reasonable. However, some young ice patches, for example the ice patches between the Svalbard archipelago, are misclassified as the first-year ice. Figure 8 shows another daily mosaic made by the images acquired on the same day of the ice chart publication. Considering notable ice drift in the backscattering images in Figure 7 and 8, the SAR-based ice

classification results in both figures look consistent, well in line with the ice drift. Although the weekly ice chart is supposed to represent the averaged ice status in the past few days, the actual ice distribution on the actual date of the publication can be largely different. This example shows a clear potential of near-real time service of ice type classification.

Figure 9 and 10 show the same mosaics for the case in the summer season. As shown in Table 6 and 7, the misclassifications for the mixed first-year ice into old ice are pronounced in the large ice patches north to Svalbard, while the ice edge positions

of the ice chart and the classification result are in well agreement with each other.

To cope with the ambiguous classification for the winter season ice types with low accuracy, we conducted a test with the three-class classification, and Table 10 and 11 show the resulting confusion matrices. The $\kappa$ for FC1, FC2, and FC3 were 0.83, 0.84, and 0.84 in DS1, and 0.75, 0.75, and 0.74 in DS2, respectively. The dramatic increase in the accuracy of the mixed first-year ice indicates that the misclassification for the new ice, young ice, and first-year ice was mostly among themselves.

However, the accuracy decrease from DS1 to DS2 was at a similar level to the case of the five-class classification. This could have been caused by inconsistent labeling in the reference ice chart.

Figure 11 shows an example of the inconsistent labeling in the reference ice chart. The SoDs from the NIC ice charts are superimposed on the Sentinel-1 backscattering images. The same ice floe (red outline) is classified differently in two different ice charts (old ice on the left panel and first-year ice on the right panel) although it looks almost the same in the corresponding

SAR backscattering images. It should be noted that training with ice chart might have included mislabeled small features even if the image selection based on ice edge matching was successful. Furthermore, the boundaries between different ice types in the ice chart are normally not as precise as those in the SAR image-based classification results. Therefore, the lower classification accuracies compared to those in the previous studies (80% in Zakhvatkina et al., 2013; 91.7% in Liu et al., 2015; 87.2% in Aldenhoff et al., 2018), which used manually classified ice maps as training and validation reference, are expected. Unfortunately, we could not find an official report regarding the accuracy of the NIC ice chart information.

Table 12 shows the confusion matrices for our three-class classifiers when their prediction results are compared with the OSI-403-c product as reference. For one-to-one comparison, it was assumed that the ideal characteristics of the mixed first-year ice and the old ice in our three-class classification are equivalent to those of the first-year ice and the multi-year ice in OSI-403-c. Comparing with the results in Table 11, the accuracies for open water decreased from by 6%; however, this is mainly because the ice concentration threshold for ice-water discrimination in OSI-403-c is 35% which is higher than 20% that we set in our preprocessing of NIC ice chart (Section 2.2.1), thus areas with low ice concentration in marginal ice zone are most likely annotated as open water in OSI-403-c. The accuracies for open water at points in the NIC charts with ice concentration between 20% and 40% only were considerably lower, with 67.4%, 67.8%, 70.1% for FC1, FC2, and FC3, respectively (not presented in Table 12). For first-year ice, large portions (72%) are misclassified as old ice. This might be partly explained from the Figure 12, which shows the ice classes in NIC ice chart and OSI-403-c for the same publication date. A large extent of old ice in NIC ice chart is annotated as multi-year ice in OSI-403-c. As our classifiers were trained with NIC ice chart, it is natural to result in more multi-year ice for the area where the ice type is classified as first-year ice in OSI-403-c. For multi-year ice, the accuracy was the highest, 98%.

The inconsistency in ice types between NIC ice chart and OSI-403-c seems persistent at least for the time coverage of DS2 (January-March in 2019). Table 13 shows averaged percent agreement of the two sea ice type products for the same publication dates over 12 weeks (12 one-to-one comparisons as the NIC ice chart is a weekly product). To make a fair comparison, the ice-covered areas with ice concentrations lower than 35%, which is the threshold for ice-water discrimination in OSI-403-c, were excluded. The percent agreement for first-year ice (58.8%) was much lower than those of open water (90.0%) and multi-year ice (99.0%), which is inline with the results in Table 12. Finding the reason for the clear discrepancy of the extent of first-year ice between the NIC ice chart and OSI-403-c is beyond the scope of this study, however, it should be noted that an elaborate future work for cross calibrating ice types in different ice charts are necessary.

The proposed algorithm has several limitations. First of all, the variations in radar backscattering and its corresponding image textures due to seasonal changes were not properly captured. Although day-of-the-year was tested as a seasonality variable in the FC3 feature configuration, the result did not show any improvement. This is because SAR image features, which partially reflect temperature fluxes and weather regimes, might not correspond to day-of-the-year. Second, the proposed method struggles when the same type of sea ice is located on different edges of the range swath of SAR images because the incidence angle dependence could not be normalized perfectly. An example of such a failure can be seen along the image boundaries at 79.5N, 45E in Figure 7 and 79N, 50E in Figure 8, approximately. Third, some artifacts were observed under large ocean swells.

In the classified results in the bottom right panel of Figure 8, there is a misclassified first-year ice patch (yellow) in the open water area. According to the high resolution sea surface winds data from SAR on the Sentinel-1 satellites (https://data.nodc.noaa.gov/cgi-bin/iso?id=gov.noaa.nodc:SAR-WINDS-S1), the wind speed ranged from 17 to 21 m/s at the time of image acquisition heavily roughing the water surface. Although we have included images with both high and low wind conditions in our training data, the image textures of wind roughened water surface and ice were confused in some cases, and the same happened in the image textures of calm water surface and smooth level ice.

## 4 Conclusion

A new semi-automated SAR-based sea ice type classification scheme was proposed in this study. For the first time several ice types were successfully identified on Sentinel-1 SAR imagery in winter season, while only an ice-water discrimination was feasible in summer season. The main technological innovation is two-fold: i) reduced manual work in the preparation of large amount of training and validation reference data using readily available public ice charts and ii) more objective evaluation of the SAR-based sea ice type classifier compared to the previous studies conducted with small number of images and customized ice type references from sources not open to the public. A conventional approach for selecting training/testing data by anonymous human ice expert is undesirable not only because it is laborious, but also due to subjectivity and lack of standardization in the assessment of the automated classifier. Therefore, the performance from different literature sources cannot be intercompared directly.

Test results from the datasets of winter season acquired over the Fram Strait and the Barents Sea area showed overall accuracies of 87% and 60% and the Cohen's kappa coefficients of 0.75 and 0.67 for the three-class and five-class ice type classifiers, respectively. These are slightly lower than the numbers in the previous studies, and the errors are attributed not only to the automated algorithm but also to the inconsistency of the ice charts and the high level of their generalization. Test results from the datasets of summer seasons showed overall accuracy of 67% and the Cohen's kappa coefficient of 0.78 for the three-class classifiers. Considering the misclassifications in different ice types were among themselves, the three-class classifiers are not really a sea ice type classifier but they performed well at least as an ice-water discriminator with accuracy of 98%.

Based on the results, we envisage that three-class ice type classification from SAR imagery would be useful for making a global sea ice type product like OSI SAF OSI-403-c with higher spatial resolution. The proposed approach importantly showed that a daily ice type mapping from the Sentinel-1 data is feasible and can help capture details of short-term changes in the stage of sea ice development. Based on the achieved results, we believe that the proposed approach may be efficiently used for operational ice charting services for supporting navigation in the Arctic.

**Code/Data availability**

Not applicable

**Author contributions**

5  JP and AK formulated the research plan, JP and AK developed the algorithm, JP implemented the algorithm and performed the data processing, JP, AK, MB, JW, MH, and HK carried out the analyses, and JP wrote the paper.

**Competing interests**

The authors declare that they have no conflict of interests.

**Financial support**

This work was supported by the French Service Hydrographique et Océanographique de la Marine (SHOM) under SHOM-ImpSIM Project 111222, the Research Council of Norway and the Russian Foundation for Basic Research under NORRUSS Project 243608, SONARC, and the Korea Polar Research Institute under grant number PE20080.

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

**Figures**

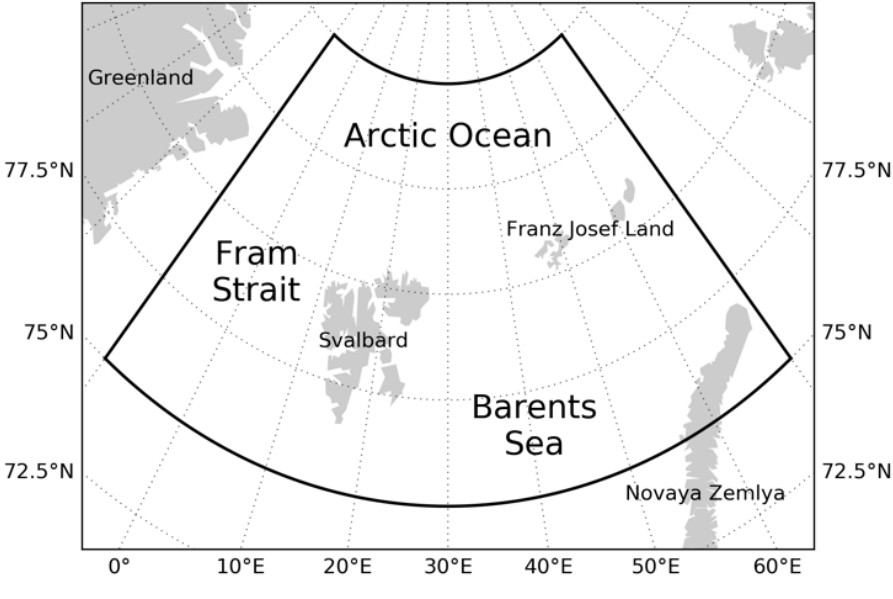

Figure 1: Study area.

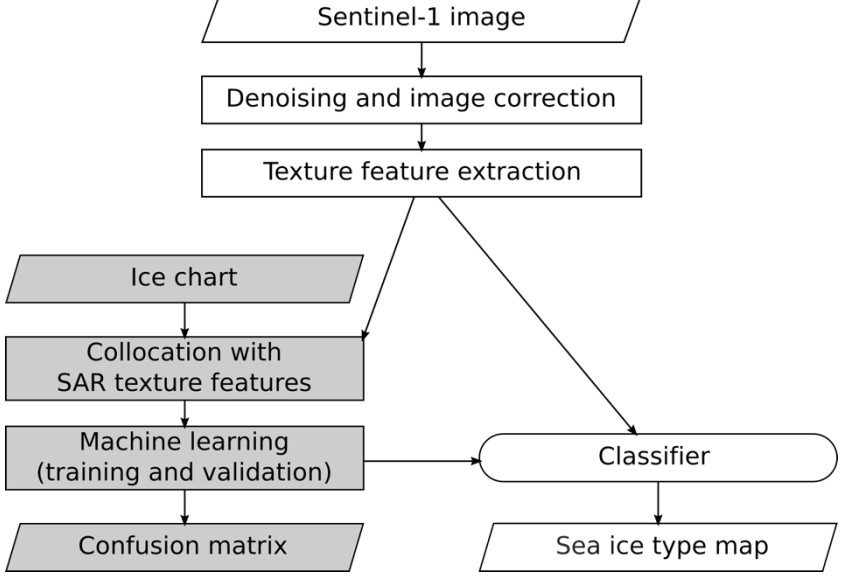

Figure 2: Processing flowchart of the proposed algorithm. The gray color shows the training phase.

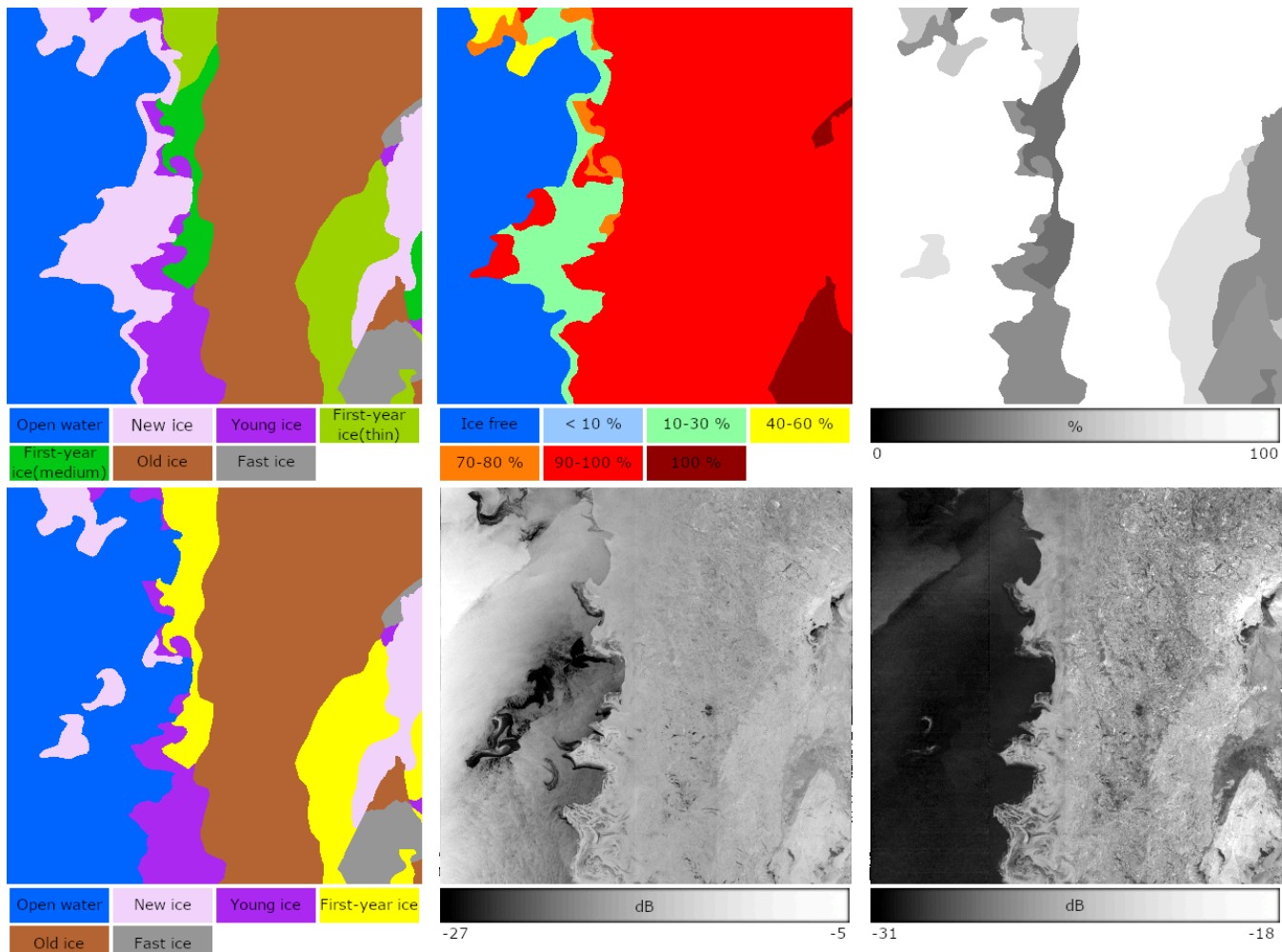

**Figure 3: An example of the ice chart preprocessing. From the ice chart, stage of development (SoD; Top left), ice concentration (CT; Top center), and partial concentration of the dominant ice type (CP; Top right) maps are extracted. Then, some of the different SoDs are merged (e.g., thin and thick first-year ice are merged into a single label as first-year ice) and the area with low ice concentration is labeled as open water. The processed map of SoD (Bottom left) is related with textural features extracted from HH and HV polarization images (Bottom center and bottom right). Note that the NIC ice chart which was published on January 25, 2018, and the Sentinel-1 product S1B_EW_GRDM_1SDH_20180122T075237_20180122T075337_009281_010A4D_65AA acquired over the Fram Strait were used in this example.**

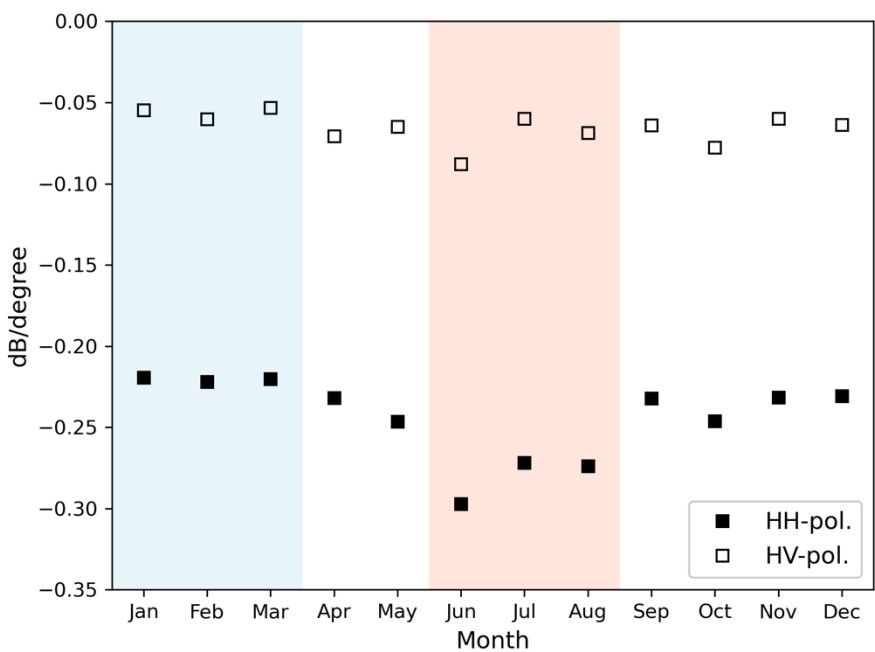

**Figure 4: Incidence angle dependences of sigma naught in HH (closed squares) and HV (open squares) polarization channels. Pixels covering various types of sea ice were merged so that the averaged property can be estimated. The blue and red zones indicate winter and summer seasons, respectively.**

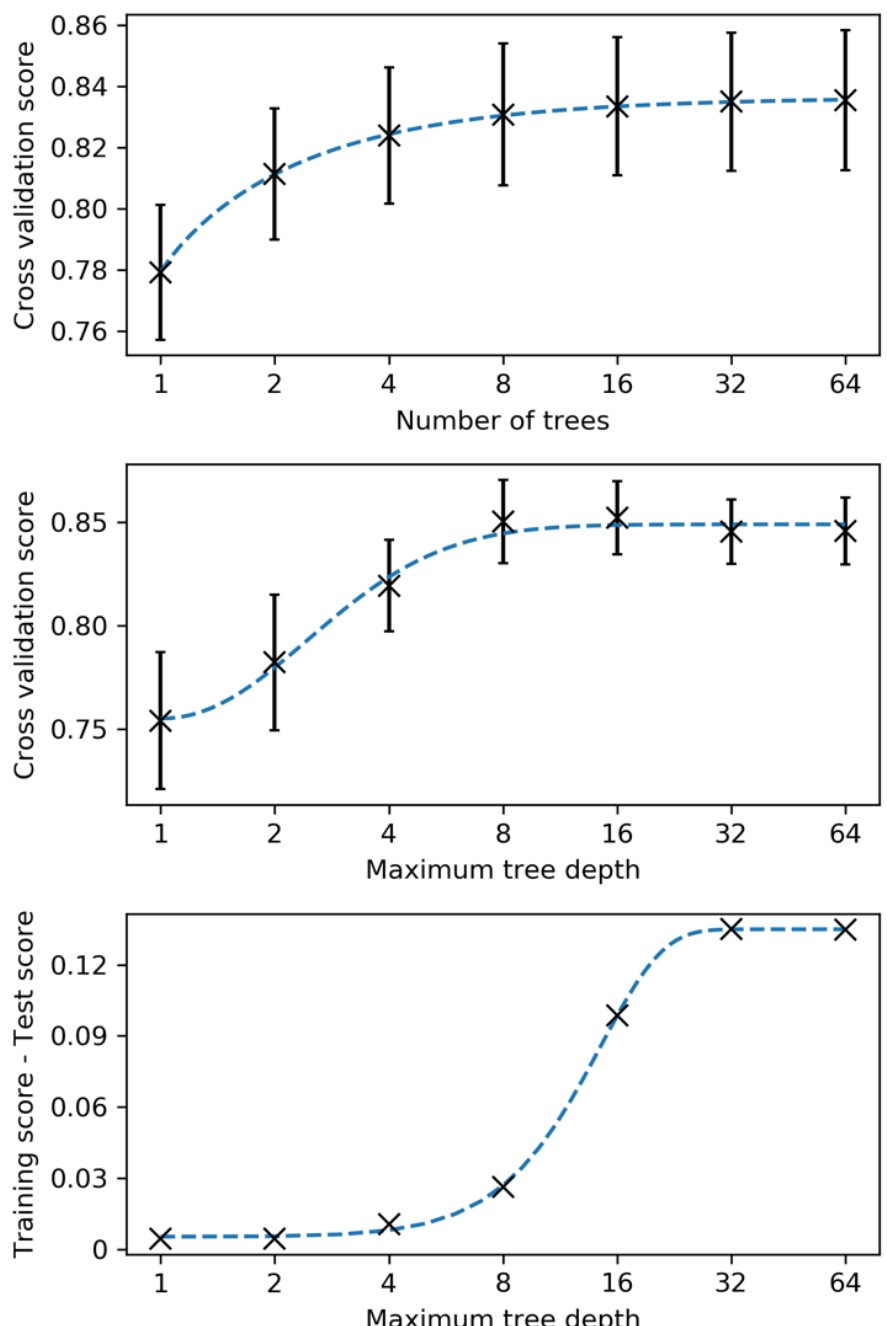

**Figure 5: Hyperparameter optimization using grid search results (cross). Dashed lines represent the best-fit Richards' Curve. (Top panel) The optimal values are extracted from the locations where the score increments per unit of each hyperparameter become lower than a threshold (e.g., 0.001). (Center panel) If the curve does not fit the grid search results well, (Bottom panel) the difference between training and test scores is used to find the locations where it does not exceed a threshold (e.g., 0.03) in order to avoid overfitting.**

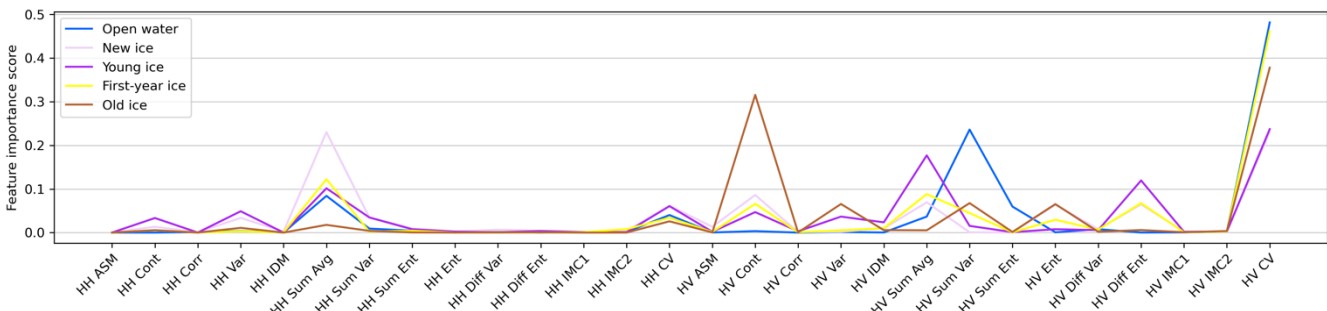

**Figure 6: Feature importances of the binary sub-classifiers. ASM: angular second moment; Cont: contrast; Corr: correlation; Var: variance; IDM: inverse difference moment; Sum Avg: sum average; Sum Var: sum variance; Sum Ent: sum entropy; Ent: entropy; Diff Var: difference variance; Diff Ent: difference entropy; IMC: information measures of correlation; CV: coefficient of variation. For definitions of each parameters, please refer to Haralick et al., 1973.**

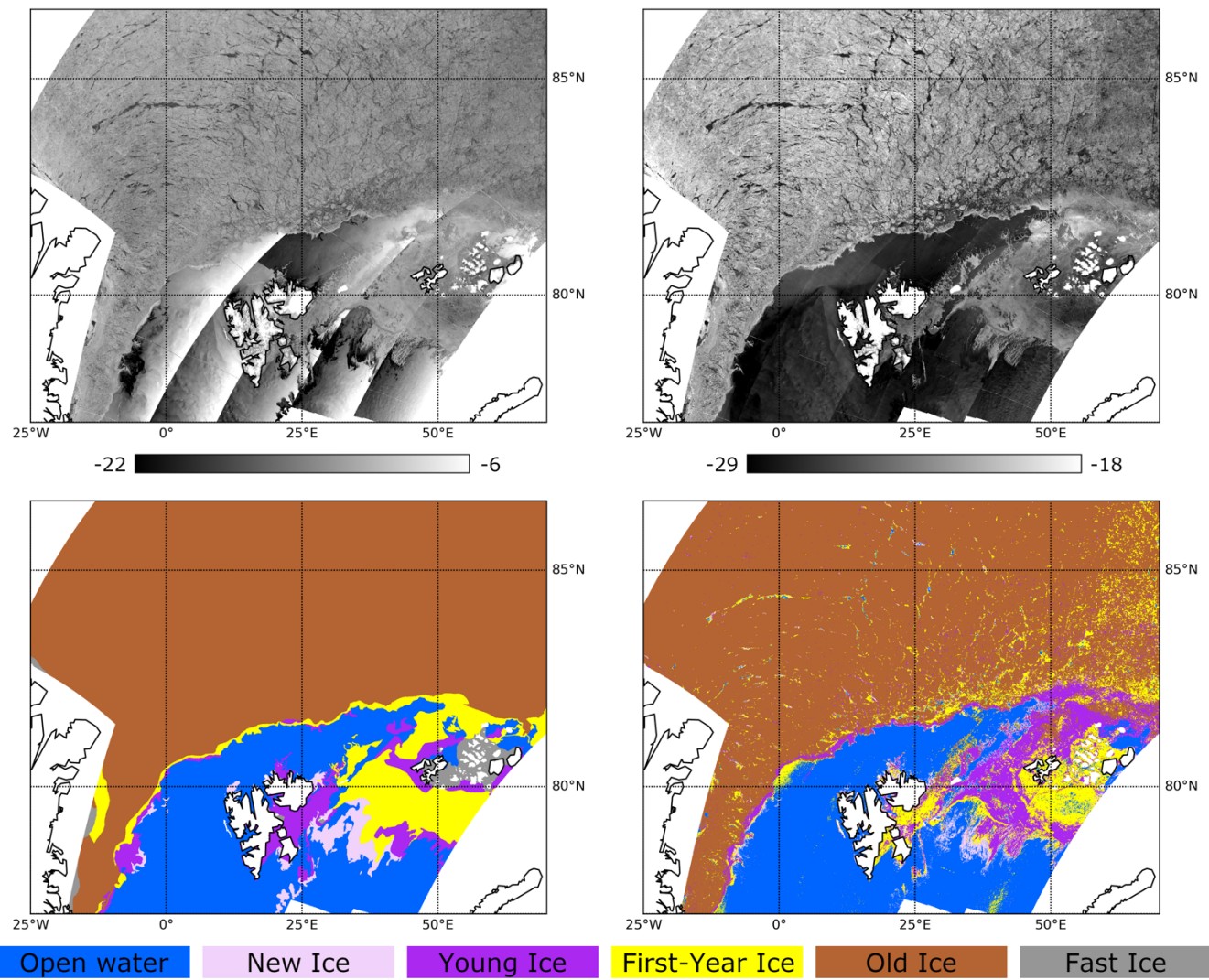

**Figure 7: One-day mosaics of Sentinel-1A/1B images (Top left: HH, Top right: HV) and the ice classification result (Bottom right) on 5 February 2019. The publication date of the reference weekly ice chart is 8 February 2019 (Bottom left).**

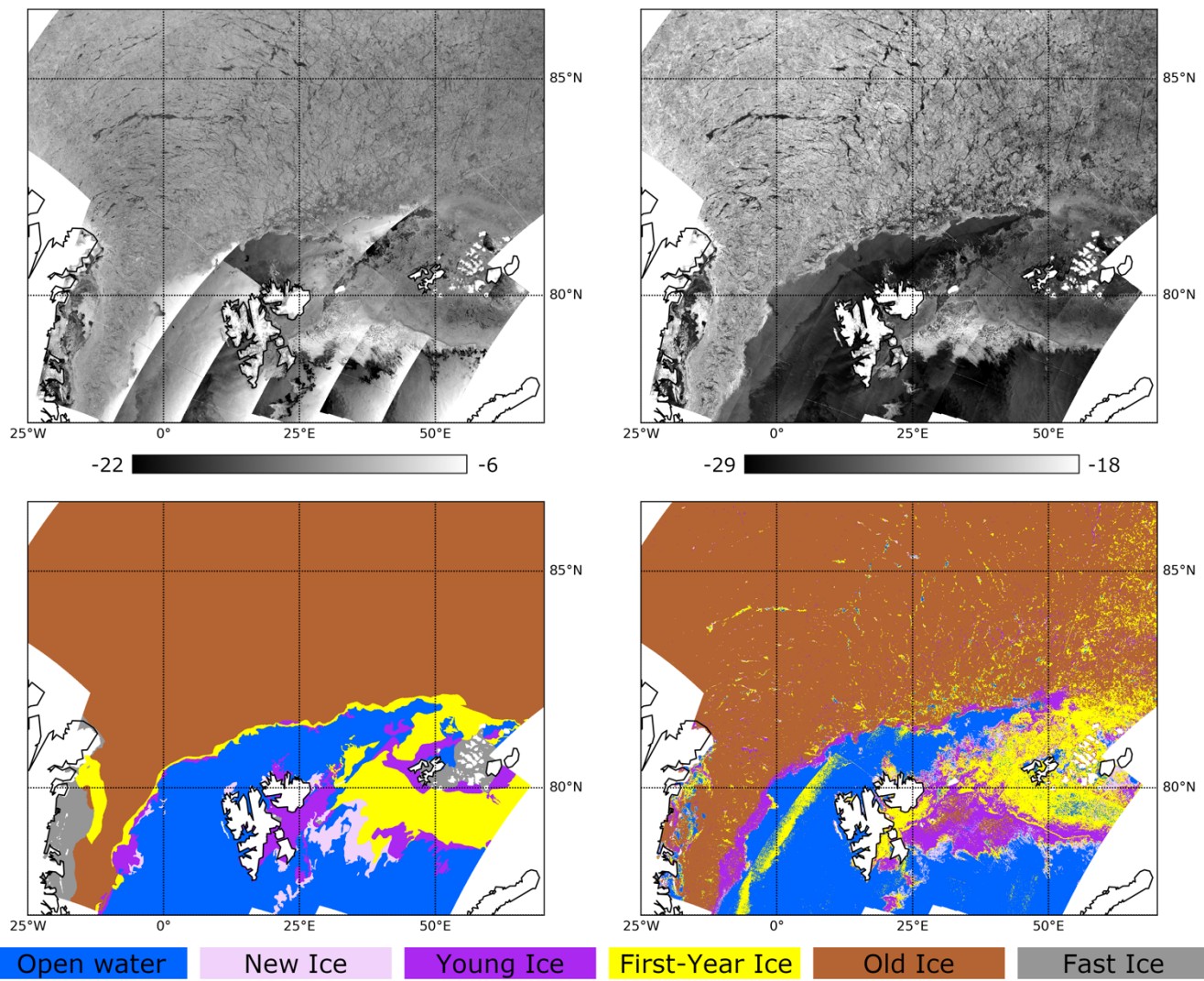

**Figure 8: One-day mosaics of Sentinel-1A/1B images (Top left: HH, Top right: HV) and the ice classification result (Bottom right) on 8 February 2019. The publication date of the reference weekly ice chart is 8 February 2019 (Bottom left).**

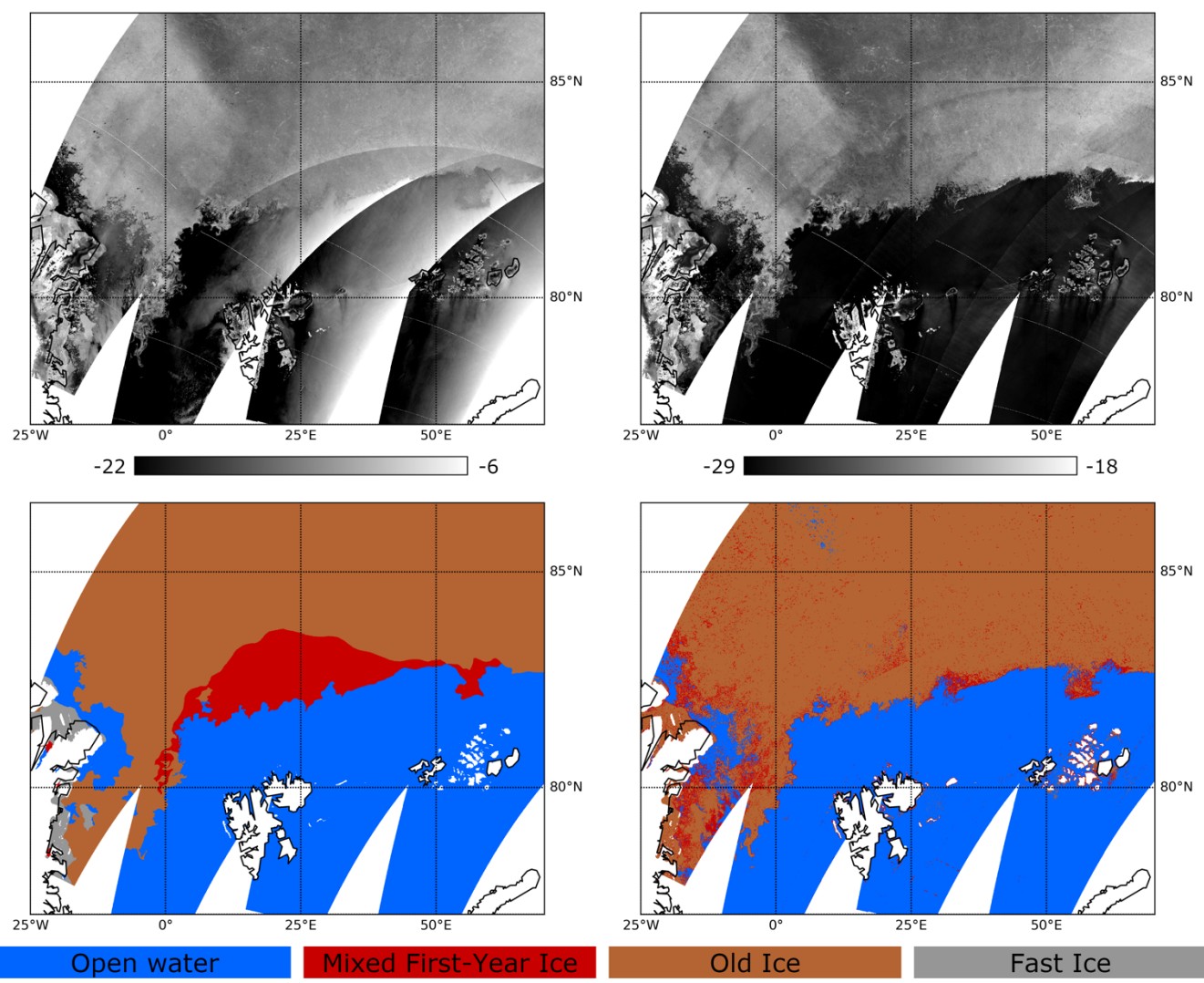

**Figure 9: One-day mosaics of Sentinel-1A/1B images (Top left: HH, Top right: HV) and the ice classification result (Bottom right) on 13 August 2018. The publication date of the reference weekly ice chart is 16 August 2018 (Bottom left).**

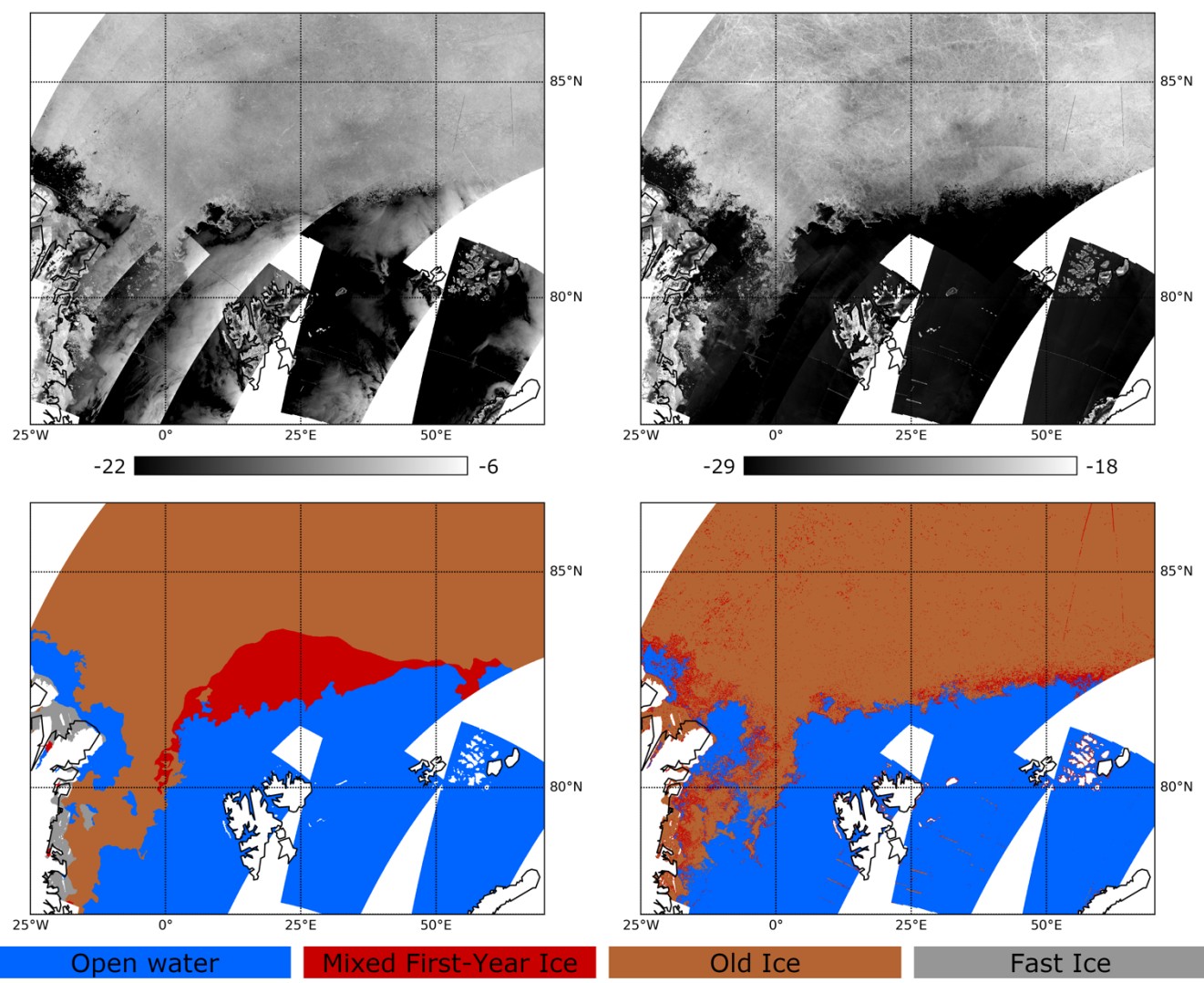

**Figure 10: One-day mosaics of Sentinel-1A/1B images (Top left: HH, Top right: HV) and the ice classification result (Bottom right) on 16 August 2018. The publication date of the reference weekly ice chart is 16 August 2018 (Bottom left).**

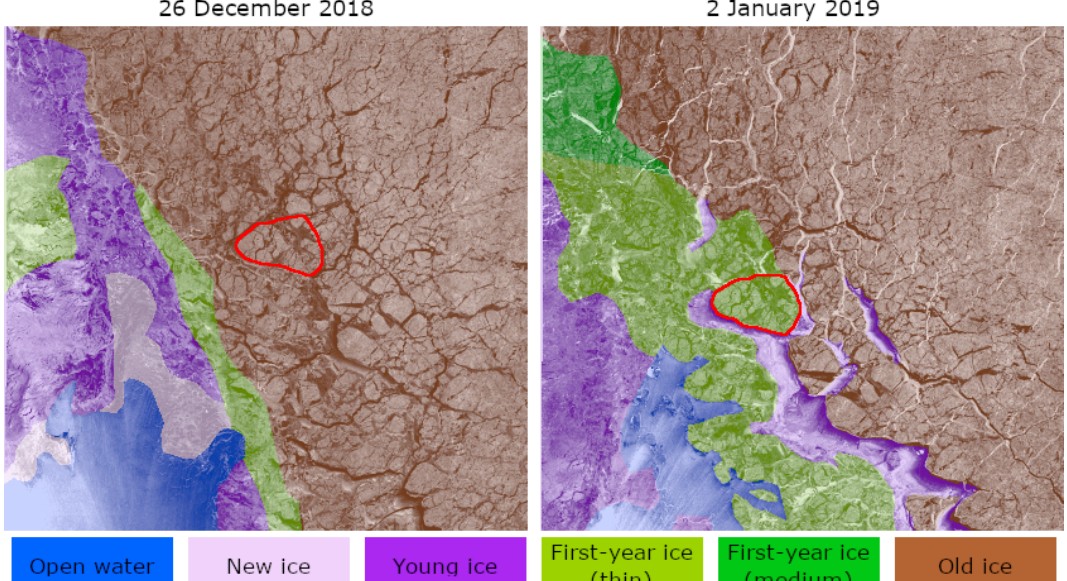

**Figure 11: An example of the inconsistency of the ice charts. Note that the SoD labels and colors are of original NIC ice chart, while those in Figure 7 and 8 are of simplified version as described in Section 2.2.1.The SoDs from the NIC ice charts on different dates (26 December 2018 and 2 January 2019) are superimposed on the Sentinel-1 backscattering image of the corresponding dates. The same ice floe (red outline) is classified differently in each ice chart (old ice on the left panel and first-year ice on the right panel) despite of the similarity in the SAR backscattering images. Source credits: U.S. National Ice Center (colors) and European Space Agency (background).**

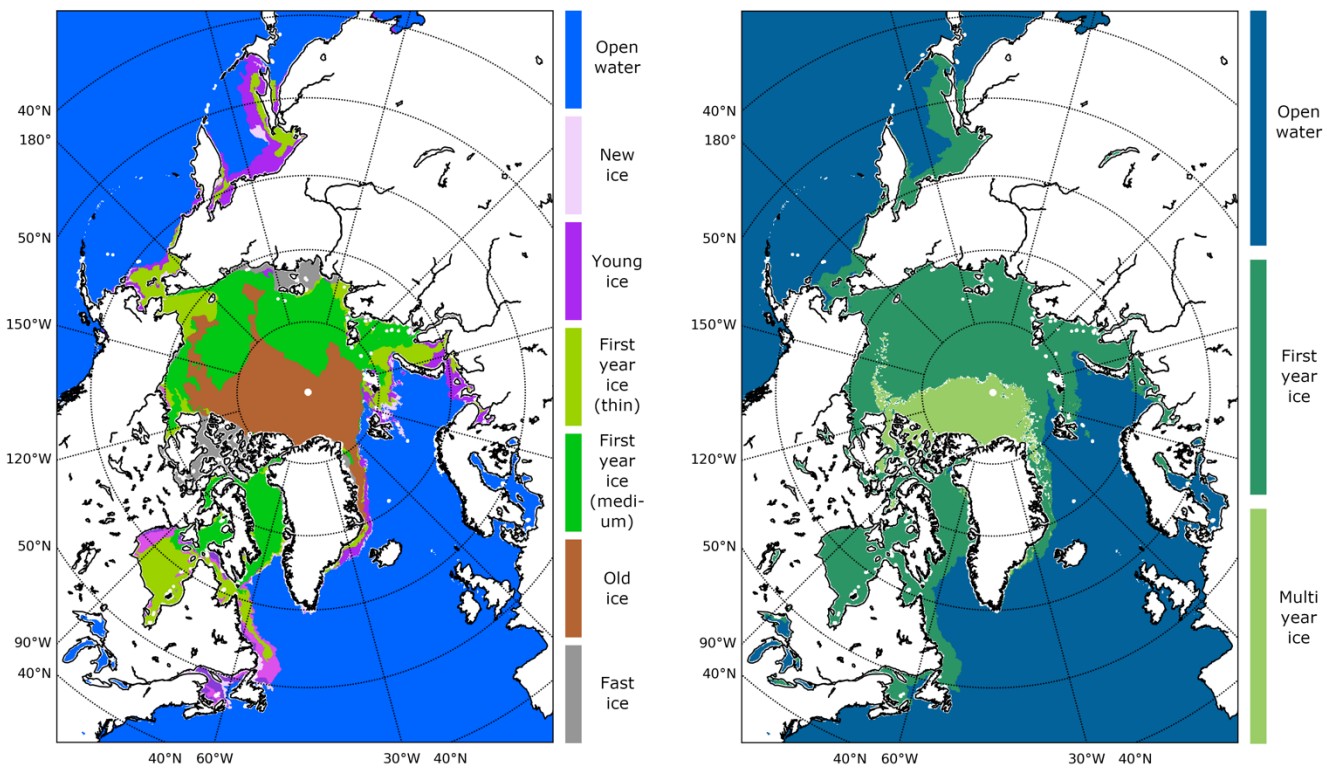

**Figure 12: The ice types in NIC ice chart (left) and OSI SAF sea ice type product (right) for the same date (8 Feb. 2019). Note that the SoD labels and colors follows those defined in each ice chart format. Source credits: U.S. National Ice Center (left) and EUMETSAT Ocean and Sea Ice Satellite Application Facilities (right).**

**Tables**

**Table 1: Hyperparameters used for grid search**

| Parameters | Values | | | | | | |
|---|---|---|---|---|---|---|---|
| $N_T$ | 1 | 2 | 4 | 8 | 16 | 32 | 64 |
| $D$ | 1 | 2 | 4 | 8 | 16 | 32 | 64 |
| $N_F$ | 1 | 2 | 4 | 8 | 16 | 28 | |

**Table 2: Distribution of the image acquisition dates prior to the publication of the reference ice chart**

| | Training and test dataset (DS1) | | | | Validation dataset (DS2) | | | |
|---|---|---|---|---|---|---|---|---|
| Days prior to the date of ice chart publication | 3 | 2 | 1 | 0 | 3 | 2 | 1 | 0 |
| Winter | 124 | 168 | 77 | 50 | 78 | 75 | 67 | 61 |
| Summer | 119 | 125 | 112 | 65 | 87 | 67 | 48 | 30 |

**Table 3: Confusion matrix of the five-class RF classifier which was trained with and applied to the DS1 winter dataset**

| | | Predicted | | | | | | | | | | | | | | |
|---|---|---|---|---|---|---|---|---|---|---|---|---|---|---|---|---|
| | | OW (open water) | | | NI (new ice) | | | YI (young ice) | | | FYI (first-year ice) | | | OI (old ice) | | |
| | case | FC1 | FC2 | FC3 | FC1 | FC2 | FC3 | FC1 | FC2 | FC3 | FC1 | FC2 | FC3 | FC1 | FC2 | FC3 |
| Actual | OW | **90.1** | **91.3** | **92.4** | 1.3 | 1.1 | 1.2 | 1.6 | 1.7 | 1.6 | 6.9 | 5.9 | 4.8 | 0.0 | 0.0 | 0.0 |
| | NI | 30.0 | 28.0 | 26.1 | **21.9** | **23.8** | **45.0** | 27.1 | 26.7 | 13.3 | 16.7 | 17.7 | 11.8 | 4.2 | 3.9 | 3.8 |
| | YI | 3.7 | 3.7 | 3.9 | 5.1 | 4.8 | 5.2 | **58.8** | **60.1** | **62.6** | 24.3 | 24.0 | 21.1 | 8.1 | 7.4 | 7.2 |
| | FYI | 5.0 | 4.7 | 3.9 | 1.7 | 1.5 | 1.7 | 18.7 | 19.0 | 19.4 | **64.4** | **65.3** | **65.9** | 10.1 | 9.6 | 9.1 |
| | OI | 0.1 | 0.1 | 0.2 | 0.3 | 0.3 | 0.6 | 3.1 | 2.9 | 3.0 | 6.4 | 6.1 | 5.6 | **90.1** | **90.6** | **90.6** |

**Table 4: Confusion matrix of the five-class RF classifier which was trained with DS1 winter dataset and applied to the DS2 winter dataset**

| | | Predicted | | | | | | | | | | | | | | |
|---|---|---|---|---|---|---|---|---|---|---|---|---|---|---|---|---|
| | | OW (open water) | | | NI (new ice) | | | YI (young ice) | | | FYI (first-year ice) | | | OI (old ice) | | |
| | case | FC1 | FC2 | FC3 | FC1 | FC2 | FC3 | FC1 | FC2 | FC3 | FC1 | FC2 | FC3 | FC1 | FC2 | FC3 |
| Actual | OW | **89.1** | **90.2** | **90.3** | 1.3 | 1.1 | 1.9 | 3.3 | 3.5 | 4.2 | 6.4 | 5.2 | 3.7 | 0.0 | 0.0 | 0.0 |
| | NI | 45.1 | 45.0 | 56.5 | **31.9** | **30.6** | **17.6** | 6.0 | 5.7 | 13.3 | 15.1 | 17.3 | 11.2 | 2.0 | 1.5 | 1.5 |
| | YI | 7.1 | 7.1 | 9.2 | 6.3 | 5.9 | 8.5 | **47.6** | **48.0** | **55.0** | 28.7 | 29.2 | 17.3 | 10.4 | 9.8 | 9.9 |
| | FYI | 5.6 | 5.0 | 5.8 | 3.8 | 3.5 | 3.2 | 32.8 | 33.0 | 35.0 | **38.4** | **39.7** | **37.3** | 19.3 | 18.8 | 18.7 |
| | OI | 0.3 | 0.3 | 0.7 | 0.5 | 0.4 | 0.7 | 1.9 | 1.8 | 1.9 | 4.6 | 4.8 | 4.5 | **92.8** | **92.8** | **92.6** |

**Table 5: Classification accuracies before and after applying textural denoising**

| class | case | | | | | | | | |
|---|---|---|---|---|---|---|---|---|---|
| | FC1 | | | FC2 | | | FC3 | | |
| | Thermal denoising only | Textural denoising applied | difference | Thermal denoising only | Textural denoising applied | difference | Thermal denoising only | Textural denoising applied | difference |
| Open water | 88.0 | 89.1 | +1.1 | 88.7 | 90.2 | +1.5 | 89.4 | 90.3 | +1.9 |
| New ice | 32.2 | 31.9 | -0.3 | 32.2 | 30.6 | -1.6 | 20.0 | 17.6 | -2.4 |
| Young ice | 45.2 | 47.6 | +2.4 | 44.7 | 48.0 | +3.3 | 48.8 | 55.0 | +6.2 |
| First-year ice | 38.6 | 38.4 | -0.2 | 39.4 | 39.7 | +0.3 | 36.4 | 37.3 | +0.9 |
| Old ice | 88.9 | 92.8 | +3.9 | 88.3 | 92.8 | +4.5 | 88.6 | 92.6 | +4.0 |
| kappa | 0.59 | 0.67 | +0.08 | 0.59 | 0.67 | +0.08 | 0.59 | 0.67 | +0.08 |

5 **Table 6: Confusion matrix of the three-class RF classifier which was trained with and applied to the DS1 summer dataset**

| | | Predicted | | | | | | | | |
|---|---|---|---|---|---|---|---|---|---|---|
| | | OW (open water) | | | mFYI (mixed first-year ice) | | | OI (old ice) | | |
| | Case | FC1 | FC2 | FC3 | FC1 | FC2 | FC3 | FC1 | FC2 | FC3 |
| Actual | OW | **98.1** | **97.9** | **98.7** | 0.7 | 0.7 | 0.6 | 1.2 | 1.4 | 0.6 |
| | mFYI | 4.1 | 3.9 | 2.4 | **14.9** | **15.5** | **41.8** | 81.1 | 80.6 | 55.8 |
| | OI | 1.5 | 1.4 | 1.0 | 5.5 | 5.7 | 4.9 | **93.0** | **92.9** | **94.1** |

**Table 7: Confusion matrix of the three-class RF classifier which was trained with DS1 summer dataset and applied to the DS2**
10 **summer dataset**

| | | Predicted | | | | | | | | |
|---|---|---|---|---|---|---|---|---|---|---|
| | | OW (open water) | | | mFYI (mixed first-year ice) | | | OI (old ice) | | |
| | Case | FC1 | FC2 | FC3 | FC1 | FC2 | FC3 | FC1 | FC2 | FC3 |
| Actual | OW | **99.5** | **99.4** | **96.2** | 0.2 | 0.2 | 3.2 | 0.3 | 0.4 | 0.6 |
| | mFYI | 5.4 | 5.1 | 3.0 | **12.0** | **11.2** | **25.8** | 82.5 | 83.7 | 71.2 |
| | OI | 2.9 | 2.7 | 2.2 | 5.8 | 5.8 | 13.4 | **91.2** | **91.4** | **84.4** |

**Table 8: Confusion matrix of the three-class RF classifier which was trained with DS1 summer dataset and applied to the DS1 summer dataset of each month**

| | | Predicted (FC1) | | | | | | | | |
|---|---|---|---|---|---|---|---|---|---|---|
| | | OW (open water) | | | mFYI (mixed first-year ice) | | | OI (old ice) | | |
| | Case | June | July | August | June | July | August | June | July | August |
| Actual | OW | **99.0** | **99.0** | **98.1** | 0.9 | 0.3 | 0.3 | 0.1 | 0.7 | 1.6 |
| | mFYI | 4.7 | 2.1 | 3.9 | **60.6** | **32.6** | **26.5** | 34.7 | 65.3 | 69.6 |
| | OI | 0.6 | 0.8 | 2.1 | 7.7 | 7.7 | 10.1 | **91.7** | **91.5** | **87.8** |

**Table 9: Confusion matrix of the three-class RF classifier which was trained with DS1 summer dataset and applied to the DS2 summer dataset of each month**

| | | Predicted (FC1) | | | | | | | | |
|---|---|---|---|---|---|---|---|---|---|---|
| | | OW (open water) | | | mFYI (mixed first-year ice) | | | OI (old ice) | | |
| | Case | June | July | August | June | July | August | June | July | August |
| Actual | OW | **90.6** | **99.5** | **99.7** | 8.1 | 0.2 | 0.1 | 1.3 | 0.3 | 0.2 |
| | mFYI | 3.4 | 4.0 | 4.2 | **55.6** | **41.4** | **11.0** | 41.0 | 54.9 | 84.9 |
| | OI | 1.4 | 3.4 | 3.4 | 11.2 | 10.9 | 6.3 | **87.5** | **85.6** | **90.3** |

**Table 10: Confusion matrix of the three-class RF classifier which were trained and applied to the DS1 winter dataset**

| | | Predicted | | | | | | | | |
|---|---|---|---|---|---|---|---|---|---|---|
| | | OW (open water) | | | mFYI (mixed first-year ice) | | | OI (old ice) | | |
| | Case | FC1 | FC2 | FC3 | FC1 | FC2 | FC3 | FC1 | FC2 | FC3 |
| Actual | OW | **92.2** | **92.5** | **93.6** | 7.8 | 7.4 | 6.3 | 0.0 | 0.0 | 0.0 |
| | mFYI | 6.4 | 5.5 | 5.5 | **83.8** | **85.3** | **85.5** | 9.8 | 9.2 | 9.0 |
| | OI | 0.2 | 0.2 | 0.2 | 8.9 | 8.5 | 8.7 | **90.9** | **91.3** | **91.1** |

**Table 11: Confusion matrix of the three-class RF classifier which was trained with the DS1 winter dataset and applied to the DS2 winter dataset**

| | | Predicted | | | | | | | | |
|---|---|---|---|---|---|---|---|---|---|---|
| | | OW (open water) | | | mFYI (mixed first-year ice) | | | OI (old ice) | | |
| | Case | FC1 | FC2 | FC3 | FC1 | FC2 | FC3 | FC1 | FC2 | FC3 |
| Actual | OW | **91.4** | **91.7** | **91.7** | 8.6 | 8.3 | 8.3 | 0.0 | 0.0 | 0.0 |
| | mFYI | 9.4 | 8.3 | 9.9 | **75.0** | **76.5** | **74.6** | 15.6 | 15.2 | 15.5 |
| | OI | 0.3 | 0.3 | 0.3 | 6.3 | 6.5 | 6.6 | **93.3** | **93.2** | **93.1** |

**Table 12: Confusion matrix of the three-class RF classifier which was trained with DS1 winter dataset and applied to the DS2 winter dataset with reference to OSI SAF sea ice type product (OSI-403-c)**

| | | Predicted (classifier was trained with NIC ice chart) | | | | | | | | |
|---|---|---|---|---|---|---|---|---|---|---|
| | | Open water | | | First-year ice | | | Multi-year ice | | |
| | Case | FC1 | FC2 | FC3 | FC1 | FC2 | FC3 | FC1 | FC2 | FC3 |
| Reference (OSI SAF) | Open water | **85.9** | **86.1** | **86.2** | 12.6 | 12.4 | 12.1 | 15.7 | 15.3 | 16.2 |
| | First-year ice | 1.9 | 1.6 | 2.0 | **26.0** | **26.8** | **26.9** | 72.1 | 71.6 | 71.2 |
| | Multi-year ice | 0.1 | 0.1 | 0.1 | 1.5 | 1.4 | 1.4 | **98.4** | **98.5** | **98.5** |

**Table 13: Averaged percent agreement of NIC weekly ice chart and OSI SAF daily sea ice type product (OSI-403-c) for the same publication dates (12 different days) in the studied domain during January-March, 2019**

| | | NIC | | |
|---|---|---|---|---|
| | | Open water | First-year ice | Multi-year ice |
| OSI SAF | Open water | **90.0** | 10.0 | 0.1 |
| | First-year ice | 0.9 | **58.8** | 40.3 |
| | Multi-year ice | 0.0 | 1.0 | **99.0** |