# Peer review of "Classification of Sea Ice Types in Sentinel-1 SAR images"

_The Cryosphere, 2019_

## Referee Comment (RC1) · Anonymous Referee #1 · 28 Jun 2019

The manuscript by Park et al presents a machine learning technique to provide operational sea ice charts. The use of machine learning is interesting from the operational side as it enables processing of large volumes of data in a consistent manner. Such sea ice charts are of significant use for the both scientific community as well as the general community.

The manuscript is well structured, though would benefit from a spell and grammar check, e.g. Sentinel is misspelled several times. The authors claim that the method is free from subjective judgements though it requires the input of manually derived open water vs sea ice charts.

**Specific comments**
The authors claim that by using operationally provide sea ice charts they are avoiding

subjective decisions in the sea ice classification training data. To my knowledge the sea ice charts provided by the US National Ice Center are based on exactly such subjective decisions and are made to the best ability of the excellent sea ice experts working there. Please clarify how the training and validation data used here are not subjected to such decisions. The manually derived open water vs sea ice maps used as input training data could also be viewed as subjectively derived data.

It is unclear if only the sea ice parts of the images were incidence angle corrected using the sea ice estimated slope or if the whole image was corrected using the slopes derived for the sea ice part of the images. Please clarify. If the incidence angle slope derived for the sea ice were used over the entire image how would this affect the open water areas of the image? How are the slopes presented here derived? Mahmud et al 2018 showed that different sea ice types have different incidence angle dependent slopes. Have you considered using a sea ice type dependent slope factor? Moreover, how does the work by Mahmud et al 2018 fit in with the incidence angle dependencies presented here?

How do you define "good match" (P7 R3)? Temporal overlap? Spatial overlap? How was this manual selection of images performed? How were the open water vs sea ice charts, that are used as initial input into the classifier, derived? Are the open water areas separated from the sea ice areas at this stage of the classification process? Or are they classified at the same time as the sea ice types?

Given that one possible reason for the low accuracy in 2019 were stated to be insufficient training data (P9 R9-11), have you tested how the classification improves/remains the same if additional training data is added? Such an assessment would add strength to the accuracy of the method presented here.

Daily ice charts using Sentinel-1 data covering at least part of the study areas used here are provided by the Norway Ice Service. A comparison with their ice charts when they overlap, spatially and temporally, would have been beneficial and added strength

to the accuracy assessment. Partially as it would have provided daily instead of weekly ice charts to compare to.

Would your method work also outside the winter season? Has it been tested for other seasons? A majority of the shipping industry is dependent on sea ice charts year-round and a consistent method employed year-round is therefore beneficial.

Day of year might not correspond to the same temperature, fluxes and weather regimes. Have you considered using a weather variant input parameter instead of day of year? Such a parameter might be more suitable to capture the seasonality within the scenes.

How is the general accuracy derived? Is it a normalised average and does it account for the varying amounts of the different sea ice types? The overall accuracy of the sea ice classifier is only provided in the abstract and the conclusion. Please also provide it with the general results and discussion.

Is all level sea ice considered to have low backscatter? (P4 R7.)

How does the spatial resolution of 1 km affect the results? Have you tested using different spatial resolution sizes?

How was the value of 3 km set? (P7 R14)

Sea ice does not form bergy bits, the terminology used in Figure 3 of bergy water may therefore be misleading. Please change to a more appropriate term.

Visual inspection of Figure 7 seems to indicate that the method struggles with SAR image edges and that the same sea ice on different side of an image edge is classified differently, e.g. top right corner where one size of the edge is young ice and on the other side there is first-year ice. Please comment. Is the First-year ice observed in Figure 8 to the east of the old ice an artefact of a beam problem for the method?

What is your definition of New ice? For the ice types used here are you using the WMO

definitions? In Figure 9 the sea ice type FYI thin is include in the sea ice classification results. Please clarify what thin FYI means. Why is this class not used throughout?

Consider adding something indicating the semi-automatic aspect of the manuscript in the title. And also indicate that the Sentinel-1 scenes used here only reflect the winter season.

**Technical comments**
In many places references to the appropriate work is missing, e.g. P1, R27, P4 R15-16 and P6 R9-10. Please carefully revise the manuscript to include references to earlier work.
The method is claimed to be semi-automatic in the body of the manuscript though the word semi- is left out of the abstract of the manuscript. Please correct.
P1 R22-24. Unclear sentence, consider rewriting.
P.1, R.14-15. Unclear sentence, consider rewriting.
Minor grammatical errors are present throughout the manuscript, e.g. P2 R5 ... particularly in the cross-polarization... P2 R7 ... considering the relative... p2 R31 ...to train the classifier... P3 R15 To take advantage of the objective... Please carefully revise the English language throughout and pay particular attention to "the"
P2 R24 Region of interest -> region of study
P2 R25 coexit -> are found
P3 R2 National Ice Center (NIC) -> US National Ice Center
P3 R29. What does the precision of decimals mean here? That e.g. that the sea ice concentration can be 10.1%?
P4 R5. What is strong noise?
Consider using the commonly used term "open water" instead of "ocean" throughout. The same goes for ice free (IF) please use the much more commonly used open water (OW). MFYI could easily be confused with Multi-first-year ice. If MFYI is meant to indicate Mixed First Year Ice please change this or at least include this information in the header of the tables 4 and 5. In Table 4 and 5 are the MFYI meant to be used also

on the predicted header?

P4 R12. Senitnel -> Sentinel

P5 R9 "Furthermore, further..." consider changing this.

P6 R10-12 you argue that when training dataset are prepared manually the sample size is usually less then 20 images. Please provide several references to support this statement.

P7 R6. Unclear sentence please revise.

P7 R19-21 appears to be a description of the method please move them to the methods section.

P8 R13-14. Substantial work studying young ice type has been carried out by e.g. Dierking 2010. Consider referencing such works. The young sea ice has a large range in backscatter values from very low to high and are also often subjected to frost flowers, see e.g. excellent work by Isleifson. Stating that young ice "typically just look dark in the SAR images" may therefore not be strictly true.

P9 R13-15. Unclear sentence, please revise.

P9 R16. What does "charted in" mean?

P9 R17. What does "included wrong samples" mean? What makes a sample wrong?

P9 R19. When mentioning previous studies please provide references to these studies.

P9 R31-32. Daily ice charts building on Sentinel-1 are provided by the Ice service at the Meteorological office in Norway, so it is certainly true that this can be done.

Figure 4. What does the colorbar represent?

Figure 8. In Figure 3 and 7 the ice chart from 3 days later is used yet in Figure 8 the image from the same day is used. Please be consistent in which time interval is used for these weekly ice charts.

**References**

Mahmud, M.S., T. Geldsetzer, S. E. L. Howell, J. J. Yackel, V. Nandan, and R. K. Scharien, Incidence Angle Dependence of HH-Polarized C and L-Band Wintertime

Backscatter over Arctic Sea Ice, EEE Transactions on Geoscience and Remote Sensing, 2018.

W. Dierking, Mapping of Different Sea Ice Regimes Using Images from Sentinel-1 and ALOS Synthetic Aperture Radar, IEEE Transactions on Geoscience and Remote Sensing, vol. 48, no. 3, pp. 1045–1058, March 2010.

---

## Referee Comment (RC2) · Anonymous Referee #2 · 16 Jul 2019

In this paper the authors propose a classification method for determining ice types in Sentinel-1 SAR images. The structure and methodology are similar to those found in other studies, and overall the paper reads reasonably well. The study makes good use of recent published work by the authors on denoising Sentinel-1 SAR images, and improvements to the calculation of texture information in these images.

Specific comments:

To the best of my knowledge, this is the first study to examine the classification of Sentinel-1 SAR images for determination of sea ice types. These images are of great interest to the scientific and operational community. As the authors point out, the images are noisy, and the residual noise after the ESA correction is still significant. Certainly, the ability to classify ice types from such noisy images is of great use. However,

[Figure]

I have difficulty following some of the claims made, in particular in the abstract and introduction. Primarily, I am not certain if it is clear to the authors that operational ice charts are generated manually, and contain significant bias and other possible errors of subjectivity. It is a little difficult to find information about this online, but the studies by Partington et al. (2003) and in the text by Johannessen et al. (2006) clearly state that the preparation of NIC charts (former reference) and AARI charts (latter reference) is through manual inspection of various sources of satellite imagery and other sources of data. Other studies (such as J. Karvonen, 2015) look at the accuracy of manual analyses by ice analysts. Training using a large volume of these charts would reduce operator-to-operator bias, but not the overall bias these charts are believed to contain since they are produced in the interest of marine safety. Based on this, the claim in the abstract and elsewhere that the use of ice charts allows training/testing data 'void of biased subjective decisions' should be revised.

The 'novelty' of using ice charts in this way as training data should be clarified. These charts are fairly similar to the training data that was used for the sea ice type classi-fication study by Zakhvatkina [2013], where homogeneous areas identified by trained ice analysts are used. Image analysis charts, which are very similar to daily ice charts with the exceptions that they are based only on the SAR imagery, are used directly as training data in the study by Wang et al. [2017]. In that study the ice concentration information was used directly in the same manner as ice type in the present study (the available charts were mapped to the SAR image latitude and longitude), however it was ice concentration information that was used, not ice type. These similarities should be discussed.

Random forest classifiers are very popular at this time, and have been shown to be useful in many studies. To better motivate the present study, I suggest the authors compare their method to a multi-class random forest. In particular, the reference given for choosing the one-vs-all classification scheme as compared to multiclass problem is not closely related to the problem at hand. Did the authors try the multiclass method?
Given that the motivation here is for operational implementation, it would be of interest to know if the mutliclass method performs similarly, and the computation time difference between the binary one-vs-all method and multiclass method. I have a similar question regarding the use of all Haralick texture features. How long did it take to calculate these features over the 64 grey levels used here? Are all features needed, or is it not relevant (in the sense that the additional time required and change in accuracy is not significant). If the main contribution is to be the classifier itself, then a more careful examination of the method should be carried out. It would also be very interesting to see how the denoising methods they have developed lead to improved ice type classification. I am not sure if that would be difficult. Without this information, others are likely to attempt ice type classification without following rigorous denoising procedures. With this information, this piece of work could be a much stronger contribution to the sea ice community.

In the end, it is found that the classification accuracies are higher when considering only three classes, first-year ice, muliti-year ice and open water. Could the authors add a little discussion to the conclusions as to if they envision a three-class or five-class operational implementation? If it is three-class, would they recommend using ice types from another sensor as training data? Some discussion on how the method is expected to work for other times of year should also be included.

page 6 - line 10 - Can the authors explain what they mean here by a 'sparse dataset' and why a dataset used for ice/water and ice types from SAR imagery would be considered a 'sparse dataset'? I am not sure I follow this line of reasoning.

page 6 - line 32 - Why is the 'Richard's curve' chosen over a typical curve fit? Do the authors obtain more robust or interpretable results using this method? Please provide more context.

page 7 - If I understand correctly, the authors manually selected 57 image (or do the authors mean scenes here?) for training and testing from a set consisting of 958 images (or again is this scenes)? Can they say something about these 57? Are they from similar geographic regions? times of year? specific features? Going through 958 images manually to choose a training data set is not automated. Using an ice type product generated in an automated manner from another sensor (for example open water/FYI/MYI from passive microwave data or scatterometer data) could provide an automated workflow.

page 8 and Figure 6 - What method was used to determine the feature importance score and why was this method chosen? How is this score calculated?

Technical comments

1) abstract - overall accuracies vs. overall accuracy - Be consistent in your use of plurals here

2) abstract - In what way would this work support automated ice charting? Were the authors thinking that fewer operational (manual) charts would need to be produced? Clarification of this point would be helpful.

3) page 1 - line 12 - 'In most of the previous works...'... please provide a few references in this sentence to the works you have in mind.

4) page 1 - lines 12-15 and lines 20. I reiterate my earlier point. Ice charts are generated manually by trained analysts. Although they are available in the public domain, and this means using these charts directly relieves the individual designing the classification algorithm from the 'laborious' and possibly biased process of manually choosing training and testing data, it does not enable an automated workflow.

5) page 1 - line 20 - Again, ice charts are generated by humans. They contain human error. They are often produced under a strong time constraint, and in the interest of marine safety (the latter point meaning they likely contain bias to ensure safety).

6) page 3 - line 25 - 'ice edge determined from AMSR-E' - an ice edge cannot be determined from AMSR-E without using an algorithm. Which algorithm was used?

[Figure]

Please revise.

7) page 3 - lines 28-29 - I don't know what the authors mean by 'has a precision of decimals'.

8) page 3 - line 26 - Similar comment regarding the ice edge determined from SAR - a methodology must have been used to get this ice edge. Was it visual inspection, or another method? Please revise.

9) page 4 - line 3 - better than what?

10) page 5 - line 2 - wording is not specific - many of the previously developed methods - methods for what? These references are a mixture of ice/water and ice type classification studies. These two tasks are different from the perspective of a computer algorithm. Also, a reference to Shokr [1991] should be included.

11) page 5 - lines 9-11 - I don't understand this sentence, what is averaged for multiple distances, and what is the normalized GLCM?

12) page 5 - line 5 - direction? or should this be orientation?

13) page 5 - line 12 - The term spatial resolution is not clear. Some authors consider this the scales that are resolved. It may be better to state that the spacing between the GLCM texture feature windows is 1km? (or please reword if I am not interpreting this point correctly).

14) page 5 - It would be nice to have the Haralick features listed in a table, and to provide a brief rationale for including all of them in the study. Information as to how long it took to calculate these features using 64 grey levels for their set of imagery is also important.

15) page 5 - the number of Haralick features is referred to inconsistently as 13 on line 7 and 26 on line 25. The 26 is likely just accounting for the two polarizations, but the two should be referred to in a consistent manner. Similarly on page 7 lines 22-23, please

use either 'Haralick texture features' or 'texture features' consistently when describing the three classifiers.

16) page 6 - line 16 'they are' - what are 'they'? Is this the number of operations?

17) page 7 - If a binary ice/water classifier is 'simple' (line 6), why are the authors starting with ice type classification? I suggest this be reworded.

18) page 8 - lines 20-21 - The sentence starting with, 'Since the training and test datasets were extracted from the same...' I find a little out of place. With this placement, it seems like it is trying to account for the results from FC2 and FC3. It might be better to start this one with 'When the evaluation is carried out with the 2018 data, the training and test datasets....'

19) page 9 - line 32 - 'capturing' should be 'to capture'

20) page 9 - line 32 -'more details' - as compared to what?

21) Figure 3 - Could the authors provide some information in the text as to what the map of partial concentration is (top right). Is this the partial concentration of the dominant ice type for the given polygon?

22) Figures 3,7,8 and 9 should have geolocation data provided.

23) all numbers less than ten should be written out in words, eg., 3 -> three

References

A comparison between high-resolution EO-based and ice analyst-assigned sea ice concentrations, Juha Karvonen and others, IEEE Journal of Selected Topics in Applied Earth Observations and Remote Sensing, 8(4):1-9, 2015.

Evaluation of second-order texture parameters for sea ice classification from radar images, Mohammed E. Shokr, Journal of Geophysical Research, 96(C6),10,625-10640, 1991.

Late twentieth century Northern Hemisphere sea-ice record from the U.S. National Ice Center ice charts, Kim Partington, Tom Flynn, Doug Lamb, Cheryl Bertoia and Kyle Dedrick, Journal of Geophysical Research, 108(C11), doi:10.1029/2002JC001623, 2003.

Remote sensing of sea ice in the Northern Sea route: Studies and applications, Ola M. Johannessen and others, Springer Science and Business Media, 472 pages, 2006.

Sea ice concentration estimation during freeze-up from SAR imagery using a convolutional neural network, Lei Wang, K Andrea Scott and David A Clausi, Remote Sensing, 9(5), doi:10.3390/rs9050408, 2017.

---

## Referee Comment (RC3) · Anonymous Referee #3 · 17 Jul 2019

General comments The manuscript by Park et al presents Random Forest based classifier (Python Scikit-Learn) for sea ice type classification from dual pol (HH-HV) Sentinel-1 images which were collected during the winter period only. Training dataset were collected from the National Ice Center weekly ice chart and the classification algorithm exploits standard GLCM features along with some additional features. Since the launch of Sentinel-1 SAR sensors (a+b), it continuously monitoring Arctic Sea ice with high spatial and temporal resolution and an automated sea ice type classification product from high resolution SAR is highly desirable and relevant for the sea ice monitoring community. Having said that, there are several limiting factors which are preventing researchers to come up with a robust SAR based sea ice type classification scheme, (1) Backscatter variation due to varying incidence angle, along with sensor

specific noise related issues (2) Backscatter variation due to seasonal changes (winter – melt- early summer - summer) In this manuscript authors performed a denoising technique which was developed by authors previously (Park et al., 2018), and a standard linear incidence angle correction (Section 2.2.3). It is important to note here that different sea ice types have different incidence angle dependency. Moreover, if the incidence angle correction was applied over all classes (i.e. including open water), it will most likely not contribute to the robustness of the classifier. Backscatter variation due to seasonal changes is completely ignored in the presented manuscript. As authors aimed to develop an operational system, in my opinion authors cannot ignore this major issue completely. Specific comments The proposed classification scheme is based on Python Scikit-Learn library and GLCM features, this kind of classification scheme is well known and published several times for different frequency bands. Therefore the current manuscript is very limited in terms of innovation. What I find slightly different is use of weekly ice chart for training data generation. However I am also concerned about the automated training data generation as it is not clear which images were used to generate the ice chart by NIC and there is a high probability that that the ice chart polygons will not match that Sentinel-1 mosaics ice types. Hence there is a high risk that the classifiers were trained with wrong training data. Authors mentioned that 57 images were selected manually where only 'ice egdes' match well with the ice chart. In my opinion, this is also a manual selection of training data which authors criticised in the introduction section. The selection (and definitions) of ice types for SAR based sea ice classification scheme is crucial. The 5 class classifier has some classes which might be very close to each other in terms of backscatter and texture. This might be the main reason for significantly low classification accuracy. I would recommend the authors to restrict the classifier for 4 classes (Open Water, Young Ice, FYI and old ice). A seasonal assessment of the classification scheme is missing. It the most important issue to address and without this assessment it would not be reasonable to claim the scheme to be either operational or innovative. Due to above mentioned major issues I didn't listed any technical corrections, I would kindly invite the authors to address this

issues in future. Due to the lack of innovation and failed to address the basic issues, at current stage I can only recommend the manuscript to be rejected despite it is well within the scope of The Cryosphere.

―――――――――――――――――――――――――――

---

## Author Comment (AC1) · 2 Dec 2019

Please see pdfs in the supplement

Please also note the supplement to this comment:
https://www.the-cryosphere-discuss.net/tc-2019-127/tc-2019-127-AC1-supplement.zip
* * *

---

## Author Comment (AC2) · 2 Dec 2019

Please see pdfs in the supplement

Please also note the supplement to this comment:
https://www.the-cryosphere-discuss.net/tc-2019-127/tc-2019-127-AC2-supplement.zip
* * *

---

## Author Comment (AC3) · 2 Dec 2019

Please see pdfs in the supplement

Please also note the supplement to this comment:
https://www.the-cryosphere-discuss.net/tc-2019-127/tc-2019-127-AC3-supplement.zip

---

## Author Response (AR1)

**General comments**

The manuscript by Park et al presents a machine learning technique to provide operational sea ice charts. The use of machine learning is interesting from the operational side as it enables processing of large volumes of data in a consistent manner. Such sea ice charts are of significant use for the both scientific community as well as the general community.

The manuscript is well structured, though would benefit from a spell and grammar check, e.g. Sentinel is misspelled several times. The authors claim that the method is free from subjective judgements though it requires the input of manually derived open water vs sea ice charts.

First of all, we would like to thank the reviewer for the positive evaluation and providing important comments. We hope that all your concerns will be cleared after reading our responses and modifications made to the manuscript. Please find below our answers (in green) and modifications (deleted in red and added in blue) to your comments/suggestions/questions.

**Specific comments**

The authors claim that by using operationally provide sea ice charts they are avoiding subjective decisions in the sea ice classification training data. To my knowledge the sea ice charts provided by the US National Ice Center are based on exactly such subjective decisions and are made to the best ability of the excellent sea ice experts working there. Please clarify how the training and validation data used here are not subjected to such decisions. The manually derived open water vs sea ice maps used as input training data could also be viewed as subjectively derived data.

As the reviewer pointed, ice charting cannot be free from subjective decisions but it requires best knowledges from credible experts like national/international ice services rather than from anonymous individuals. The aim of this study is to address such an issue; however, we admit that the manual selection of training/validation dataset causes another subjectivity issue. Regarding the concern on subjective judgement in selecting training dataset, the open water vs sea ice charts are not derived manually; the boundary between open water and sea ice is extracted from the electronic ice chart. The manual part is to check if the SAR image show high image contrast at the position of the extracted ice-water boundary. Although the visual inspection along the ice-water boundary overlaid on SAR image is also not completely free from subjective judgements, it requires much less effort and expertise compared to drawing manual ice chart from the SAR image itself. We apologize for this confusion; we omitted an essential description. In the revised manuscript, we added detailed explanations about the procedure on selecting training dataset.

> **[Section 2.2.6]**
> In order to automate image selection, the ice edges in SAR images need to be identified first.  Since even  an  ice/water classifier has not been well developed yet for Sentinel-1, the image selection procedure has to be done manually in the beginning. However, once a classifier is generated with high accuracy, it can be used to automate the procedure, then the whole process in the proposed scheme will be fully automated. This is why the proposed algorithm is named "semi-" automated for now. Nevertheless, the manual selection is done by visual inspection of ice-water boundaries overlaid on SAR images. The ice-water boundary can be extracted easily from the reprojected ice chart by selecting the pixel borders of open water class. Then the SAR backscattering image contrasts across the ice-water boundaries are examined both in HH- and HV-polarization because the image contrast between ice-water is larger in HV and smooth level ice is better recognizable in HH.

It is unclear if only the sea ice parts of the images were incidence angle corrected using the sea ice estimated slope or if the whole image was corrected using the slopes derived for the sea ice part of the images. Please

clarify. If the incidence angle slope derived for the sea ice were used over the entire image how would this affect the open water areas of the image? How are the slopes presented here derived?

The whole image was corrected using the slopes derived for the sea ice regardless of the surface type. Since this correction is prior to the classification, sea ice type-specific correction cannot be made in this stage. Nevertheless, the bulk slope correction has known to be effective in literature (Zakhvatkina et al., 2013; Zakhvatkina et al., 2017). We added the following explanations and references to Section 2.2.3:
* * *
**[Section 2.2.3]**
Figure 4. shows two-dimensional histograms of incidence angle versus sigma nought for sea ice pixels in HH and HV polarization channels from Sentinel-1 data collected over sea ice and open water in the study area in winter 2018. From the Sentinel-1 dataset described in Section 2.1, sea ice pixels were extracted by using daily global sea ice edge products available from the EUMETSAT Ocean and Sea Ice Satellite Application Facilities (OSISAF). For HH polarization, the estimated slope was -0.200 dB/degree, which is slightly different from the estimation of the first-year ice (-0.24 dB/degree) in Mäkynen and Karvonen (2017) and in between the estimations for first-year ice (-0.22 dB/degree) and multi-year ice (-0.16 dB/degree) in Mahmud et al. (2018). For HV polarization, the estimated slope was only -0.025 dB/degree, which is much lower than the estimation in Mäkynen and Karvonen (2017), however, it is in line with the estimations from RADARSAT-2 (Leigh et al., 2014; Liu et al., 2015). We compensate for the incidence angle  dependence using the estimated slopes,  with respect to the nominal scene center angle of 34.5 degrees as reference. Although the incidence angle dependence changes with ice type and radar frequency (Mahmud et al., 2018), the compensation is done for all pixels in the image using a single value of mean slope because the ice types are not identified in this stage. Open water areas of the image are also affected; however, the correction is also beneficial since the incidence angle dependence for open water is stronger (-0.65 dB/degree for wind velocity of 5 m/s, computed from CMOD5 C-band geophysical model function in Hersbach et al., 2007), thus the corrected image has less incidence angle dependence.

**[References]**
Mahmud, M. S., Geldsetzer, T., Howell, S. E. L., Yackel, J. J., Nandan, V., and Scharien, R. K.: Incidence angle dependence of HH-polarized C- and L-band wintertime backscatter over Arctic sea ice, IEEE T. Geosci. Remote, 56(11), 6686-6698, doi:10.1109/TGRS.2018.2841343, 2018.
Hersbach, H., Stoffelen, A., and de Haan, S.: An improved C-band scatterometer ocean geophysical model function: CMOD5, J. Geophys. Res., 112, C03006, doi:10.1029/2006JC003743, 2007.
* * *
Mahmud et al 2018 showed that different sea ice types have different incidence angle dependent slopes. Have you considered using a sea ice type dependent slope factor? Moreover, how does the work by Mahmud et al 2018 fit in with the incidence angle dependencies presented here?

Typically, the slope for open water is higher than that for sea ice, thus the correction works in a way reducing the difference in sigma naught for open water as well. As the review pointed, different sea ice types have different incidence angle dependent slopes; however again, ice type-specific correction prior to ice type classification is controversial. Although estimating ice type dependent slope is not a part of this manuscript, we provide the values derived from the training/validation dataset for the review purpose only.

[Figure]

How do you define "good match" (P7 R3)? Temporal overlap? Spatial overlap? How was this manual selection of images performed? How were the open water vs sea ice charts, that are used as initial input into the classifier, derived?

Both temporal and spatial overlaps are important. Since the SAR image itself is potentially one of the sources for ice charting at the ice services, some images spatially match well with the shape of the polygons in ice chart. Temporal window length of 3 days from the publication date of ice chart was used for squeezing the number of images to make decisions of use/discard for training. We added a paragraph that explains processing details in the revised manuscript as follows:

**[Section 2.2.6]**

To train an ice type classifier, a set of collocated SAR images and ice charts is required. After the preprocessing of the ice chart including reprojection into the SAR image geometry, only the samples with spatially and temporally good matches should be fed to the training phase. Image selection is trivial, but not easy to automate. Since the weekly ice chart is made partly based on the SAR images acquired in the past three3 days from the date of publication, the ice edges in some images match well with those in the ice chart. In order to automate image selection, the ice edges in SAR images need to be identified first.  Since even  an  ice/water classifier has not been well developed yet for Sentinel-1, the image selection procedure has to be done manually in the beginning. However, once a classifier is generated with high accuracy, it can be used to automate the procedure, then the whole process in the proposed scheme will be fully automated. This is why the proposed algorithm is named "semi-" automated for now. Nevertheless, the manual selection is done by visual inspection of ice-water boundaries overlaid on SAR images. The ice-water boundary can be extracted easily from the reprojected ice chart by selecting the pixel borders of open water class. Then the SAR backscattering image contrasts across the ice-water boundaries are examined both in HH- and HV-polarization because the image contrast between ice-water is larger in HV and smooth level ice is better recognizable in HH.

After the image selection, the samples in the selected images are split randomly into training and test datasets with a ratio of 7:3. For the training dataset, further data selection is made by excluding the samples residing close to the polygon boundaries. This is to account for possible mismatch due to various reasons (e.g., ice drift, vector mapping error, image geocoding error, etc.). In this study, only the data from pixels more than 3 km away from the polygon boundaries was fed into the training process. Once the hyperparameter optimization is done, the RF classifier is trained for the training dataset. The trained classifier is then applied to the test dataset. For performance evaluation, we use confusion matrix and Cohen's kappa coefficient $\kappa$ (Cohen, 1960), which measures the agreement between two raters (in this study, they are the trained classifier and the reference ice chart) with taking account of the possibility of the agreement occurring by chance . The validation is done in the same way but using a completely independent dataset. The 2018 data was used to run the training phase. Among 958 images in total, we selected 57 images of which ice edges match well with the collocated ice chart. From the selected images, 6.4 million samples covering open water and sea ice were divided into training and test dataset.

Are the open water areas separated from the sea ice areas at this stage of the classification process? Or are they classified at the same time as the sea ice types?

At this stage, no classifier is introduced. The open water is classified at the same time as the sea ice types in the later stage.

Given that one possible reason for the low accuracy in 2019 were stated to be insufficient training data (P9 R9-11), have you tested how the classification improves/remains the same if additional training data is added? Such an assessment would add strength to the accuracy of the method presented here.

We tested the changes in classification accuracy with varying number of trained images. As the results shown below indicates, the accuracy increases rapidly until adding 15 images, but the improvement was not great after adding 20-25 images.

[Figure]

By following the test result, we revised the manuscript as below.
* * *
**[Section 3]**
However, the accuracy decrease from 2018 data to 2019 data was  at a similar level to the case of the five-class classification.This could have been caused by  inconsistent labeling in the reference ice chart.
* * *
Daily ice charts using Sentinel-1 data covering at least part of the study areas used here are provided by the Norway Ice Service. A comparison with their ice charts when they overlap, spatially and temporally, would have been beneficial and added strength to the accuracy assessment. Partially as it would have provided daily instead of weekly ice charts to compare to.

As far as we understand, the daily ice chart serviced by Met.no does not provide ice types; it provides ice concentration instead.

Would your method work also outside the winter season? Has it been tested for other seasons? A majority of the shipping industry is dependent on sea ice charts year-round and a consistent method employed year-round is therefore beneficial.

The method itself would work for other seasons if multiple classifiers are trained for each of the seasons. It is challenging to develop a single universal classifier that works for all seasons. Although we tried to make a classifier adapt to seasonal changes by including day of year as a feature (FC3), the result was not promising.

Day of year might not correspond to the same temperature, fluxes and weather regimes. Have you considered using a weather variant input parameter instead of day of year? Such a parameter might be more suitable to capture the seasonality within the scenes.

No, we didn't. As sea ice drifts continuously, weather variant information for each of the ice floes in the SAR image at the image acquisition time needs to be calculated and joined, and this may require complex and rigorous works. In order to simplify the problem, we used day of year only, but certainly other parameters need to be tested in the follow up paper. We are very interested in combining sea ice drift with ice type-specific texture changes in our future study. We added a discussion paragraph to the end of Section 3.
* * *
**[Section 3]**
Unfortunately, the proposed algorithm has several limitations. First of all, the variations in radar backscattering and its corresponding image textures due to seasonal changes were not properly captured. Although day-of-the-year was tested as a seasonality variable in the FC3 feature configuration, the result did not show any improvement. This is because day-of-the-year might not correspond to the same temperature, fluxes, and weather regimes.
* * *
How is the general accuracy derived? Is it a normalised average and does it account for the varying amounts of the different sea ice types? The overall accuracy of the sea ice classifier is only provided in the abstract and the conclusion. Please also provide it with the general results and discussion.

The overall accuracy in the abstract and conclusion is an average of the accuracies for each of the classes, thus the varying amounts of the different sea ice types are accounted for. In the revised manuscript, we adopted Cohen's Kappa to support the performance evaluation.
* * *
**[Section 2.2.6]**
For performance evaluation, we use confusion matrix and Cohen's kappa coefficient $\kappa$ (Cohen, 1960), which measures the agreement between two raters (in this study, they are the trained classifier and the reference ice chart) with taking account of the possibility of the agreement occurring by chance .
* * *
Is all level sea ice considered to have low backscatter? (P4 R7.)

No. They can have high backscatter in case frost flowers exist.
* * *
**[Section 2.2.2]**
For surfaces with low backscattering such as calm  open water and level sea ice without the presence of frost flowers on top, the effects from thermal noise contamination are visible not only in the backscattering image but also in some of the texture images (Park et al., 2019).

**[Section 3]**
This might be because the new ice has different types of recently formed ice including nilas, which is smooth but rafting can make rough features, and frost flowers, which introduces high surface roughness and volume scattering (Isleifson et al., 2014), thus the new ice can appear either featureless dark or complex bright in SAR image (Dierking, 2010). The large range in backscatter values makes it hard to define characteristic texture in the new ice patch.
* * *
How does the spatial resolution of 1 km affect the results? Have you tested using different spatial resolution sizes?

The spatial resolution was set as 1 km to meet with two conflicting requirements. The size of subwindow for texture calculation must be larger than the spatial scales of the major sea ice structures, e.g., cracks, ridges, rafting patterns, narrow leads, which are up to a few hundred meters. The resolution needs to be comparable with modern high-resolution sea ice products from SAR. The regional ice concentration map published by Met.no has also 1 km resolution.

How was the value of 3 km set? (P7 R14)

There is no specific reason but we assumed that positional disagreements can be up to this amount in case the same SAR image was also used in NIC ice charting.

Sea ice does not form bergy bits, the terminology used in Figure 3 of bergy water may therefore be misleading. Please change to a more appropriate term.

You are right. The "bergy water" label was removed from Figure 3.

Visual inspection of Figure 7 seems to indicate that the method struggles with SAR image edges and that the same sea ice on different side of an image edge is classified differently, e.g. top right corner where one size of the edge is young ice and on the other side there is first-year ice. Please comment. Is the First-year ice observed in Figure 8 to the east of the old ice an artefact of a beam problem for the method?

As the reviewer pointed, the proposed method partly fails in resulting consistent result when the same sea ice on different side of an image edge is classified. Regarding the artifacts in Figure 8, it is most likely due to the wind roughened surface. When the sea surface is roughened by wind, it creates texture and easily misclassified as ice.
https://www.star.nesdis.noaa.gov/sod/mecb/sar/AKDEMO_products/APL_winds/wind_images2/2019-02/S1A_ESA_2019_02_08_07_21_08_0602925668_003.77E_79.54N_HH_C5_GFS05CDF_wind.png
Regarding these limitations, we added a discussion paragraph to the end of Section 3.

> **[Section 3]**
>
> Unfortunately, the proposed algorithm has several limitations. First of all, the variations in radar backscattering and its corresponding image textures due to seasonal changes were not properly captured. Although day-of-year was tested as a seasonality variable in the FC3 feature configuration, the result did not show any improvement. This is because day-of-the-year might not correspond to the same temperature, fluxes, and weather regimes. Second, the proposed method struggles when the same sea ice is located on different marginal sides of SAR images because the incidence angle dependence could not be normalized perfectly. An example of such a failure can be seen along the image boundaries at 80N, 35E and 82.5N, 60E, approximately. Third, some artifacts were observed under an extreme marine condition. In the classified results in the bottom right panel of Figure 8, there is a misclassified FYI patch (yellow) in the open water area. According to the NOAA SAR wind image service, ANSWRS 2.0, the wind speed ranged from 17 to 21 m/s at the time of image acquisition heavily roughing the water surface.

What is your definition of New ice? For the ice types used here are you using the WMO definitions?

Yes, we follow the WMO definitions as the NIC ice chart does. In the WMO Sea Ice Nomenclature (WMO No. 259, volume 1 – Terminology and Codes), the definition of New ice is, "A general term for recently formed ice which includes *frazil ice*, *grease ice*, *slush* and *shuga*. These types of ice are composed of ice crystals which are only weakly frozen together (if at all) and have a definite form only while they are afloat."

In Figure 9 the sea ice type FYI thin is include in the sea ice classification results. Please clarify what thin FYI means. Why is this class not used throughout?

The label was wrong. It was just FYI without "thin". As the bottom panels (SAR results) are irrelevant in the context of the corresponding description, they were removed in the revised manuscript.

Consider adding something indicating the semi-automatic aspect of the manuscript in the title. And also indicate that the Sentinel-1 scenes used here only reflect the winter season.

We changed the title as "Classification of Winter Sea Ice Types in Sentinel-1 SAR images". The word "semi-automatic" and "semi-automated" appears several times in the revised manuscript including abstract.

**Technical comments**

In many places references to the appropriate work is missing, e.g. P1, R27, P4 R15-16 and P6 R9-10. Please carefully revise the manuscript to include references to earlier work.
Corrected.

The method is claimed to be semi-automatic in the body of the manuscript though the word semi- is left out of the abstract of the manuscript. Please correct.
Corrected.

P1 R22-24. Unclear sentence, consider rewriting.
Corrected.

P.1, R.14-15. Unclear sentence, consider rewriting.
Corrected.

Minor grammatical errors are present throughout the manuscript, e.g. P2 R5 : : : particularly in the cross-polarization: : : P2 R7 : : : considering the relative: : : p2 R31 : : :to train the classifier: : : P3 R15 To take advantage of the objective: : : Please carefully revise the English language throughout and pay particular attention to "the"
Corrected.

P2 R24 Region of interest -> region of study
Corrected.

P2 R25 coexit -> are found
Corrected.

P3 R2 National Ice Center (NIC) -> US National Ice Center
Corrected.

P3 R29. What does the precision of decimals mean here? That e.g. that the sea ice concentration can be 10.1%?
Reworded.

> Note that the ice concentration label in the SIGRID-3 format is assigned in an increment of 10%.

P4 R5. What is strong noise?
Image contaminations as in Park et al., 2018

Consider using the commonly used term "open water" instead of "ocean" throughout. The same goes for ice free (IF) please use the much more commonly used open water (OW). MFYI could easily be confused with Multi-first-year ice. If MFYI is meant to indicate Mixed First Year Ice please change this or at least include this information in the header of the tables 4 and 5. In Table 4 and 5 are the MFYI meant to be used also on the predicted header?
Corrected.

P4 R12. Senitnel -> Sentinel
Corrected.

P5 R9 "Furthermore, further: : :" consider changing this.
Corrected.

P6 R10-12 you argue that when training dataset are prepared manually the sample size is usually less then 20 images. Please provide several references to support this statement.

> When training dataset is prepared by manual work (i.e., manual classification by human expert), the number of images is not large, usually less than 20 (e.g., 12 scenes in Zakhvatkina et al., 2013; 20 scenes in Leigh et al., 2013; 2 scenes in Liu et al., 2015; 4 scenes in Ressel et al., 2015).

P7 R6. Unclear sentence please revise.
Corrected.

P7 R19-21 appears to be a description of the method please move them to the methods section.
Following the reviewer's suggestion, they were moved to the end of the Section 2.

P8 R13-14. Substantial work studying young ice type has been carried out by e.g. Dierking 2010. Consider referencing such works. The young sea ice has a large range in backscatter values from very low to high and are also often subjected to frost flowers, see e.g. excellent work by Isleifson. Stating that young ice "typically just look dark in the SAR images" may therefore not be strictly true.
Revised.

> This might be because the new ice has different types of recently formed ice including nilas, which is smooth but rafting can make rough features, and frost flowers, which introduces high surface roughness and volume scattering (Isleifson et al., 2014), thus the new ice can appear either featureless dark or complex bright in SAR image (Dierking, 2010). The large range in backscatter values makes it hard to define characteristic texture in the new ice patch.

P9 R13-15. Unclear sentence, please revise.
Corrected.

P9 R16. What does "charted in" mean?

Reworded.

> The  same ice floe (red outline) is  classified differently in  two different ice charts (old ice on the left panel and first-year ice on the right panel) although it looks almost the same in the corresponding SAR backscattering images. It should be noted that training with ice chart might have included  mislabeled small features  even if the image selection based on ice edge matching was successful.

P9 R17. What does "included wrong samples" mean? What makes a sample wrong?

Reworded.

> It should be noted that training with ice chart might have included  mislabeled small features  even if the image selection based on ice edge matching was successful.

P9 R19. When mentioning previous studies please provide references to these studies.

Added numbers from the references.

> Therefore, the lower classification accuracies compared to those in the previous studies (80% in Zakhvatkina et al., 2013; 91.7% in Liu et al., 2015; 87.2% in Aldenhoff et al., 2018), which used manually classified ice maps as training and validation reference, are expected.

P9 R31-32. Daily ice charts building on Sentinel-1 are provided by the Ice service at the Meteorological office in Norway, so it is certainly true that this can be done.

The ice chart from Met.no does not provide ice type but ice concentration.

Figure 4. What does the colorbar represent?

We accidentally omitted the title for the colorbar. It represents the number of subimages used for computing the histogram.

Figure 8. In Figure 3 and 7 the ice chart from 3 days later is used yet in Figure 8 the image from the same day is used. Please be consistent in which time interval is used for these weekly ice charts.

Since the reference ice chart is published weekly, the same NIC ice chart both in Figure 7 and 8 is supposed to be valid for both dates of Figure 7 and 8, but it is not as shown. The common time interval of 3 days in Figure 3 and 7 is just a coincidence. It can be one day or two days depending on the date of SAR image that used as source materials for ice charting at ice services. If the satellite images acquired two days prior to the publication of weekly ice chart, then the overall distribution of ice would represent the status of that time. Note that among the 57 images used for training, 42 and 35 percent of the images were acquired 2 and 3 days prior to the publication date of the corresponding weekly ice chart, respectively.

**Anonymous Referee #2**

**General comments**

In this paper the authors propose a classification method for determining ice types in Sentinel-1 SAR images. The structure and methodology are similar to those found in other studies, and overall the paper reads reasonably well. The study makes good use of recent published work by the authors on denoising Sentinel-1 SAR images, and improvements to the calculation of texture information in these images.

First of all, we would like to thank the reviewer for the positive evaluation and providing important comments. We hope that all your concerns will be cleared after reading our responses and modifications made to the manuscript. Please find below our answers (in green) and modifications (deleted in red and added in blue) to your comments/suggestions/questions.

**Specific comments**

To the best of my knowledge, this is the first study to examine the classification of Sentinel-1 SAR images for determination of sea ice types. These images are of great interest to the scientific and operational community. As the authors point out, the images are noisy, and the residual noise after the ESA correction is still significant. Certainly, the ability to classify ice types from such noisy images is of great use. However, I have difficulty following some of the claims made, in particular in the abstract and introduction. Primarily, I am not certain if it is clear to the authors that operational ice charts are generated manually, and contain significant bias and other possible errors of subjectivity. It is a little difficult to find information about this online, but the studies by Partington et al. (2003) and in the text by Johannessen et al. (2006) clearly state that the preparation of NIC charts (former reference) and AARI charts (latter reference) is through manual inspection of various sources of satellite imagery and other sources of data. Other studies (such as J. Karvonen, 2015) look at the accuracy of manual analyses by ice analysts. Training using a large volume of these charts would reduce operator-to-operator bias, but not the overall bias these charts are believed to contain since they are produced in the interest of marine safety. Based on this, the claim in the abstract and elsewhere that the use of ice charts allows training/testing data 'void of biased subjective decisions' should be revised.

Thank you very much for pointing an important issue. We revised the abstract and some parts in introduction as belows.
* * *
**[Abstract]**
A new Sentinel-1 image-based sea ice classification algorithm using a machine learning-based model trained in a semi-automated manner is proposed to support  daily ice charting. Previous studies mostly rely on manual work in selecting training and validation data. We show that the  readily available ice charts from  the operational ice services can reduce the number of manual works in preparation of large amounts of training/testing data. Furthermore, they reduce the inconsistent decisions in the classification algorithm by indirectly exploiting the best ability of the sea ice experts working at the operational ice services.

**[Section 1]**
The use of a public ice chart as training and validation reference data may help in solving the validation problem and enabling automation. The preparation of a public ice chart is also through manual inspection of various sources of satellite imagery and other sources of data (Partington et al., 2003; Johannessen et al., 2006); however, training using a large volume of these charts would reduce operator-to-operator bias. The overall bias may exist since the public ice charts are produced in the interest of marine safety. Nevertheless, as the human interpretation available in the ice chart is currently considered as the best available information of sea ice (Karvonen et al., 2015), the best practice to make a sea ice type classifier is to train with the public ice chart so that the best knowledge of certified ice analysts is mimicked.
* * *
The 'novelty' of using ice charts in this way as training data should be clarified. These charts are fairly similar to the training data that was used for the sea ice type classification study by Zakhvatkina [2013], where homogeneous areas identified by trained ice analysts are used. Image analysis charts, which are very similar to

daily ice charts with the exceptions that they are based only on the SAR imagery, are used directly as training data in the study by Wang et al. [2017]. In that study the ice concentration information was used directly in the same manner as ice type in the present study (the available charts were mapped to the SAR image latitude and longitude), however it was ice concentration information that was used, not ice type. These similarities should be discussed.

They are now included in the introduction as below.
* * *
**[Section 1]**

In  many of the previous works on ice-water and/or sea ice classification (Soh and Tsatsoulis, 1999; Zakhvatkina et al., 2013; Leigh et al., 2014; Liu et al., 2015; Ressel et al., 2015; Zakhvatkina et al., 2017; Aldenhoff et al., 2018), the training and validation were done using manually produced ice maps. Although the authors claimed that the manual ice maps were drawn by ice experts, the selection of SAR scenes and interpretation can be inconsistent, and the number of samples was not enough to generalize the results because of the laborious manual work. Therefore, increasing objectivity is crucial, and automating the classification process is encouraged. The idea of training using SAR images and accompanying image analysis charts, which is a raw interpretation of SAR images by trained ice analysts working at operational ice services, were tested for sea ice concentration estimation by Wang et al. (2017); however, such image analysis charts are not accessible to the public.
* * *
Random forest classifiers are very popular at this time, and have been shown to be useful in many studies. To better motivate the present study, I suggest the authors compare their method to a multi-class random forest. In particular, the reference given for choosing the one-vs-all classification scheme as compared to multiclass problem is not closely related to the problem at hand. Did the authors try the multiclass method? Given that the motivation here is for operational implementation, it would be of interest to know if the mutliclass method performs similarly, and the computation time difference between the binary one-vs-all method and multiclass method.

Yes, we tried the multi-class random forest as well. The main reason for using one-vs-all scheme is to check the difference in feature importance per each of the classes. As the multi-class random forest gives a single feature importance, it is impossible to see the differences among the classes. The performance of the one-vs-all binary approach were slightly better than that of the multi-class method as shown in the table below, and this is in line with the results in Adnan and Islam, 2015. However, the computation times of the multi-class method were 1/3 and 1/2 compared to those of the binary one-vs-all method for the cases of 5- and 3-class.

| Feature configurations | FC1 | | FC2 | | FC3 | |
|---|---|---|---|---|---|---|
| Number of classes | 5 classes | 3 classes | 5 classes | 3 classes | 5 classes | 3 classes |
| Overall accuracy (multi-class) | 58.5 | 86.2 | 58.0 | 86.1 | 57.4 | 75.5 |
| Overall accuracy (One-vs-all) | 58.8 | 86.2 | 58.4 | 86.7 | 54.6 | 75.8 |
| Cohen's kappa (multi-class) | 0.66 | 0.79 | 0.66 | 0.79 | 0.52 | 0.53 |
| Cohen's kappa (One-vs-all) | 0.67 | 0.80 | 0.67 | 0.80 | 0.49 | 0.53 |

Adnan, M. N., and Islam, M. Z., One-Vs-All Binarization Technique in the Context of Random Forest, Proc. European Symposium on Artificial Neural Networks, Computational Intelligence and Machine Learning, Bruges (Belgium), 22-24 April 2015.

I have a similar question regarding the use of all Haralick texture features. How long did it take to calculate these features over the 64 grey levels used here? Are all features needed, or is it not relevant (in the sense that the additional time required and change in accuracy is not significant).

There are some high correlations between the features as shown in the heatmap below. For example, ASM and Diff Var, Cont and Var, Sum Ent and Ent have highly correlated each other. Removing some of them may not lead to significant decrease in prediction accuracy, but the computational efficiency is out of the scope of this study. The computation time for extracting Haralick texture features per image is approximately five minutes in the given conditions (64 grey levels, 25 x 25 pixels of subwindow size) with an Intel i7 quad-core processor.

[Figure]

If the main contribution is to be the classifier itself, then a more careful examination of the method should be carried out. It would also be very interesting to see how the denoising methods they have developed lead to improved ice type classification. I am not sure if that would be difficult. Without this information, others are likely to attempt ice type classification without following rigorous denoising procedures. With this information, this piece of work could be a much stronger contribution to the sea ice community.

We conducted an additional test by following your suggestion, and the results are added to the revised manuscript. As shown in the table below, the textural denoising led to improved accuracies for all the classes except New ice.

**[Section 3]**

To see how the denoising step in Section 2.2.2 led to improvements in the classification accuracies, the same training and evaluation were conducted for the same dataset without applying the textural noise correction (Table 4). In both FC1 and FC2, the accuracies improved for young ice (+8.2-9.8%) and first-year ice (+9.2-11.6%) which were most pronounced compared to those for open water (+1.7%) and old ice (+1.2-1.7%). On the contrary, a small accuracy decrease was observed for new ice (-2.8-4.7%). Nevertheless, the improvement in kappa (+0.05) demonstrates a clear improvement in the overall classification result.

Table 4: Changes in classification accuracies before and after applying textural denoising

| class | case | | | | | | | | |
|---|---|---|---|---|---|---|---|---|---|
| | FC1 | | | FC2 | | | FC3 | | |
| | Thermal denoising only | Textural denoising applied | difference | Thermal denoising only | Textural denoising applied | difference | Thermal denoising only | Textural denoising applied | difference |
| OW | 88.4 | 90.1 | +1.7 | 88.9 | 90.6 | +1.7 | 88.0 | 85.4 | -2.6 |
| NI | 30.2 | 28.0 | -2.8 | 27.7 | 23.0 | -4.7 | 31.8 | 23.9 | -7.9 |
| YI | 34.9 | 44.7 | +9.8 | 36.2 | 44.6 | +8.2 | 43.4 | 51.5 | +8.1 |
| FYI | 29.3 | 38.9 | +9.6 | 30.4 | 42.0 | +11.6 | 38.0 | 47.0 | +9.0 |
| OI | 91.5 | 92.7 | +1.2 | 90.3 | 91.7 | +1.4 | 75.2 | 66.3 | -8.9 |
| kappa | 0.62 | 0.67 | +0.05 | 0.62 | 0.67 | +0.05 | 0.54 | 0.49 | -0.05 |

In the end, it is found that the classification accuracies are higher when considering only three classes, first-year ice, muliti-year ice and open water. Could the authors add a little discussion to the conclusions as to if they envision a three-class or five-class operational implementation? If it is three-class, would they recommend using ice types from another sensor as training data? Some discussion on how the method is expected to work for other times of year should also be included.

We added relevant discussions to Section 3 and 4.

**[Section 4]**

Based on the results, we envisage that three-class ice type classification from SAR imagery would be useful for making a global sea ice type product like EUMETSAT OSI-403-C (Aaboe et al., 2014) with higher spatial resolution.

**[Section 3]**

Unfortunately, the proposed algorithm has several limitations. First of all, the variations in radar backscattering and its corresponding image textures due to seasonal changes were not properly captured. Although day-of-year was tested as a seasonality variable in the FC3 feature configuration, the result did not show any improvement. This is because day-of-the-year might not correspond to the same temperature, fluxes, and weather regimes.

page 6 - line 10 - Can the authors explain what they mean here by a 'sparse dataset' and why a dataset used for ice/water and ice types from SAR imagery would be considered a 'sparse dataset'? I am not sure I follow this line of reasoning.

The sentence was reworded as belows.

In the literatures about sea ice classification, the SVM was used often because by nature it works relatively well even when the for sparse number of dataset datasets are small. When training dataset is prepared by manual work (i.e., manual classification by human expert), the number of images is not large, usually less than 20 (e.g., 12 scenes in Zakhvatkina et al., 2013; 20 scenes in Leigh et al., 2013; 2 scenes in Liu et al., 2015; 4 scenes in Ressel et al., 2015).

page 6 - line 32 - Why is the 'Richard's curve' chosen over a typical curve fit? Do the authors obtain more robust or interpretable results using this method? Please provide more context.

We added more explanations.

> Classification scores with values ranging from 0 (worst performance) to 1 (best performance) are evaluated for each node of the grid and are interpolated between the nodes by curve fitting. The Richards' Curve (Richards, 1959) was used as the fit model because it allows easy estimation of the model's maximum value.

page 7 - If I understand correctly, the authors manually selected 57 image (or do the authors mean scenes here?) for training and testing from a set consisting of 958 images (or again is this scenes)? Can they say something about these 57? Are they from similar geographic regions? times of year? specific features? Going through 958 images manually to choose a training data set is not automated. Using an ice type product generated in an automated manner from another sensor (for example open water/FYI/MYI from passive microwave data or scatterometer data) could provide an automated workflow.

If you mean the image-subimage things, it is SCENES here. As in Section 2.1, a total of 958 scenes were acquired, and the selected 57 scene are from various geographic region within the study area and various time of year. Using ice type product from passive microwave data or scatterometer data cannot help the image selection procedure due to the large difference in spatial resolution of them and SAR. Regarding the automation issue, we clarify throughout the revised manuscript that the developed algorithm is "semi-" automated.

> **[Section 2.2.6]**
> In order to automate image selection, the ice edges in SAR images need to be identified first.  Since even  an  ice/water classifier has not been well developed yet for Sentinel-1, the image selection procedure has to be done manually in the beginning. However, once a classifier is generated with high accuracy, it can be used to automate the procedure, then the whole process in the proposed scheme will be fully automated. This is why the proposed algorithm is named "semi-" automated for now. Nevertheless, the manual selection is done by visual inspection of ice-water boundaries overlaid on SAR images. The ice-water boundary can be extracted easily from the reprojected ice chart by selecting the pixel borders of open water class. Then the SAR backscattering image contrasts across the ice-water boundaries are examined both in HH- and HV-polarization because the image contrast between ice-water is larger in HV and smooth level ice is better recognizable in HH.

page 8 and Figure 6 - What method was used to determine the feature importance score and why was this method chosen? How is this score calculated?

> **[Section 2.3]**
> For each sub-classifier, each  texture feature has a different weight in decision making. The fraction of the samples that each texture feature contributes can be used to compute the relative importance of the features, and the averaged estimates of them over several randomized trees serve as an indicator of feature importance (Louppe, 2014).
>
> **[References]**
> Louppe, G.: Understanding random forests: From theory to practice, PhD Thesis, U. of Liege, 2014.

**Technical comments**

1) abstract - overall accuracies vs. overall accuracy - Be consistent in your use of plurals here
   Corrected.

2) abstract - In what way would this work support automated ice charting? Were the authors thinking that fewer operational (manual) charts would need to be produced? Clarification of this point would be helpful.

Revised.

> A new Sentinel-1 image-based sea ice classification algorithm using a machine learning-based model trained in a semi-automated manner is proposed to support  daily ice charting.

3) page 1 - line 12 - 'In most of the previous works...'... please provide a few references in this sentence to the works you have in mind.

References added.

> In  many of the previous works on ice-water and/or sea ice classification (Soh and Tsatsoulis, 1999; Zakhvatkina et al., 2013; Leigh et al., 2014; Liu et al., 2015; Ressel et al., 2015; Zakhvatkina et al., 2017; Aldenhoff et al., 2018), the training and validation were done using manually produced ice maps.

4) page 1 - lines 12-15 and lines 20. I reiterate my earlier point. Ice charts are generated manually by trained analysts. Although they are available in the public domain, and this means using these charts directly relieves the individual designing the classification algorithm from the 'laborious' and possibly biased process of manually choosing training and testing data, it does not enable an automated workflow.

Revised.

> A new Sentinel-1 image-based sea ice classification algorithm using a machine learning-based model trained in a semi-automated manner is proposed to support  daily ice charting. Previous studies mostly rely on manual work in selecting training and validation data. We show that the  readily available ice charts from  the operational ice services can reduce the number of manual works in preparation of large amounts of training/testing data. Furthermore, they reduce the inconsistent decisions in the classification algorithm by indirectly exploiting the best ability of the sea ice experts working at the operational ice services.

5) page 1 - line 20 - Again, ice charts are generated by humans. They contain human error. They are often produced under a strong time constraint, and in the interest of marine safety (the latter point meaning they likely contain bias to ensure safety).

Revised.

> The use of a public ice chart as training and validation reference data may help in solving the validation problem and enabling automation. The preparation of a public ice chart is also through manual inspection of various sources of satellite imagery and other sources of data (Partington et al., 2003; Johannessen et al., 2006); however, training using a large volume of these charts would reduce operator-to-operator bias. The overall bias may exist since the public ice charts are produced in the interest of marine safety. Nevertheless, as the human interpretation available in the ice chart is currently considered as the best available information of sea ice (Karvonen et al., 2015), the best practice to make a sea ice type classifier is to train with the public ice chart so that the best knowledge of certified ice analysts is mimicked.

6) page 3 - line 25 - 'ice edge determined from AMSR-E' - an ice edge cannot be determined from AMSR-E without using an algorithm. Which algorithm was used? Please revise.

Revised.

> Heinrichs et al. (2006) reported that the ice edge determined from  AMSR-E passive microwave radiometer data using the isoline of 15% concentration matches best the ice edge determined from RADARSAT-1 SAR data using visual inspection.

7) page 3 - lines 28-29 - I don't know what the authors mean by 'has a precision of decimals'.

Revised.

> Note that the ice concentration label in the SIGRID-3 format is assigned in an increment of 10%.

8) page 3 - line 26 - Similar comment regarding the ice edge determined from SAR - a methodology must have been used to get this ice edge. Was it visual inspection, or another method? Please revise.

Revised.

> Heinrichs et al. (2006) reported that the ice edge determined from  AMSR-E passive microwave radiometer data using the isoline of 15% concentration matches best the ice edge determined from RADARSAT-1 SAR data using visual inspection.

9) page 4 - line 3 - better than what?

Revised.

> Comparing the original SoD in the top left panel with the processed SoD in the bottom left panel, it is clear that the ice edge of the processed SoD match better with the SAR backscattering images.

10) page 5 - line 2 - wording is not specific - many of the previously developed methods - methods for what? These references are a mixture of ice/water and ice type classification studies. These two tasks are different from the perspective of a computer algorithm. Also, a reference to Shokr [1991] should be included.

Revised and added reference.

> Like many of the previously developed sea ice type classification methods (Shokr, 1991; Barber and LeDrew, 1991; Soh and Tsatsoulis, 1999; Deng and Clausi, 2005; Zakhvatkina et al., 2013; Leigh et al., 2014; Liu et al., 2015), the proposed approach starts from gray level co-occurrence matrices (GLCM) calculation.

11) page 5 - lines 9-11 - I don't understand this sentence, what is averaged for multiple distances, and what is the normalized GLCM?

Normalized GLCM is the GLCM divided by the sum of all elements, representing probability of co-occurrence. As there are multiple normalized GLCMs, one for each of the co-occurrence distances, the averaged values were used to reduce the dimensionality of the data to analyze.

12) page 5 - line 5 - direction? or should this be orientation?

Revised. It should be orientation in the context.

13) page 5 - line 12 - The term spatial resolution is not clear. Some authors consider this the scales that are resolved. It may be better to state that the spacing between the GLCM texture feature windows is 1km? (or please reword if I am not interpreting this point correctly).

Revised.

> In this study, we set $w$ as 25 so that the grid spacing of the result of texture analysis is 1 km.

14) page 5 - It would be nice to have the Haralick features listed in a table, and to provide a brief rationale for including all of them in the study. Information as to how long it took to calculate these features using 64 grey levels for their set of imagery is also important.

As the usefulness of GLCM-based texture features for sea ice classification has been demonstrated in literature (Shokr, 1991; Soh and Tsatsoulis, 1999; Deng and Clausi, 2005; Zakhvatkina et al., 2013; Leigh et al., 2014; Liu et al., 2015) and the Haralick features include most of them, it might be not necessary to list all the features in the manuscript. The computation time for extracting Haralick texture features per image is approximately five minutes in the given conditions (64 grey levels, 25 x 25 pixels of subwindow size) with an Intel i7 quad-core processor.

15) page 5 - the number of Haralick features is referred to inconsistently as 13 on line 7 and 26 on line 25. The 26 is likely just accounting for the two polarizations, but the two should be referred to in a consistent manner. Similarly on page 7 lines 22-23, please use either 'Haralick texture features' or 'texture features' consistently when describing the three classifiers.

Revised.

> In addition to  13 Haralick features, the coefficient of variation (CV) which is reported as a useful feature for ice-water discrimination (Keller et al., 2017) is included. The CV is defined as follows:
> …
> We trained three RF classifiers with different feature configurations: i) FC1:  Haralick texture features and CV, ii) FC2: Haralick texture features, CV, and  incidence angle, iii) FC3: Haralick texture features, CV, incidence angle, and day-of-the-year.

16) page 6 - line 16 'they are' - what are 'they'? Is this the number of operations?
Revised.

> For the SVM, the number of operations is $O(n^2 p + n^3)$ and $O(n_{sv}p)$ for training and prediction while for RF, $O(n^2 p n_{tr})$ and $O(n_{tr}p)$, respectively, where $n$ is the number of samples, $p$ is the number of features, $n_{sv}$ for the number of support vectors, and $n_{tr}$ for the number of trees.

17) page 7 - If a binary ice/water classifier is 'simple' (line 6), why are the authors starting with ice type classification? I suggest this be reworded.
Revised.

> Since even  a  ice/water classifier has not been well developed yet for Sentinel-1, the image selection procedure has to be done manually in the beginning.

18) page 8 - lines 20-21 - The sentence starting with, 'Since the training and test datasets were extracted from the same...' I find a little out of place. With this placement, it seems like it is trying to account for the results from FC2 and FC3. It might be better to start this one with 'When the evaluation is carried out with the 2018 data, the training and test datasets....'
Revised.

19) page 9 - line 32 - 'capturing' should be 'to capture'
Corrected.

20) page 9 - line 32 -'more details' - as compared to what?
Corrected.

21) Figure 3 - Could the authors provide some information in the text as to what the map of partial concentration is (top right). Is this the partial concentration of the dominant ice type for the given polygon?
Revised.

22) Figures 3,7,8 and 9 should have geolocation data provided.
Added geolocation grid to Figure 7 and Figure 8. In Figure 3 and Figure 9, the geolocation information may be irrelevant for understanding the contents.

23) all numbers less than ten should be written out in words, eg., 3 -> three
Corrected

References above are now in the reference list of the revised manuscript.

**Anonymous Referee #3**

First of all, we would like to thank the reviewer for the providing important comments. We hope you reconsider your decision based on our responses and modifications made to the manuscript. Please find below our answers (in green) and modifications (deleted in red and added in blue) to your comments/suggestions/questions.

**General comments**

The manuscript by Park et al presents Random Forest based classifier (Python Scikit-Learn) for sea ice type classification from dual pol (HH-HV) Sentinel-1 images which were collected during the winter period only. Training dataset were collected from the National Ice Center weekly ice chart and the classification algorithm exploits standard GLCM features along with some additional features. Since the launch of Sentinel-1 SAR sensors (a+b), it continuously monitoring Arctic Sea ice with high spatial and temporal resolution and an automated sea ice type classification product from high resolution SAR is highly desirable and relevant for the sea ice monitoring community. Having said that, there are several limiting factors which are preventing researchers to come up with a robust SAR based sea ice type classification scheme, (1) Backscatter variation due to varying incidence angle, along with sensor specific noise related issues (2) Backscatter variation due to seasonal changes (winter–melt- early summer - summer). In this manuscript authors performed a denoising technique which was developed by authors previously (Park et al., 2018), and a standard linear incidence angle correction (Section 2.2.3). It is important to note here that different sea ice types have different incidence angle dependency. Moreover, if the incidence angle correction was applied over all classes (i.e. including open water), it will most likely not contribute to the robustness of the classifier. Backscatter variation due to seasonal changes is completely ignored in the presented manuscript. As authors aimed to develop an operational system, in my opinion authors cannot ignore this major issue completely.

First of all, we would like to thank the reviewer for the providing important comments. Please find below our answers (in green) and modifications (deleted in red and added in blue) to your comments and suggestions.

- The use of Sentinel-1 in ice type classification is highly demanded for sea ice monitoring community but has not going well because of the two reasons that the reviewer pointed. We partly solved the problem of radar backscatter variation due to sensor noise issue, and this is one of the main significances of this study.

- The fact that different sea ice types have different incidence angle dependency is well known, but it is not true that applying the incidence angle correction over all classes will not contribute to the robustness of the classifier. Typically, the slope for open water is higher than that for sea ice, thus the correction works in a way reducing the backscatter variation for open water as well. Moreover, ice type-specific correction prior to ice type classification is controversial.

- As the reviewer pointed, backscatter variation due to seasonal changes is important for operational ice charting. Since the developed algorithm was tested for winter season only, we changed the title as "Classification of **Winter** Sea Ice Types in Sentinel-1 SAR images" and the limitation related to the variations in radar backscattering and its corresponding image textures due to seasonal changes is added to the end of Section 3 in the revised manuscript.

- The only variable that we introduced to capture the seasonality, day of year, might not correspond to the same temperature, fluxes and weather regimes. Weather variant input parameters may be more suitable. However, as sea ice drifts continuously, weather variant information for each of the ice floes in the SAR image at the image acquisition time needs to be calculated and joined, and this may require complex and rigorous works. In order to simplify the problem, we used day of year only, but certainly other parameters need to be tested in the follow up paper. We are very interested in combining sea ice drift with ice type-specific texture changes in our future study.

**Specific comments**

The proposed classification scheme is based on Python Scikit-Learn library and GLCM features, this kind of classification scheme is well known and published several times for different frequency bands. Therefore the current manuscript is very limited in terms of innovation. What I find slightly different is use of weekly ice chart for training data generation.

Our manuscript includes several significance and innovations:

i) This is the first study to examine the classification of Sentinel-1 SAR images for determination of sea ice types.

ii) This study demonstrates the ability to classify ice types from the noisy Sentinel-1 image by adopting our previous development, textural denoising method, and in the revised manuscript, we show how the denoising methods they have developed lead to improved ice type classification.

**[Section 3]**

To see how the denoising step in Section 2.2.2 led to improvements in the classification accuracies, the same training and evaluation were conducted for the same dataset without applying the textural noise correction (Table 4). In both FC1 and FC2, the accuracies improved for young ice (+8.2-9.8%) and first-year ice (+9.2-11.6%) which were most pronounced compared to those for open water (+1.7%) and old ice (+1.2-1.7%). On the contrary, a small accuracy decrease was observed for new ice (-2.8-4.7%). Nevertheless, the improvement in kappa (+0.05) demonstrates a clear improvement in the overall classification result.

Table 4: Changes in classification accuracies before and after applying textural denoising

| class | case | | | | | | | | |
|---|---|---|---|---|---|---|---|---|---|
| | FC1 | | | FC2 | | | FC3 | | |
| | Thermal denoising only | Textural denoising applied | difference | Thermal denoising only | Textural denoising applied | difference | Thermal denoising only | Textural denoising applied | difference |
| OW | 88.4 | 90.1 | +1.7 | 88.9 | 90.6 | +1.7 | 88.0 | 85.4 | -2.6 |
| NI | 30.2 | 28.0 | -2.8 | 27.7 | 23.0 | -4.7 | 31.8 | 23.9 | -7.9 |
| YI | 34.9 | 44.7 | +9.8 | 36.2 | 44.6 | +8.2 | 43.4 | 51.5 | +8.1 |
| FYI | 29.3 | 38.9 | +9.6 | 30.4 | 42.0 | +11.6 | 38.0 | 47.0 | +9.0 |
| OI | 91.5 | 92.7 | +1.2 | 90.3 | 91.7 | +1.4 | 75.2 | 66.3 | -8.9 |
| kappa | 0.62 | 0.67 | +0.05 | 0.62 | 0.67 | +0.05 | 0.54 | 0.49 | -0.05 |

iii) the use of public ice chart for training is new, and the difference from the conventional approaches were discussed to the introduction section.

**[Section 1]**

In  many of the previous works on ice-water and/or sea ice classification (Soh and Tsatsoulis, 1999; Zakhvatkina et al., 2013; Leigh et al., 2014; Liu et al., 2015; Ressel et al., 2015; Zakhvatkina et al., 2017; Aldenhoff et al., 2018), the training and validation were done using manually produced ice maps. Although the authors claimed that the manual ice maps were drawn by ice experts, the selection of SAR scenes and interpretation can be  inconsistent, and the number of samples  was not enough to generalize the results because of the laborious manual work. Therefore, increasing objectivity is crucial, and automating the classification process is encouraged. The idea of training using SAR images and accompanying image analysis charts, which is a raw interpretation of SAR images by trained ice analysts working at operational ice services, were tested for sea ice concentration estimation by Wang et al. (2017); however, such image analysis charts are not accessible to the public.

However I am also concerned about the automated training data generation as it is not clear which images were used to generate the ice chart by NIC and there is a high probability that that the ice chart polygons will not match that Sentinel-1 mosaics ice types. Hence there is a high risk that the classifiers were trained with wrong training data. Authors mentioned that 57 images were selected manually where only 'ice egdes' match well with

the ice chart. In my opinion, this is also a manual selection of training data which authors criticized in the introduction section.

We criticized conventional approach for two reasons: selecting training/testing data by anonymous human ice expert can be subjective, and preparing a large volume of such dataset is laborious. If public ice charts are used for training and validation, these two issues are partly resolved. Although the public ice chart is also made by human expert thus it cannot be completely free from subjective decisions, training using a large volume of these charts would reduce operator-to-operator bias. Another advantage of the use of public ice chart is that the best knowledge of certified ice analysts rather than anonymous expert is to be mimicked through the machine learning process. Regarding the manual selection of training data, the visual inspection along the ice edges overlaid on SAR image is also not automated and this is why the proposed method is "semi-" automated algorithm, but it requires much less effort and expertise compared to drawing manual ice chart from the SAR image itself.

> **[Section 2.2.6]**
> In order to automate image selection, the ice edges in SAR images need to be identified first.  Since even  an  ice/water classifier has not been well developed yet for Sentinel-1, the image selection procedure has to be done manually in the beginning. However, once a classifier is generated with high accuracy, it can be used to automate the procedure, then the whole process in the proposed scheme will be fully automated. This is why the proposed algorithm is named "semi-" automated for now. Nevertheless, the manual selection is done by visual inspection of ice-water boundaries overlaid on SAR images. The ice-water boundary can be extracted easily from the reprojected ice chart by selecting the pixel borders of open water class. Then the SAR backscattering image contrasts across the ice-water boundaries are examined both in HH- and HV-polarization because the image contrast between ice-water is larger in HV and smooth level ice is better recognizable in HH.

The selection (and definitions) of ice types for SAR based sea ice classification scheme is crucial. The 5 class classifier has some classes which might be very close to each other in terms of backscatter and texture. This might be the main reason for significantly low classification accuracy. I would recommend the authors to restrict the classifier for 4 classes (Open Water, Young Ice, FYI and old ice).

As the reviewer pointed, the five classes do not fit very well to the characteristic radar signatures from sea ice. We have tested for the four classes and the results are as below.

| | | Predicted | | | | | | | | | | |
|---|---|---|---|---|---|---|---|---|---|---|---|---|
| | | OW (open water) | | | NYI (New + Young ice) | | | FYI (First-year ice) | | | OI (old ice) | | |
| | Case | FC1 | FC2 | FC3 | FC1 | FC2 | FC3 | FC1 | FC2 | FC3 | FC1 | FC2 | FC3 |
| Actual | OW | **91.9** | **92.1** | **87.3** | 2.0 | 2.3 | 6.0 | 6.1 | 5.7 | 6.7 | 0.0 | 0.0 | 0.0 |
| | NYI | 11.2 | 10.2 | 10.3 | **47.7** | **49.1** | **54.8** | 32.7 | 34.2 | 30.2 | 8.4 | 6.6 | 4.6 |
| | FYI | 5.5 | 4.6 | 27.8 | 26.8 | 27.4 | 27.8 | **41.0** | **43.5** | **48.1** | 26.7 | 24.5 | 19.1 |
| | OI | 0.6 | 0.5 | 8.1 | 3.5 | 3.8 | 8.1 | 3.1 | 4.0 | 25.9 | **92.8** | **91.6** | **65.5** |

Compared to the results of five-class scheme in Table 3, the accuracy increases for merged young ice (new ice and young ice) were 3.0-4.5%. However, the misclassification among young ice and first-year ice was still significant.

A seasonal assessment of the classification scheme is missing. It the most important issue to address and without this assessment it would not be reasonable to claim the scheme to be either operational or innovative.

As the reviewer pointed, we did not conduct seasonal assessment. Since the developed algorithm was tested for winter season only, we changed the title as "Classification of Winter Sea Ice Types in Sentinel-1 SAR images" and the limitation related to the variations in radar backscattering and its corresponding image textures due to seasonal changes is added to the end of Section 3 in the revised manuscript.

Due to above mentioned major issues I didn't listed any technical corrections. I would kindly invite the authors to address this issues in future. Due to the lack of innovation and failed to address the basic issues, at current stage I can only recommend the manuscript to be rejected despite it is well within the scope of The Cryosphere.

We kindly ask the reviewer to reconsider his/her decision based on our responses and modifications made to the manuscript.

[revised manuscript text omitted]

26 December 2018      2 January 2019

| Ice free | Bergy water | New ice | Young ice | FYI thin | FYI medium | Old ice |

| Ice free | New ice | Young ice | FYI thin | Old ice |

[Figure]

**Figure 9: An example of  inconsistency of the ice charts. The SoDs from the NIC ice charts on different dates (26 December 2018 and 2 January 2019) are superimposed on the Sentinel-1 backscattering image of the corresponding dates. The  same ice floe (red outline) is  classified differently in each ice chart  (old ice on the left panel and first-year ice on the right panel) despite of the similarity in the SAR backscattering images.**

**Tables**

5  **Table 1:** **Hyperparameters** used for grid search

| Parameters | Values | | | | | | |
|---|---|---|---|---|---|---|---|
| $N_T$ | 1 | 2 | 4 | 8 | 16 | 32 | 64 |
| $D$ | 1 | 2 | 4 | 8 | 16 | 32 | 64 |
| $N_F$ | 1 | 2 | 4 | 8 | 16 | 28 | |

**Table 2: Confusion matrix  of the 5-class RF classifier which was trained with and applied to the 2018 dataset**

| | | Predicted | | | | | | | | | | | | | | |
|---|---|---|---|---|---|---|---|---|---|---|---|---|---|---|---|
| | | IF (ice free) | | | NI (new ice) | | | YI (young ice) | | | FYI (first-year ice) | | | OI (old ice) | | |
| | case | FC1 | FC2 | FC3 | FC1 | FC2 | FC3 | FC1 | FC2 | FC3 | FC1 | FC2 | FC3 | FC1 | FC2 | FC3 |
| Actual | IF | **93.7** | **94.6** | **95.6** | 1.8 | 1.4 | 0.9 | 0.4 | 0.4 | 0.4 | 4.1 | 3.7 | 3.1 | 0.0 | 0.0 | 0.0 |
| | NI | 20.4 | 19.1 | 18.7 | **32.5** | **33.8** | **58.3** | 31.4 | 31.4 | 14.6 | 13.3 | 12.8 | 5.9 | 2.5 | 2.9 | 2.6 |
| | YI | 2.0 | 2.0 | 1.9 | 4.5 | 3.9 | 6.9 | **60.5** | **59.1** | **61.3** | 26.5 | 29.4 | 25.2 | 6.5 | 5.7 | 4.6 |
| | FYI | 4.4 | 4.2 | 3.4 | 3.1 | 2.8 | 2.9 | 22.3 | 19.8 | 17.8 | **56.8** | **60.7** | **64.5** | 13.3 | 12.5 | 11.5 |
| | OI | 0.3 | 0.3 | 0.4 | 0.9 | 0.9 | 1.7 | 5.8 | 5.3 | 3.6 | 7.9 | 7.6 | 6.2 | **85.1** | **85.9** | **88.1** |
| | | Predicted | | | | | | | | | | | | | | |
| | | OW (open water) | | | NI (new ice) | | | YI (young ice) | | | FYI (first-year ice) | | | OI (old ice) | | |
| | case | FC1 | FC2 | FC3 | FC1 | FC2 | FC3 | FC1 | FC2 | FC3 | FC1 | FC2 | FC3 | FC1 | FC2 | FC3 |
| Actual | OW | **94.5** | **95.2** | **96.7** | 1.4 | 1.1 | 0.6 | 0.3 | 0.3 | 0.3 | 3.7 | 3.4 | 2.4 | 0.0 | 0.0 | 0.0 |
| | NI | 19.3 | 17.3 | 14.8 | **33.1** | **38.9** | **68.7** | 33.3 | 31.1 | 9.3 | 12.1 | 10.6 | 5.5 | 2.2 | 2.1 | 1.7 |
| | YI | 1.9 | 1.8 | 1.6 | 3.8 | 3.6 | 6.7 | **62.3** | **62.8** | **64.5** | 26.1 | 27.5 | 23.8 | 5.9 | 4.3 | 3.4 |
| | FYI | 4.2 | 3.6 | 2.4 | 2.6 | 2.6 | 2.1 | 21.7 | 20.6 | 15.6 | **58.1** | **61.1** | **69.8** | 13.4 | 12.1 | 10.1 |
| | OI | 0.3 | 0.3 | 0.4 | 0.6 | 0.8 | 1.4 | 5.8 | 5.0 | 3.0 | 7.3 | 7.3 | 4.1 | **86.0** | **86.7** | **91.2** |

10  **Table 3: Confusion matrix  of the 5-class RF classifier which was trained  with 2018 dataset and applied to the 2019 dataset**

| | | Predicted |
|---|---|---|
| | | |

| | | IF (ice free) | | | NI (new ice) | | | YI (young ice) | | | FYI (first-year ice) | | | OI (old ice) | | |
|---|---|---|---|---|---|---|---|---|---|---|---|---|---|---|---|---|
| | case | FC1 | FC2 | FC3 | FC1 | FC2 | FC3 | FC1 | FC2 | FC3 | FC1 | FC2 | FC3 | FC1 | FC2 | FC3 |
| Actual | IF | 89.6 | 90.0 | 87.3 | 3.5 | 3.5 | 5.1 | 1.1 | 1.1 | 1.7 | 5.7 | 5.4 | 5.9 | 0.0 | 0.0 | 0.0 |
| | NI | 19.9 | 22.9 | 31.2 | 29.1 | 25.3 | 20.2 | 40.3 | 40.7 | 41.1 | 8.0 | 8.5 | 5.7 | 2.8 | 2.7 | 1.8 |
| | YI | 7.3 | 7.2 | 6.9 | 4.1 | 3.6 | 2.3 | 42.7 | 41.4 | 50.4 | 36.2 | 38.9 | 33.6 | 9.7 | 8.8 | 6.8 |
| | FYI | 6.2 | 5.8 | 5.9 | 4.3 | 3.9 | 2.0 | 24.4 | 23.3 | 25.1 | 39.2 | 41.9 | 47.6 | 25.9 | 25.0 | 19.4 |
| | OI | 0.6 | 0.6 | 0.6 | 1.4 | 1.3 | 0.5 | 2.4 | 2.7 | 7.1 | 3.1 | 3.5 | 21.1 | 92.5 | 92.0 | 70.8 |

Predicted

| | | OW (open water) | | | NI (new ice) | | | YI (young ice) | | | FYI (first-year ice) | | | OI (old ice) | | |
|---|---|---|---|---|---|---|---|---|---|---|---|---|---|---|---|---|
| | case | FC1 | FC2 | FC3 | FC1 | FC2 | FC3 | FC1 | FC2 | FC3 | FC1 | FC2 | FC3 | FC1 | FC2 | FC3 |
| Actual | OW | 90.1 | 90.6 | 85.4 | 3.1 | 2.7 | 5.7 | 1.0 | 1.1 | 2.0 | 5.8 | 5.7 | 6.9 | 0.0 | 0.0 | 0.0 |
| | NI | 20.1 | 24.5 | 28.3 | 28.0 | 23.0 | 23.9 | 42.0 | 42.4 | 40.9 | 7.6 | 7.9 | 5.5 | 2.4 | 2.1 | 1.4 |
| | YI | 6.7 | 6.1 | 6.3 | 3.3 | 3.4 | 3.1 | 44.7 | 44.6 | 51.5 | 36.0 | 38.2 | 33.5 | 9.3 | 7.7 | 5.7 |
| | FYI | 5.4 | 4.4 | 4.9 | 3.6 | 3.8 | 2.7 | 25.8 | 25.3 | 27.5 | 38.9 | 42.0 | 46.0 | 26.3 | 24.5 | 18.9 |
| | OI | 0.5 | 0.5 | 0.5 | 1.3 | 1.2 | 0.7 | 2.7 | 3.0 | 7.7 | 2.8 | 3.6 | 24.9 | 92.7 | 91.7 | 66.3 |

**Table 4: Classification accuracies before and after applying textural denoising**

| class | case | | | | | | | | |
|---|---|---|---|---|---|---|---|---|---|
| | FC1 | | | FC2 | | | FC3 | | |
| | Thermal denoising only | Textural denoising applied | difference | Thermal denoising only | Textural denoising applied | difference | Thermal denoising only | Textural denoising applied | difference |
| OW | 88.4 | 90.1 | +1.7 | 88.9 | 90.6 | +1.7 | 88.0 | 85.4 | -2.6 |
| NI | 30.2 | 28.0 | -2.8 | 27.7 | 23.0 | -4.7 | 31.8 | 23.9 | -7.9 |
| YI | 34.9 | 44.7 | +9.8 | 36.2 | 44.6 | +8.2 | 43.4 | 51.5 | +8.1 |
| FYI | 29.3 | 38.9 | +9.6 | 30.4 | 42.0 | +11.6 | 38.0 | 47.0 | +9.0 |
| OI | 91.5 | 92.7 | +1.2 | 90.3 | 91.7 | +1.4 | 75.2 | 66.3 | -8.9 |
| kappa | 0.62 | 0.67 | +0.05 | 0.62 | 0.67 | +0.05 | 0.54 | 0.49 | -0.05 |

**Table 6: Confusion matrix  of the 3-class RF classifier which were trained  and applied to the 2018 dataset**

| Actual | | Predicted | | | | | | | | |
|---|---|---|---|---|---|---|---|---|---|---|
| | | IF (ice free) | | | FYI (first-year ice) | | | OI (old ice) | | |
| | Case | FC1 | FC2 | FC3 | FC1 | FC2 | FC3 | FC1 | FC2 | FC3 |
| | IF | 96.5 | 96.7 | 96.9 | 3.5 | 3.3 | 3.1 | 0.0 | 0.0 | 0.0 |
| | MFYI | 5.8 | 5.7 | 4.8 | 84.5 | 85.8 | 87.2 | 9.8 | 8.6 | 7.9 |
| | OI | 0.5 | 0.5 | 0.5 | 14.4 | 13.9 | 12.4 | 85.1 | 85.6 | 87.1 |
| | | Predicted | | | | | | | | |
| | | OW (open water) | | | mFYI (mixed FYI) | | | OI (old ice) | | |
| | Case | FC1 | FC2 | FC3 | FC1 | FC2 | FC3 | FC1 | FC2 | FC3 |
| | OW | 96.7 | 97.3 | 99.1 | 3.3 | 2.6 | 0.9 | 0.0 | 0.0 | 0.0 |
| | mFYI | 5.2 | 4.7 | 2.5 | 85.8 | 87.6 | 92.3 | 9.0 | 7.7 | 5.2 |
| | OI | 0.4 | 0.4 | 0.2 | 13.2 | 12.4 | 6.0 | 86.4 | 87.2 | 93.8 |

10   **Table 6: Confusion matrix  of the 3-class RF classifier which was trained  with the 2018 dataset and applied to the 2019 dataset**

| Actual | | Predicted | | | | | | | | |
|---|---|---|---|---|---|---|---|---|---|---|
| | | IF (ice free) | | | FYI (first-year ice) | | | OI (old ice) | | |
| | Case | FC1 | FC2 | FC3 | FC1 | FC2 | FC3 | FC1 | FC2 | FC3 |
| | IF | 93.4 | 93.4 | 91.9 | 6.5 | 6.6 | 8.1 | 0.0 | 0.0 | 0.0 |
| | MFYI | 9.8 | 9.2 | 8.9 | 71.0 | 72.9 | 75.3 | 19.2 | 17.9 | 15.8 |
| | OI | 0.7 | 0.7 | 0.6 | 6.8 | 7.7 | 18.5 | 92.5 | 91.6 | 81.0 |
| | | Predicted | | | | | | | | |
| | | OW (open water) | | | mFYI (mixed FYI) | | | OI (old ice) | | |
| | Case | FC1 | FC2 | FC3 | FC1 | FC2 | FC3 | FC1 | FC2 | FC3 |
| | OW | 93.6 | 93.6 | 86.3 | 6.4 | 6.4 | 13.6 | 0.0 | 0.0 | 0.0 |
| | mFYI | 8.8 | 7.5 | 7.1 | 72.4 | 75.3 | 81.6 | 18.8 | 17.2 | 11.3 |
| | OI | 0.6 | 0.6 | 0.4 | 6.7 | 8.1 | 39.8 | 92.7 | 91.4 | 59.7 |

---

## Referee Report (RR1)

The manuscript has been improved as compared to the previous version. However, there are still points to which I would like to hear from the authors:

1. The authors do not give guidance as to the accuracy of the NIC chart ice type information. It is difficult to assess the quality of the classifier output when it is trained with, and evaluated against, only NIC charts. The accuracy could be investigated by comparing NIC charts with Canadian ice service charts for regions/dates on which both are available. Alternatively, the output from the classifier (ice type) should be compared against another data source (eg. OSI-403-c).

2. The contribution of the study still needs to be clarified. I don't feel that the main technological innovations claimed in the conclusions have been demonstrated. There is no clear demonstration that the time in running the semi-automated method is much less than the manual approach (in particular given that there is a manual step in the method, and given the tools available to pull samples from images that could be used in a study that is manually generating samples), nor is it clear to me how the evaluation of the classifier is more objective (more objective as compared to what? another method? or independent data? - see above comment).

3. This study does demonstrate several important problems that need to be resolved when using ice type from operational charts to train a classifier. For example, the labels from NIC have accuracy issues (or inconsistency issues as stated in the conclusions), there seem to be possible issues with the incidence angle correction etc. How do the authors propose moving forward? For example, if the authors are really proposing a higher resolution three-class classification, how are they first planning to overcome the problems observed in the present study? A thoughtful discussion of these issues would be a nice contribution.

Specific comments:

- page 2: line 18, What evidence is there that previous studies were less generalizable than the current one? This statement should be backed up.

- page 2: line 22, Public ice charts are generated manually. I don't see how using them to train a classifier enables automation.

- page 3: line 25: objective identification → I assume what the authors are referring to is the way that a large quantity of training labels are pulled from the charts objectively? as compared to manually. Please make this more clear.

- page 7: line 33: The method is semi-automated in part because of the way the samples are generated, but also because of the use of manually generated ice charts as training data.

- page 11: lines 5-6: 'This is because ay of year might not correspond to the same temperature...' - Might be better stated 'because SAR image features, which partially reflect temperature fluxes and weather regimes, might not correspond to day of year.

- page 11: line 8: Please refer to the figure number.

- Figure 8: there are clearly some problems regarding wind roughened water (misclassifications in Fig 8) - The authors acknowledge this, but don't suggest a future path to resolve this issue. There is also some water identified in what might be ice cover in this same figure.

English language (not an exhaustive list):

- abstract: line 18, dataset → datasets

- page 1: line 7, is impacting seriously → can seriously impact

- page 2: line12, statistics thus → statistics, and thus

- page 2: line 14: works → studies

- page 3: line 9: images from the first year is → images from the first year are

- page 3: line10: is used for validation → are used for validation

- page 3: line 22: each steps → each step

- page 6: each polarization images → each polarization image

- page 6: 'In the literatures about sea ice classification' → Previous studies about sea ice classification

- page 8: line4, 'better recognizable' → 'more easily identified'

- page 8: line 11 'raters' → rasters

- page 8: line 11, 'trained classifier' → output from the trained classifier

- page 9: line 32, Do not start a sentence with a Greek letter (please correct at other locations in the manuscript).

- page 10: line 31, 'ices' → ice

- page 11: line 11: ANSWRS (use of acronym).

---

## Referee Report (RR2)

The authors have addressed my previous comments. Further comments given below:

1. Although the authors did carry out a comparison between the NIC charts and the OSI-SAF charts, their results do not go into depth and are inconclusive. For example, it is stated that the accuracy of open water decreases because of the different ice concentration thresholds used for the OSI-SAF charts and for their classification method, but this is not shown. For example, it could be shown by retraining their classifier to use a different threshold, or by systematically comparing the misclassification rates for points in the NIC charts with ice concentration between 20% and 30% or 40% with misclassification rates for points with higher ice concentrations. On a similar note, for the high misclassification rate of first year ice, only one date is shown to demonstrate that the ice classes are different between the NIC charts and the OSI-SAF charts. It would be good to know how widespread these differences are.

2. page 10 lines 10-14 - here the authors compare their scores for the various classifier configurations for DS1 and DS2 (Tables 6 and 7). The fact that FC3 is overfitting for DS1, means that the accuracies for mFYI are very low (disregard the column for FC3 for Table 6), and there is considerable mFYI that is misclassified as old ice. It is not clear at all as to whether this problem (eg. 15% for mFYI for DS1) is due to the classifier itself or the data. Similar to the previous comment, it would be ideal to have a more thorough analysis of the ice charts themselves to unravel this problem, although I agree this could be left to a later study.

3. page 2 lines 25-24 - I believe the image analyses used in (Wang 2017) and those used in the present study (NIC) charts are similar in that both are manual analyses. The image analyses used in (Wang 2017) are each a direct manual interpretation of a SAR image by a trained ice analyst. They do represent analyses by 'certified ice analysts' (bottom of page 2). These image analyses are visually similar to the SAR data, as expected, and this is in agreement with the steps carried out in the submitted manuscript, where the authors visually check the agreement between the SAR imagery and the NIC charts. I think the only two differences are 1) the image analyses used in (Wang 2017) are not open to the public, and 2) they are an interpretation of the SAR image only. It's not clear from the manuscript what data sources the NIC charts are based on (though I may have missed it), and this information should be included.

4. I don't find the present manuscript demonstrates 'minimized manual work in the preparation of training and validation reference data' as the NIC charts are manual analyses, and the method itself also uses a manual step to select the training data.

- page 1 line 12 works should be work

- page 1 line 12 'reduce inconsistent decisions' - this was not shown. Please remove.

- page 2 line 20 and line 25 - what is the difference between a 'manual' interpretation and a 'raw' interpretation.

---

## Author Response (AR2)

**General comments**

I would like to thank the authors for addressing the comments given on the first round of revisions. There are a few more comments that I would like the authors to address before publication. Below are the original comments, authors responses start with * and new comments with _.

First of all, we would like to thank the reviewer for the positive evaluation and providing important comments. We hope that all your concerns will be cleared after reading our responses and modifications made to the manuscript. Please find below our answers (in green) and modifications (deleted in red and added in blue) to your comments/suggestions/questions.

**Specific comments**

Mahmud et al 2018 showed that different sea ice types have different incidence angle dependent slopes. Have you considered using a sea ice type dependent slope factor? Moreover, how does the work by Mahmud et al 2018 fit in with the incidence angle dependencies presented here?
*Typically, the slope for open water is higher than that for sea ice, thus the correction works in a way reducing the difference in sigma naught for open water as well. As the review pointed, different sea ice types have different incidence angle dependent slopes; however again, ice type-specific correction prior to ice type classification is controversial. Although estimating ice type dependent slope is not a part of this manuscript, we provide the values derived from the training/validation dataset for the review purpose only.
_Studying the figures provided in the review, the HV sigma naught values for the different ice types and open water used in the training/validation data, it is striking that there is hardly any variation across incidence angle, but also that there is limited difference in backscatter values between the different types of surface. The values in the HV- data presented appear to be very low, especially considering that there is a difference in the backscatter values, e.g. for the open water, presented in Figure 7 and 8. As a side note would a scalebar in dB for Figure 7 and 8 be beneficial. Moreover, the values are close to the nominal noise floor provided with the meta data in the Sentinel-1 images. Is the lack of trend an effect of the thermal noise correction applied to the images?
_Work by e.g. Isleifson et al 2010 found slopes for new ice and all 4 channels given a range of incidence angles and sufficient SNR. Work by Gill et al 2015 also found slopes of -0.25 for the HV-channel and Scharien et al, 2014 also found slopes for the HV-channel. How does this work relate to those studies?

We had made a mistake in estimating the incidence angle dependence in our previous results. After subtracting the thermal noise power, we had added the mean noise power back to the denoised image in order to prevent the denoised sigma naught in linear scale from falling down into negative values. As adding offset in linear scale is equivalent to scaling in log scale (decibel), the incidence angle dependence estimated in log scale had been corrupted. After revising this problem, we got new results summarized in the figure below.

[Figure]

The averaged incidence angle dependences of mixed ice types (squares) in mid-winter season (Jan-Mar) are -0.21 and -0.06 for HH and HV polarizations, respectively. These values are lower than those in our previous results, which were -0.200 and -0.025 for HH and HV polarizations, respectively. The values of new and young ice (triangles) are much lower especially when the season is close to summer. The large slope of -0.18 for HV polarization in May is comparable to the findings in Gill et al. (2015), which is -0.25 estimated from smooth first-year ice in May. Based on the reprocessed results, we changed Figure 4 and added explanations in revised manuscript.

**[page 4, line 31-32] and [page 5, line 1-3]**

of the satellite orbit and image acquisition geometry. By following the methods developed in Park et al. (2018, 2019), each of the Sentinel-1 images was denoised before further processes are applied. As the noise power subtraction yields negative intensity values where the backscattering power is close to the noise floor, especially in HV polarization, we added mean of the noise power back to the denoised result so that those pixels do not turn into NaN (not a number) by the sigma naught conversion of linear scale to log scale (decibel).

**[page 5, lines 15-31]**

15  Figure 4 shows incidence angle dependence in the SAR backscattering intensity for mixed sea ice types. From the Sentinel-1 dataset described in Section 2.1, sea ice pixels were extracted by using daily global sea ice edge products available from the EUMETSAT Ocean and Sea Ice Satellite Application Facilities (OSI SAF). For mid-winter season (Jan-Mar displayed by blue

20 background),  the estimated mean slope in HH polarization was  -0.21 dB/degree, which is slightly different from the estimation of the first-year ice (-0.24 dB/degree) in Mäkynen and Karvonen (2017) and in between the estimations for first-year ice (-0.22 dB/degree) and multi-year ice (-0.16 dB/degree) in Mahmud et al. (2018). For HV polarization, the estimated slope was only  -0.06 dB/degree, which is much lower than  -0.16 dB/degree for deformed first-year ice in Mäkynen and Karvonen (2017), however, it is in line with the estimations

25  in Liu et al., 2015. Work by Leigh et al. (2014) stated that the HV polarization backscatter signatures are largely unaffected by incidence angle variation in their RADARSAT-2 dataset. For summer season (Jun-Aug displayed by red background), the mean slopes increased to -0.28 and -0.08 dB/degree in HH and HV polarization, respectively. Scharien et al. (2014) reported significant slopes for ice adjacent to melt ponds in June, and Gill et al. (2015) also found slopes of -0.33 and -0.25 for smooth first-year ice in May in HH and HV polarization, respectively. The smaller slopes in our

30 estimation are likely due to the mixed ice types and structures; the SAR backscattering of deformed ice has lower incidence angle dependency as shown in Mäkynen and Karvonen (2017).
* * *
**[Figure 4] – replaced**

[Figure]

Figure 4: Incidence angle dependences of sigma naught in HH (closed squares) and HV (open squares) polarization channels. Pixels covering various types of sea ice were merged so that the averaged property can be estimated. The blue and red zones indicate winter and summer seasons, respectively.
* * *
How do you define "good match" (P7 R3)? Temporal overlap? Spatial overlap? How was this manual selection of images performed? How were the open water vs sea ice charts, that are used as initial input into the classifier, derived?

*Both temporal and spatial overlaps are important. Since the SAR image itself is potentially one of the sources for ice charting at the ice services, some images spatially match well with the shape of the polygons in ice chart. Temporal window length of 3 days from the publication date of ice chart was used for squeezing the number of images to make decisions of use/discard for training.

_What is a good match? Please state this clearly in the manuscript. The sentence now simply states that spatially and temporally good matches were used. How large is the spatial overlap? What is meant with squeezing the number of images?

As no scene identifiers or time window information of the satellite image products used for ice charting are annotated in the ice chart, is it crucial to identify SAR image – ice chart pairs sharing the same time instance or ice conditions. The good match in this study is the visual fitness of the ice-water boundaries. "Temporal window length of 3 days from the publication date of ice chart" is the time window of source information publication date annotated in the XML data comes with ice chart. As the time window does not cover the entire 7 days in week, only the images acquired from 3 days prior to the date of ice chart publication need to be examined, and this is what we meant with "squeezing the number of images". We added explanations in revised manuscript.

> **[page 8, lines 9-26]**
>
> To train an ice type classifier, a set of collocated SAR images and ice charts is required. After the preprocessing of the ice
> 10  chart including reprojection into the SAR image geometry, only the samples with spatially and temporally good matches should
> be fed to the training phase. The goodness of matching should be examined as the weekly ice chart is produced by merging
> information from many image sources acquired in different time instances, hence the ice locations and conditions are unlikely
> match to those in every SAR image. As no explicit scene identifier or time information of the images used in ice charting is
> provided with the ice chart itself, the basic strategy in image selection is to find a pair of SAR image and ice chart which match
> 15  well visually. Such an iImage selection is trivial, but not easy to automate. Since the weekly ice chart is made partly based on
> the SAR images acquired in the past three days from the date of publication, the ice edges in some images match well with
> those in the ice chart.
> In order to automate image selection, the ice edges in SAR images need to be identified first. Since even an ice/water classifier
> has not been well developed yet for Sentinel-1, the image selection procedure has to be done manually in the beginning.
> 20  However, once a classifier is generated with high accuracy, it can be used to automate the procedure, then the whole process
> in the proposed scheme will be fully automated. This is why the proposed algorithm is named "semi-" automated for now.
> Nevertheless, the manual selection to guarantee a "good match" is done by visual inspection of ice-water boundaries overlaid
> on SAR images. The ice-water boundary can be extracted easily from the reprojected ice chart by selecting the pixel borders
> of open water class. Then the SAR backscattering image contrasts across the ice-water boundaries are examined both in HH-
> 25  and HV-polarization because the image contrast between ice-water is larger in HV and while smooth level ice is better
> recognizablemore easily identified in HH.

Figure 8. In Figure 3 and 7 the ice chart from 3 days later is used yet in Figure 8 the image from the same day is used. Please be consistent in which time interval is used for these weekly ice charts.
*Since the reference ice chart is published weekly, the same NIC ice chart both in Figure 7 and 8 is supposed to be valid for both dates of Figure 7 and 8, but it is not as shown. The common time interval of 3 days in Figure 3 and 7 is just a coincidence. It can be one day or two days depending on the date of SAR image that used as source materials for ice charting at ice services. If the satellite images acquired two days prior to the publication of weekly ice chart, then the overall distribution of ice would represent the status of that time. Note that among the 57 images used for training, 42 and 35 percent of the images were acquired 2 and 3 days prior to the publication date of the corresponding weekly ice chart, respectively.
_For the validation data in 2018 and 2019 how many of those images were acquired at 2 and 3 days prior to the ice chart used to validate them?

The distribution of the number of images in the extended dataset of the revised manuscript is added to the revised manuscript.

> **[page 9, lines 3-7]**
>
> phase. Among  4485 images in total, we selected  840 images (419 for winter season and 421 for summer season) of which ice edges match well with the collocated ice chart. From the selected images, 120 million samples covering open water and sea ice were divided into training and test dataset. The DS2 was used to evaluate the performance of the trained classifier using temporally independent dataset of 513 images (281 for winter season and 232 for summer season). The distribution of the image acquisition dates prior to the publication of the reference ice chart is shown in Table 2.

> **[Table 2]**
>
> **Table 2: Distribution of the image acquisition dates prior to the publication of the reference ice chart**
>
> | | Training and test dataset (DS1) | | | | Validation dataset (DS2) | | | |
> |---|---|---|---|---|---|---|---|---|
> | Days prior to the date of ice chart publication | 3 | 2 | 1 | 0 | 3 | 2 | 1 | 0 |
> | Winter | 124 | 168 | 77 | 50 | 78 | 75 | 67 | 61 |
> | Summer | 119 | 125 | 112 | 65 | 87 | 67 | 48 | 30 |

_The new ice areas likely change at short time scales. What is the impact of having 42% (2-days) and 35% (3-days) time separation between the image acquisition and the sea ice chart? Is there a consistent time difference between the 2018 and 2019 data with respect to the Sentinel-1 images and the sea ice charts? Comparing the accuracy of the sea ice classifications with the temporal difference might provide some answers here.

As in the table above, the is no consistent time difference between DS1 and DS2.

**Technical comments**

In many places references to the appropriate work is missing, e.g. P1, R27, P4 R15-16 and P6 R9-10. Please carefully revise the manuscript to include references to earlier work.
*Corrected.
_The P6 R9-10 still has no reference to previous work.
Corrected.

P6 R10-12 you argue that when training dataset are prepared manually the sample size is usually less then 20 images. Please provide several references to support this statement.

_In Liu et al 2015 only one image is used for the training data please revise the manuscript accordingly (P6 R29). Another image is used for the validation of their study but only one for the training data.
Corrected.

_The reference Leigh et al 2014 does not provide slopes for either the HH- or the HV-channel in their study. Please revise the manuscript accordingly.
Corrected.

In Figure 9 the sea ice type FYI thin is include in the sea ice classification results. Please clarify what thin FYI means. Why is this class not used throughout?
*The label was wrong. It was just FYI without "thin". As the bottom panels (SAR results) are irrelevant in the context of the corresponding description, they were removed in the revised manuscript.
_In Figure 9 the label thin FYI is still present.
This is because Figure 9 displays the original SoDs in the NIC ice chart, not the simplified SoDs of the proposed method. We added an explanation to the caption of Figure (now Figure 11) in the revised manuscript.

[Figure]

**[Figure 9]**

Figure 11: An example of the inconsistency of the ice charts. Note that the SoD labels and colors are of original NIC ice chart, while those in Figure 7 and 8 are of simplified version as described in Section 2.2.1.The SoDs from the NIC ice charts on different dates (26 December 2018 and 2 January 2019) are superimposed on the Sentinel-1 backscattering image of the corresponding dates. The same ice floe (red outline) is classified differently in each ice chart (old ice on the left panel and first-year ice on the right panel) despite of the similarity in the SAR backscattering images.

_Throughout the manuscript first year ice is not consistently named: e.g. Figure 3 FYI (top left) FY ice (bottom left), Figure 7 and 8 First-year ice. Page 3, Row 9. First year. Moreover, FYI is not defined within the manuscript.
Corrected.

_P1 R11. Number -> amount
Corrected.

_P1. R20 What is operational manner?
Corrected.

_P2. R1-3. Reference to "HH and HV widely used for ice edge detection etc" is missing.
Corrected.

_P2. R29-30. Please specify type of bias.
Corrected.

_P4. R.23 "…noise, some which originate…"
Corrected.

_P8 R13. Clarify easily or remove it. "…borders of the open water …"
Corrected.

_P8 R13-15. Unclear sentence please revise.
Corrected.

_P10 R18. For FC1 and FC2 there is an improvement, however for FC3 there is a -0.05 reduction in the Kappa values. Please revise the sentence to account for this.
Corrected.

_P11 R20. "different marginal sides". What does this mean?
Corrected. "different edges of the range swath"

_P11 R21. Which figure is this image boundary visible in?
Corrected.

_P11 R22. What does extreme marine conditions mean? Please clarify.
Corrected.

_P11 R23-25. Are high wind conditions not accounted for/included in your training data?
An explanation regarding this misclassification was added to the revised manuscript.

> **[page 12, lines 24-29]**
>
> 8, there is a misclassified  first-year ice patch (yellow) in the open water area. According to the high resolution sea surface
>
> 25     winds data from SAR on the Sentinel-1 satellites (https://data.nodc.noaa.gov/cgi-bin/iso?id=gov.noaa.nodc:SAR-WINDS-S1), the wind speed ranged from 17 to 21 m/s at the time of image acquisition
>
> heavily roughing the water surface. Although we have included images with both high and low wind conditions in our training
>
> data, the image textures of wind roughened water surface and ice were confused in some cases, and the same happened in the
>
> image textures of calm water surface and smooth level ice.

**Anonymous Referee #2**

The manuscript has been improved as compared to the previous version. However, there are still points to which I would like to hear from the authors:

First of all, we would like to thank the reviewer for the positive evaluation and providing important comments. We hope that all your concerns will be cleared after reading our responses and modifications made to the manuscript. Please find below our answers (in green) and modifications (deleted in red and added in blue) to your comments/suggestions/questions.

1. The authors do not give guidance as to the accuracy of the NIC chart ice type information. It is difficult to assess the quality of the classifier output when it is trained with, and evaluated against, only NIC charts. The accuracy could be investigated by comparing NIC charts with Canadian ice service charts for regions/dates on which both are available. Alternatively, the output from the classifier (ice type) should be compared against another data source (eg. OSI-403-c).

   By adopting the reviewer's suggestion, we compared the output of our three-class classifiers against the OSI SAF sea ice type product, OSI-403-c, and the corresponding results and discussions were included in the revised manuscript.
* * *
**[page 11, lines 32-35] and [page 12, lines 1-14]**

Unfortunately, we could not find an official report regarding the accuracy of the NIC ice chart information. It might be not enough to assess the quality of the classifier output when it is trained with, and evaluated against, only NIC ice charts. The accuracy could be indirectly investigated by comparing the output from our classifier against another data source, such as OSI SAF sea ice type product (OSI-403-c). The ice classes of OSI-403-c are assigned from atmospherically corrected brightness temperatures of passive microwave radiometers (SSMIS and AMSR2) and backscatter values of radar scatterometer (ASCAT), using a Bayesian approach (Aaboe et al., 2018). Table 10 shows the confusion matrices for our three-class classifiers when their prediction results are compared with the OSI-403-c product as reference. Comparing with the results in Table 9, the

5  accuracies for open water decreased from by 6%; however, this is mainly because the ice concentration threshold for ice-water discrimination in OSI-403-c is 35% which is higher than 20% that we set in our preprocessing of NIC ice chart (Section 2.2.1), thus areas with low ice concentration in marginal ice zone are most likely annotated as open water in OSI-403-c. For first-year ice, large portions (72%) are misclassified as old ice. This might be partly explained from the Figure 12, which shows the ice classes in NIC ice chart and OSI-403-c for the same publication date. A large extent of old ice in NIC ice chart is annotated as

10  multi-year ice in OSI-403-c. As our classifiers were trained with NIC ice chart, it is natural to result in more old ice for the area where the ice type is classified as first-year ice in OSI-403-c. For old ice, the accuracy was the highest, 98%. Finding the reason for the clear discrepancy of the extent of first-year ice between the NIC ice chart and OSI-403-c is beyond the scope of this study, however, it should be noted that an elaborate future work for cross calibrating ice types in different ice charts are necessary.

**[Table 10]**

Table 10: Confusion matrix of the three-class RF classifier which was trained with DS1 winter dataset and applied to the DS2 winter dataset with reference to OSI SAF sea ice type product (OSI-403-c)

| | | Predicted (classifier was trained with NIC ice chart) | | | | | | | | |
| --- | --- | --- | --- | --- | --- | --- | --- | --- | --- | --- |
| | | Open water | | | Mixed first-year ice | | | Old ice | | |
| | Case | FC1 | FC2 | FC3 | FC1 | FC2 | FC3 | FC1 | FC2 | FC3 |
| Reference (OSI SAF) | Open water | **85.9** | **86.1** | **86.2** | 12.6 | 12.4 | 12.1 | 15.7 | 15.3 | 16.2 |
| | First-year ice | 1.9 | 1.6 | 2.0 | **26.0** | **26.8** | **26.9** | 72.1 | 71.6 | 71.2 |
| | Multi-year ice | 0.1 | 0.1 | 0.1 | 1.5 | 1.4 | 1.4 | **98.4** | **98.5** | **98.5** |

**[Figure 11]**

[Figure]

Figure 12: The ice types in NIC ice chart (left) and OSI SAF sea ice type product (right) for the same date (8 Jan. 2019). Note that the SoD labels and colors follows those defined in each ice chart format.

2. The contribution of the study still needs to be clarified. I don't feel that the main technological innovations claimed in the conclusions have been demonstrated. There is no clear demonstration that the time in running the semi-automated method is much less than the manual approach (in particular given that there is a manual step in the method, and given the tools available to pull samples from images that could be used in a study that is manually generating samples), nor is it clear to me how the evaluation of the classifier is more objective (more objective as compared to what? another method? or independent data? see above comment).

Additional experimental results were added to the manuscript and the conclusions were revised.

[page 12, line 30-32] and [page 13, line 1-18]

30  **4 Conclusion**

[revised manuscript text omitted]

**Specific comments**

- page 2: line 18, What evidence is there that previous studies were less generalizable than the current one? This statement should be backed up.
  Corrected.

- page 2: line 22, Public ice charts are generated manually. I don't see how using them to train a classifier enables automation.
  The review is right. "enabling automation" was deleted.

- page 3: line 25: objective identification → I assume what the authors are referring to is the way that a large quantity of training labels are pulled from the charts objectively? As compared to manually. Please make this more clear.
  Corrected.

- page 7: line 33: The method is semi-automated in part because of the way the samples are generated, but also because of the use of manually generated ice charts as training data.
  Generation of the reference ice chart is not a part of this algorithm. Our algorithm uses the already exist ice charts made by credible ice experts who are not involved in this study.

- page 11: lines 5-6: `This is because day of year might not correspond to the same temperature...' - Might be better stated `because SAR image features, which partially reflect temperature fluxes and weather regimes, might not correspond to day of year.
  Corrected.

- page 11: line 8: Please refer to the figure number.
  Corrected.

- Figure 8: there are clearly some problems regarding wind roughened water (misclassifications in Fig 8) - The authors acknowledge this, but don't suggest a future path to resolve this issue. There is also some water identified in what might be ice cover in this same figure.
  Added explanations.

English language (not an exhaustive list):

- abstract: line 18, dataset → datasets
  Corrected.
- page 1: line 7, is impacting seriously → can seriously impact
  Corrected.
- page 2: line12, statistics thus → statistics, and thus
  Corrected.
- page 2: line 14: works → studies
  Corrected.
- page 3: line 9: images from the first year is → images from the first year are
  Corrected.
- page 3: line10: is used for validation → are used for validation
  Corrected.
- page 3: line 22: each steps → each step
  Corrected.
- page 6: each polarization images → each polarization image
  Corrected.
- page 6: 'In the literatures about sea ice classification' → Previous studies about sea ice classification
  Corrected.
- page 8: line4, `better recognizable' → 'more easily identified'
  Corrected.
- page 8: line 11 'raters' → rasters
  Corrected.
- page 8: line 11, `trained classifier' → output from the trained classifier
  Corrected.
- page 9: line 32, Do not start a sentence with a Greek letter (please correct at other locations in the manuscript).
  Corrected.
- page 10: line 31, `ices' → ice
  Corrected.
- page 11: line 11: ANSWRS (use of acronym).
  Corrected.

**Anonymous Referee #3**

Authors Responses to my major comments are rather disappointing:
Example 1:
My comment: A seasonal assessment of the classification scheme is missing. It the most important issue to address and without this assessment it would not be reasonable to claim the scheme to be either operational or innovative.
Authors Reply: As the reviewer pointed, we did not conduct seasonal assessment. Since the developed algorithm was tested for winter season only, we changed the title as "Classification of Winter Sea Ice Types in Sentinel-1 SAR images"
A similar kind of response also regarding the comments on the usage of ice charts to prepare the training datasets. As I mentioned, I am disappointed with the responses (e.g. A new title, rather than investigating the summer season), but I leave the final decision to the Editor.

First of all, we sincerely apologize for our previous response that made you disappointed. We extended our experiment to assess the performance of the proposed scheme in summer.

As a summary of the new experiment (summer),
- Even the three-class classifier (open water, mixed first-year ice, and old ice) failed to classify different ice types properly. The classification accuracies for old ice were higher than 90% while those for mixed first-year ice were 11-26% with different sets of feature configuration. The accuracy for open water was very high (up to 99%).
- The misclassification between mixed first-year ice and old ice was among themselves.
- These imply that the trained classifier acts like an ice-water discriminator rather than ice type classifier.

The low accuracy for mixed first-year ice might be because of the surface melting and the corresponding image textures which makes the discrimination between the mixed first-year ice and old ice difficult. Another reason might be related to the limitation of the use of microwave region. We compared the output of our three-class classifiers against the OSI SAF sea ice type product, OSI-403-c, in winter season (OSI-403-c does not provide ice types in summer season) and the result was similar to those in the summary with bullet points above. What interesting is, the extent of old ice (multi-year ice) is considerably narrow in OSI-403-c compared to that in NIC ice chart. As the OSI-403-c was produced from microwave observations only (passive microwave radiometers and radar scatterometer), the characteristics of the observed signals may have similarity to those from SAR.

We hope that your concern about the seasonal assessment will be cleared after reading our responses and modifications made to the manuscript. Please find below our modifications (deleted in red and added in blue).
* * *
**[page 1, lines 19-24]**

[revised manuscript text omitted]

---

## Author Response (AR3)

**General comments**

The manuscript has now been improved since the last round of revisions, however some changes still needs to be made before it can be accepted for publication.

First of all, we would like to thank the reviewer for the positive evaluation and providing important comments. We hope that all your concerns will be cleared after reading our responses and modifications made to the manuscript. Please find below our answers (in green) and modifications (deleted in red and added in blue) to your comments/suggestions/questions.

**Specific comments**

How is the overall accuracy calculated? If only 12% of the potentially dominant (?) class mFYI is correct how does this affect the overall accuracy of the algorithm? And if the algorithm now largely for summer is an open water vs sea ice classification, should it be clarified in the abstract that the method is not really a sea ice type classifier.

The overall accuracy was calculated by taking an unweighted average of each of the correctly predicted types over the entire study area. If it is evaluated over an area covered predominantly with mFYI for which our classifier showed the worst performance, the overall accuracy will decrease; however, the inverse is true if the area is filled with OW or MYI of which accuracies are relatively high. In the revised manuscript, the failure in mFYI classification for summer is explicitly presented.

> **[Abstract, page 1, lines 22-24]**
>
> respectively. For summer season, the classifier failed in distinguishing mixed first-year ice from old ice with accuracy of only 12%; however, it performed rather like an ice-water discriminator with high accuracy of 98% as the misclassification between the mixed first-year ice and old was among themselves. The accuracy for five cover types (open water, new ice, young ice,
>
> **[Conclusion, page 13, lines 31-32]**
>
> classifiers. Considering the misclassifications in different ice types were among themselves, the three-class classifiers are not really a sea ice type classifier but they performed well at least as an ice-water discriminator with accuracy of 98%.

As the comparison with OSI SAF data now is quite a large section of the manuscript, consider moving some of the explanations on page 12 to the methods section, e.g. a validation section.

A part of the explanations was moved to the methods section, 2.2.6.

> **[page 9, lines 8-12]**
>
> It might be not enough to assess the quality of the classifier output when it is trained with, and evaluated against, only NIC ice charts. The accuracy could be indirectly investigated by comparing the output from our classifier against another data source, such as OSI SAF sea ice type product (OSI-403-c). The ice classes of OSI-403-c are assigned from atmospherically corrected brightness temperatures of passive microwave radiometers (SSMIS and AMSR2) and backscatter values of radar scatterometer (ASCAT), using a Bayesian approach (Aaboe et al., 2018).

Moreover, I suggest to convert old ice into MYI or vice versa when comparing the two results. Just clearly state what has been done.

The following explanation was added to the revised manuscript.

> **[page 12, lines 16-18]**
>
> OSI-403-c product as reference. For one-to-one comparison, it was assumed that the ideal characteristics of the mixed first-year ice and the old ice in our three-class classification are equivalent to those of the first-year ice and the multi-year ice in OSI-403-c. Comparing with the results in Table 911, the accuracies for open water decreased from by 6%; however, this is
>
> **[Table 12]**
>
> Table 1012: Confusion matrix of the three-class RF classifier which was trained with DS1 winter dataset and applied to the DS2 winter dataset with reference to OSI SAF sea ice type product (OSI-403-c)
>
> | | | Predicted (classifier was trained with NIC ice chart) | | | | | | | | |
> | --- | --- | --- | --- | --- | --- | --- | --- | --- | --- | --- |
> | | | Open water | | | Mixed firstFirst-year ice | | | Old Multi-year ice | | |
> | | Case | FC1 | FC2 | FC3 | FC1 | FC2 | FC3 | FC1 | FC2 | FC3 |
> | Reference (OSI SAF) | Open water | **85.9** | **86.1** | **86.2** | 12.6 | 12.4 | 12.1 | 15.7 | 15.3 | 16.2 |
> | | First-year ice | 1.9 | 1.6 | 2.0 | **26.0** | **26.8** | **26.9** | 72.1 | 71.6 | 71.2 |
> | | Multi-year ice | 0.1 | 0.1 | 0.1 | 1.5 | 1.4 | 1.4 | **98.4** | **98.5** | **98.5** |

You claim that the reason for low accuracy for FYI in your method compared to the OSI-403-c is that "A large extent of old ice in NIC ice chart is annotated as multi-year ice in the OSI-403-c. This does not clarify why your algorithm has a large error for FYI classification. Please clarify.

A more extensive comparison of NIC ice chart and OSI-403c was done, and the corresponding results were added to the revised manuscript.

**[page 12, lines 15-33]**

Table  12 shows the confusion matrices for our three-class classifiers when their prediction results are compared with the OSI-403-c product as reference. For one-to-one comparison, it was assumed that the ideal characteristics of the mixed first-year ice and the old ice in our three-class classification are equivalent to those of the first-year ice and the multi-year ice in OSI-403-c. Comparing with the results in Table 11, the accuracies for open water decreased from by 6%; however, this is mainly because the ice concentration threshold for ice-water discrimination in OSI-403-c is 35% which is higher than 20% that we set in our preprocessing of NIC ice chart (Section 2.2.1), thus areas with low ice concentration in marginal ice zone are most likely annotated as open water in OSI-403-c. The accuracies for open water at points in the NIC charts with ice concentration between 20% and 40% only were considerably lower, with 67.4%, 67.8%, 70.1% for FC1, FC2, and FC3, respectively (not presented in Table 12). For first-year ice, large portions (72%) are misclassified as old ice. This might be partly explained from the Figure 12, which shows the ice classes in NIC ice chart and OSI-403-c for the same publication date. A large extent of old ice in NIC ice chart is annotated as multi-year ice in OSI-403-c. As our classifiers were trained with NIC ice chart, it is natural to result in more multi-year ice for the area where the ice type is classified as first-year ice in OSI-403-c. For  multi-year ice, the accuracy was the highest, 98%.

The inconsistency in ice types between NIC ice chart and OSI-403-c seems persistent at least for the time coverage of DS2 (January-March in 2019). Table 13 shows averaged percent agreement of the two sea ice type products for the same publication dates over 12 weeks (12 one-to-one comparisons as the NIC ice chart is a weekly product). To make a fair comparison, the ice-covered areas with ice concentrations lower than 35%, which is the threshold for ice-water discrimination in OSI-403-c, were excluded. The percent agreement for first-year ice (58.8%) was much lower than those of open water (90.0%) and multi-year ice (99.0%), which is inline with the results in Table 12. Finding the reason for the clear discrepancy of the extent of first-

**[Table 13]**

**Table 13: Averaged percent agreement of NIC weekly ice chart and OSI SAF daily sea ice type product (OSI-403-c) for the same publication dates (12 different days) in the studied domain during January-March, 2019**

|  |  | NIC | | |
|---|---|---|---|---|
|  |  | Open water | First-year ice | Multi-year ice |
| OSI SAF | Open water | **90.0** | 10.0 | 0.1 |
|  | First-year ice | 0.9 | **58.8** | 40.3 |
|  | Multi-year ice | 0.0 | 1.0 | **99.0** |

A recently accepted study by Lohse et al. 2020 investigated slopes for Sentinel-1 images and found slopes of: -0.27 (HH) and -0.26 (HV) for level FYI, -0.23 (HH and HV) for MYI and for open water -0.72 (HH) and -0.33 (HV). Consider including these as well in your comparison given that they are covering the same area (though they have a Pan-Arctic coverage) and are also using Sentinel-1 EW images. Within their work they are using 80 Sentinel-1 EW images to establish these slopes and are using overlapping optical images and expert opinions from the Norwegian Ice service to establish the training and validation data. Consider also including this among other recent sea ice type studies as it is pointed out in the paper that such studies using Sentinel-1 images are missing.

Thank you for the information. It is great to know about the recent estimates over a Pan-Arctic coverage from Sentinel-1.

> **[page 5, lines 9-12]**
>
> varying backscatter intensity confuses image interpretation. The quasi-linear slopes in the plane of incidence angle versus sigma nought in decibel scale  are reported as -0.24 and -0.16 dB/degree for typical first-year ice (Mäkynen and Karvonen, 2017), -0.27 and -0.26 dB/degree for level first-year ice, and -0.23 and -0.23 dB/degree for multi-year ice (Lohse et al., 2020)  in HH- and HV-polarization, respectively . To normalize the

**Technical comments**

Within the text when others use sea ice experts to help establish training and validation data they are referred to as: e.g. informal sources (P13, R3). Yet when you are using potentially the same experts they are referred to as: credible sources (P4, R1). In my opinion the experts are equally credible, please revise accordingly.

Revised.

> **[page 13, lines 19-22]**
>
> feasible in summer season. The main technological innovation is two-fold: i)  reduced manual work in the preparation of large amount of training and validation reference data using readily available public ice charts and ii) more objective evaluation of the SAR-based sea ice type classifier compared to the previous studies conducted with small number of images and customized ice type references from  sources not open to the public. A conventional approach for

P1R.12 works -> work

Corrected.

P1R19. Two?

Corrected.

P1R22-23. Unclear sentence please revise.

Revised.

> **[page 1, lines 22-24]**
>
> respectively. For summer season, the classifier failed in distinguishing mixed first-year ice from old ice with accuracy of only 12%; however, it performed rather like an ice-water discriminator with high accuracy of 98% as the misclassification between the mixed first-year ice and old was among themselves. The accuracy for five cover types (open water, new ice, young ice,

P5 R1. Consider rewriting this sentence. HV generally has lower backscatter values but the noise subtraction does not especially yield negative values in HV, low values in both channels will be equally affected.

Revised.

> **[page 5, lines 2-4]**
>
> images was denoised before further processes are applied. As the noise power subtraction yields negative intensity values where the backscattering power is close to the noise floor,  more often in HV polarization which has lower backscatter than in HH polarization, we added mean of the noise power back to the denoised result so that those pixels do not

P5 R11. Consider adding also Lohse et al., 2020 here.

Added.

P10 R25. The FYI also has a small accuracy decrease here, please revise.

The accuracies for FYI remained almost the same (-0.2% to +0.9%). To avoid confusion, the word "to" was used to indicate a range of values for a quantity instead of a range dash.

> **[page 10, lines 25-28]**
>
> evaluation were conducted for the same dataset without applying the textural noise correction (Table 5). In all configurations (FC1-FC3), the accuracies improved for young ice (+2.4% to +6.2%) and old ice (+3.9% to +4.5%) which were most pronounced compared to those for open water (+1.1% to +1.9%) and first-year ice (-0.2% to +0.9%). On the contrary, a small accuracy decrease was observed for new ice (=2.4% to -0.3%). Nevertheless, the

P12 R22-23. Unclear sentence please revise.

Revised.

> **[page 13, lines 19-22]**
>
> feasible in summer season. The main technological innovation is two-fold: i)  reduced manual work in the preparation of large amount of training and validation reference data using readily available public ice charts and ii) more objective evaluation of the SAR-based sea ice type classifier compared to the previous studies conducted with small number of images and customized ice type references from  sources not open to the public. A conventional approach for

**Anonymous Referee #2**

**General comments**

The authors have addressed my previous comments. Further comments given below:

First of all, we would like to thank the reviewer for the positive evaluation and providing important comments. We hope that all your concerns will be cleared after reading our responses and modifications made to the manuscript. Please find below our answers (in green) and modifications (deleted in red and added in blue) to your comments/suggestions/questions.

**Specific comments**

1. Although the authors did carry out a comparison between the NIC charts and the OSI-SAF charts, their results do not go into depth and are inconclusive. For example, it is stated that the accuracy of open water decreases because of the different ice concentration thresholds used for the OSI-SAF charts and for their classification method, but this is not shown. For example, it could be shown by retraining their classifier to use a different threshold, or by systematically comparing the misclassification rates for points in the NIC charts with ice concentration between 20% and 30% or 40% with misclassification rates for points with higher ice concentrations. On a similar note, for the high misclassification rate of first year ice, only one date is shown to demonstrate that the ice classes are different between the NIC charts and the OSI-SAF charts. It would be good to know how widespread these differences are.

By adopting the reviewer's suggestion, the classification accuracies for open water at points in the NIC charts with ice concentration between 20% and 40% were examined, and the results were added to the revised manuscript.
* * *
**[page 12, lines 18-23]**

OSI-403-c. Comparing with the results in Table 11, the accuracies for open water decreased  by 6%; however, this is mainly because the ice concentration threshold for ice-water discrimination in OSI-403-c is 35% which is higher than 20% that we set in our preprocessing of NIC ice chart (Section 2.2.1), thus areas with low ice concentration in marginal ice zone are most likely annotated as open water in OSI-403-c. The accuracies for open water at points in the NIC charts with ice concentration between 20% and 40% only were considerably lower, with 67.4%, 67.8%, 70.1% for FC1, FC2, and FC3, respectively (not presented in Table 12). For first-year ice, large portions (72%) are misclassified as old ice. This might be

**[Table 12]**

Table 12: Confusion matrix of the three-class RF classifier which was trained with DS1 winter dataset and applied to the DS2 winter dataset with reference to OSI SAF sea ice type product (OSI-403-c)

| | | Predicted (classifier was trained with NIC ice chart) | | | | | | | | |
| --- | --- | --- | --- | --- | --- | --- | --- | --- | --- | --- |
| | | Open water | | | First-year ice | | | Multi-year ice | | |
| | Case | FC1 | FC2 | FC3 | FC1 | FC2 | FC3 | FC1 | FC2 | FC3 |
| Reference (OSI SAF) | Open water | **85.9** | **86.1** | **86.2** | 12.6 | 12.4 | 12.1 | 15.7 | 15.3 | 16.2 |
| | First-year ice | 1.9 | 1.6 | 2.0 | **26.0** | **26.8** | **26.9** | 72.1 | 71.6 | 71.2 |
| | Multi-year ice | 0.1 | 0.1 | 0.1 | 1.5 | 1.4 | 1.4 | **98.4** | **98.5** | **98.5** |

And also as the reviewer pointed, a more extensive comparison of the NIC ice charts and the OSI-SAF charts is required to confirm the high misclassification rate of first-year ice. We compared the two ice charts acquired over the entire period of DS2 dataset and the results were added to the revised manuscript.

**[page 12, lines 28-33]**

The inconsistency in ice types between NIC ice chart and OSI-403-c seems persistent at least for the time coverage of DS2 (January-March in 2019). Table 13 shows averaged percent agreement of the two sea ice type products for the same publication dates over 12 weeks (12 one-to-one comparisons as the NIC ice chart is a weekly product). To make a fair comparison, the ice-covered areas with ice concentrations lower than 35%, which is the threshold for ice-water discrimination in OSI-403-c, were excluded. The percent agreement for first-year ice (58.8%) was much lower than those of open water (90.0%) and multi-year ice (99.0%), which is inline with the results in Table 12. Finding the reason for the clear discrepancy of the extent of first-

**[Table 13]**

Table 13: Averaged percent agreement of NIC weekly ice chart and OSI SAF daily sea ice type product (OSI-403-c) for the same publication dates (12 different days) in the studied domain during January-March, 2019

| | | NIC | | |
| --- | --- | --- | --- | --- |
| | | Open water | First-year ice | Multi-year ice |
| OSI SAF | Open water | **90.0** | 10.0 | 0.1 |
| | First-year ice | 0.9 | **58.8** | 40.3 |
| | Multi-year ice | 0.0 | 1.0 | **99.0** |

2. page 10 lines 10-14 - here the authors compare their scores for the various classifier configurations for DS1 and DS2 (Tables 6 and 7). The fact that FC3 is overfitting for DS1, means that the accuracies for mFYI are very low (disregard the column for FC3 for Table 6), and there is considerable mFYI that is misclassified as old ice. It is not clear at all as to whether this problem (eg. 15% for mFYI for DS1) is due to the classifier itself or the data. Similar to the previous comment, it would be ideal to have a more thorough analysis of the ice charts themselves to unravel this problem, although I agree this could be left to a later study.

By adopting the reviewer's suggestion, we checked where the problem resides in the results, and the following analysis was added to the revised manuscript.

**[page 11, lines 2-15]**

[revised manuscript text omitted]

4. I don't find the present manuscript demonstrates 'minimized manual work in the preparation of training and validation reference data' as the NIC charts are manual analyses, and the method itself also uses a manual step to select the training data.

Revised.

**[page 13, lines 19-22]**

feasible in summer season. The main technological innovation is two-fold: i)  reduced manual work in the preparation of large amount of training and validation reference data using readily available public ice charts and ii) more objective evaluation of the SAR-based sea ice type classifier compared to the previous studies conducted with small number of images and customized ice type references from  sources not open to the public. A conventional approach for

**Technical comments**

page 1 line 12 works should be work

Corrected.

page 1 line 12 `reduce inconsistent decisions' - this was not shown. Please remove.

Revised.

**[page 1, lines 12-14]**

amount of manual  work in preparation of large amounts of training/testing data. Furthermore, they  can feed highly reliable data to the  trainer by indirectly exploiting the best ability of the sea ice experts working at the operational ice services. The proposed scheme has two phases: training and

page 2 line 20 and line 25 - what is the difference between a 'manual' interpretation and a 'raw' interpretation.

Revised.
* * *
**[page 2, lines 25-27]**

classification process is encouraged. The idea of training using SAR images and accompanying image analysis charts, which is a  direct manual interpretation of SAR images by trained ice analysts working at operational ice services, were tested for sea ice concentration estimation by Wang et al. (2017); however, such image analysis charts are not accessible to the public.
* * *
**Anonymous Referee #3**

**General comments**

My major concern related to the melt/summer season is now addressed adequately.

We sincerely appreciate your comments and suggestions given to us in the previous rounds. We are happy to have made you satisfied with the revised manuscript.

[revised manuscript text omitted]